# Long-range inhibition synchronizes and updates prefrontal task activity

Kathleen K. A. Cho[1,2,3✉], Jingcheng Shi[1,2], Aarron J. Phensy[1,2], Marc L. Turner[1,2] & Vikaas S. Sohal[1,2✉]

Changes in patterns of activity within the medial prefrontal cortex enable rodents, non-human primates and humans to update their behaviour to adapt to changes in the environment—for example, during cognitive tasks[1–5]. Parvalbumin-expressing inhibitory neurons in the medial prefrontal cortex are important for learning new strategies during a rule-shift task[6–8], but the circuit interactions that switch prefrontal network dynamics from maintaining to updating task-related patterns of activity remain unknown. Here we describe a mechanism that links parvalbumin-expressing neurons, a new callosal inhibitory connection, and changes in task representations. Whereas nonspecifically inhibiting all callosal projections does not prevent mice from learning rule shifts or disrupt the evolution of activity patterns, selectively inhibiting only callosal projections of parvalbumin-expressing neurons impairs rule-shift learning, desynchronizes the gamma-frequency activity that is necessary for learning[8] and suppresses the reorganization of prefrontal activity patterns that normally accompanies rule-shift learning. This dissociation reveals how callosal parvalbumin-expressing projections switch the operating mode of prefrontal circuits from maintenance to updating by transmitting gamma synchrony and gating the ability of other callosal inputs to maintain previously established neural representations. Thus, callosal projections originating from parvalbumin-expressing neurons represent a key circuit locus for understanding and correcting the deficits in behavioural flexibility and gamma synchrony that have been implicated in schizophrenia and related conditions[9,10].

Organisms must continually update their behavioural strategies to adapt to changes in the environment. Inappropriate perseveration on outdated strategies is a hallmark of conditions such as schizophrenia and bipolar disorder, and classically manifests in the Wisconsin card sorting task[11] (WCST). It is well documented that the prefrontal cortex is responsible for such flexible cognitive control, by providing active maintenance of rule or goal representations[12–14], adaptive gating of these representations[15], and the top-down biasing of sensory processing via extensive interconnectivity with other brain regions[16,17]. Studies have shown that within the medial prefrontal cortex (mPFC), normal parvalbumin-expressing (PV) interneuron function is required for mice to perform 'rule-shift' tasks, which, similar to the WCST, involve identifying uncued rule changes and learning new rules that use cues that were previously irrelevant to trial outcomes[6,7]. Moreover, PV interneurons have a key role in generating synchronized rhythmic activity in the gamma-frequency (around 40 Hz) range[18,19]. Indeed, during rule-shift tasks, the synchrony of gamma-frequency activity between PV interneurons in the left and right mPFC increases after error trials—that is, when mice receive feedback that a previously learned rule has become outdated—and optogenetically disrupting this synchronization causes

perseveration[8]. Nevertheless, the basic relationships between circuits (synaptic connections), network dynamics (interhemispheric gamma synchrony) and neural representations (task-dependent changes in activity patterns) remain unknown.

## Callosal projections from PFC PV neurons

Gamma synchrony is commonly assumed to be transmitted across regions by excitatory synapses[20], which are the predominant form of long-range communication in the cortex. However, we explored an alternative hypothesis suggested by recent descriptions of long-range γ-aminobutyric-acid-releasing (GABAergic) connections originating from PV neurons in mPFC[21]. Specifically, we first demonstrated that PV-expressing neurons in the mPFC give rise to callosal GABAergic synapses in the contralateral mPFC (Fig. 1). We identified this anatomical link by injecting AAV-EF1α-DIO-ChR2-eYFP into one mPFC of PV-cre mice and observed virally labelled PV terminals in the contralateral PFC, particularly in deep layers 5 and 6 (Fig. 1a). To characterize this callosal PV projection and its recipient neurons, we performed recordings in the contralateral mPFC (Fig. 1b). We found that callosal PV projections

[1]Department of Psychiatry and Behavioral Sciences, Weill Institute for Neurosciences, University of California, San Francisco, San Francisco, CA, USA. [2]Kavli Institute for Fundamental Neuroscience, University of California, San Francisco, San Francisco, CA, USA. [3]Present address: Institut du Cerveau-Paris Brain Institute, Sorbonne Université, Inserm U1127–CNRS UMR 7225, Paris, France. ✉e-mail: kathleen.cho@inserm.fr; vikaas.sohal@ucsf.edu

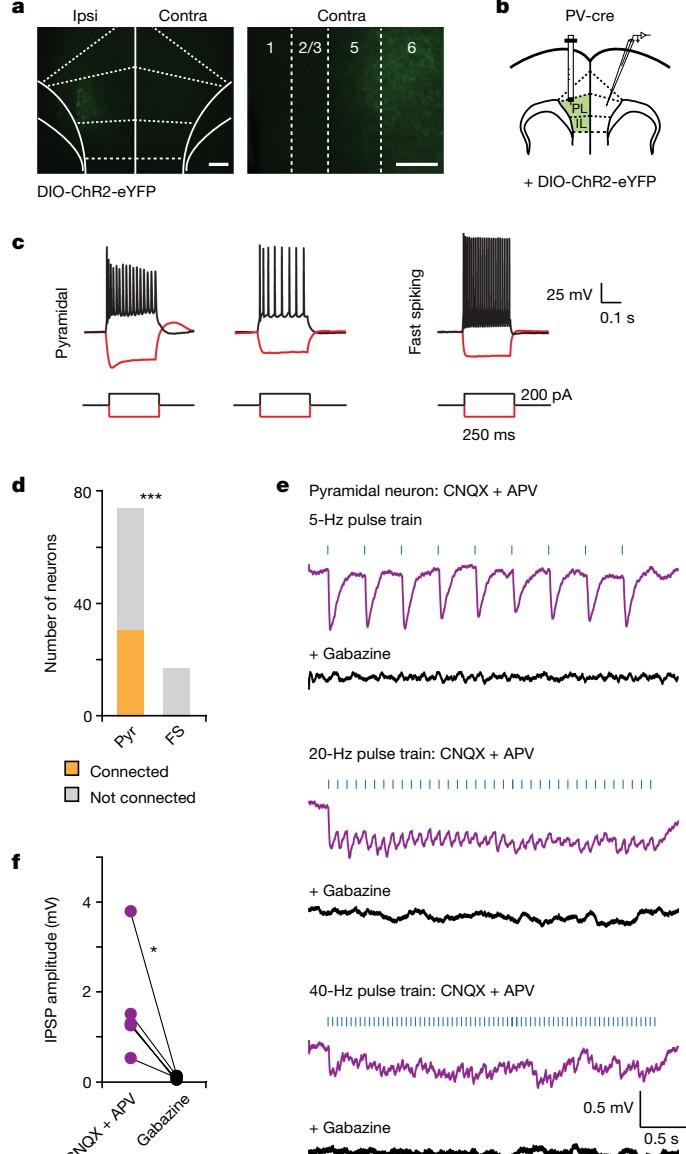

**Fig. 1 | Callosal mPFC PV projections preferentially target pyramidal neurons. a**, Injection of AAV-DIO-ChR2-eYFP in one (ipsilateral (ipsi)) mPFC of PV-cre mice enabled visualization of eYFP⁺ terminals in the contralateral (contra) mPFC. Scale bars: 250 µm (**a**, left) and 100 µm (**a**, right). **b**, Experimental design schematic. Whole-cell recordings were made contralateral to the virus injection in prefrontal brain slices. **c**, Example current-clamp responses from potential recipient neurons during injection of hyperpolarizing or depolarizing current. **d**, During optogenetic stimulation of callosal PV terminals, we observed synaptic responses in pyramidal (Pyr) neurons but not in fast-spiking (FS) interneurons (two-sided chi-square test, $P = 0.0008$; $n = 93$ cells, 14 mice). **e**, Rhythmic trains of blue light flashes (5 ms, 470 nm; denoted as blue bars) delivered through the 40× objective were used to optogenetically stimulate callosal PV terminals. Example recordings from a recipient pyramidal neuron showing optogenetically evoked IPSPs in the presence of CNQX and APV that are abolished with gabazine. **f**, IPSPs are blocked by gabazine application (two-tailed paired $t$-test, $P = 0.0419$; $n = 5$ cells, 4 mice). *$P < 0.05$, ***$P < 0.001$.

innervate pyramidal neurons (identified on the basis of their morphology and non-fast spiking physiology; 31 out of 75 connected) but not fast-spiking neurons (0 out of 18 connected) (Fig. 1d). Rhythmic trains of PV-terminal optogenetic stimulation elicited time-locked inhibitory postsynaptic potentials (IPSPs) in ChR2-negative pyramidal neurons (Fig. 1e), which were not blocked by the glutamatergic

receptor antagonists 6-cyano-7-nitroquinoxaline-2,3-dione disodium salt (CNQX) (10 µM) and D-2-amino-5-phosphonopentanoic acid (APV) (50 µM), but were completely abolished by bath application of the type A γ-aminobutyric acid (GABA_A) receptor antagonist gabazine (10 µM; Fig. 1e,f). To further characterize the specific targets of callosal mPFC PV synapses, we injected fluorescent dye-conjugated cholera toxin subunit B (CTb) into four downstream targets of mPFC, then recorded from retrogradely labelled mPFC neurons that project to the contralateral mPFC (that is, contralateral to where recordings were performed and ipsilateral to the AAV-DIO-ChR2 injection), dorsal striatum, mediodorsal (MD) thalamus, or nucleus accumbens (NAc) (Extended Data Fig. 1b–k). After a single 5-ms light pulse to optogenetically activate callosal PV⁺ terminals in the presence of glutamatergic antagonists (20 µM 6,7-dinitroquinoxaline-2,3-dione (DNQX) and 50 µM APV), we observed time-locked IPSPs in 22 out of 22 MD-projecting pyramidal neurons, compared with 0 out of 18 callosally projecting pyramidal neurons, 0 out of 21 accumbens-projecting pyramidal neurons, and 7 out of 24 dorsal striatum-projecting pyramidal neurons.

## Rule-shift learning

Although we identified a long-range inhibitory anatomical connection between the prefrontal cortices, the function of this input was not defined. We next explored the role of these callosal mPFC PV⁺ projections as mice performed a task involving the type of behavioural adaptations involved in the WCST. Variants of this task have previously been characterized[6,7,22] in which mice are first required to associate one set of sensory cues with the location of a hidden food reward (initial association). After learning the initial association, the mice must then learn to attend to a set of sensory cues that were previously present, but irrelevant to the outcome of each trial, and make decisions based on these previously irrelevant cues (rule shift) (Fig. 2a and Extended Data Fig. 2). Because cross-hemispheric gamma synchrony between PV interneurons is essential for learning the rule shift[8], callosal PV⁺ projections may be a good candidate to effectively coordinate activity across hemispheres. To investigate this, we injected AAV-EF1α-DIO-eNpHR-mCherry or control virus into one mPFC of PV-cre mice, then optogenetically silenced callosal PV⁺ terminals in the contralateral mPFC during the rule-shift portion of the task (Fig. 2b–f and Extended Data Figs. 2b and 5). Selective inhibition of callosal PV⁺ terminals is sufficient to impair rule-shift learning and induce perseveration (relative to PV-cre mice injected with a control virus; Fig. 2e,f and Extended Data Figs. 3a and 5a–h). Notably, nonspecific inhibition of all callosal projections (both GABAergic and excitatory connections) did not affect rule-shift performance (Fig. 2g–j and Extended Data Figs. 3b and 5i–p). As controls, we verified that selective inhibition of callosal PV+ terminals did not affect gross motor behaviour during rule shifts (Extended Data Fig. 4a–c,j–m and Supplementary Video 1) or performance in an intra-dimensional rule reversal (which has similar task mechanics to the rule shift but differs in that it does not depend on mPFC gamma synchrony[8]) (Extended Data Fig. 6).

Next we tested whether the behavioural deficits induced by inhibiting callosal PV⁺ projections might be persistent and reversible. For this, we took advantage of an intersectional strategy that used a Flp- and Cre-dependent AAV in PV-cre mice to drive NpHR3.3 (NpHR) and ChR2 expression specifically in callosally projecting PV neurons[23]. To label and manipulate callosal PV⁺ projections, we injected AAVretro-Flp in one mPFC of PV-cre mice, and in the contralateral mPFC we implanted an optical fibre and injected the NpHR and ChR2 Flp- and Cre-dependent AAVs (Extended Data Fig. 7). Once again, silencing callosal PV⁺ projections impaired rule-shift performance on day 1 (Extended Data Fig. 7h–j). Unexpectedly, the impairments in rule-shift performance persisted on day 2, in the absence of any further optogenetic inhibition. On day 3, 40-Hz stimulation of callosal PV⁺ projections rescued perseveration and these effects persisted in the absence of further

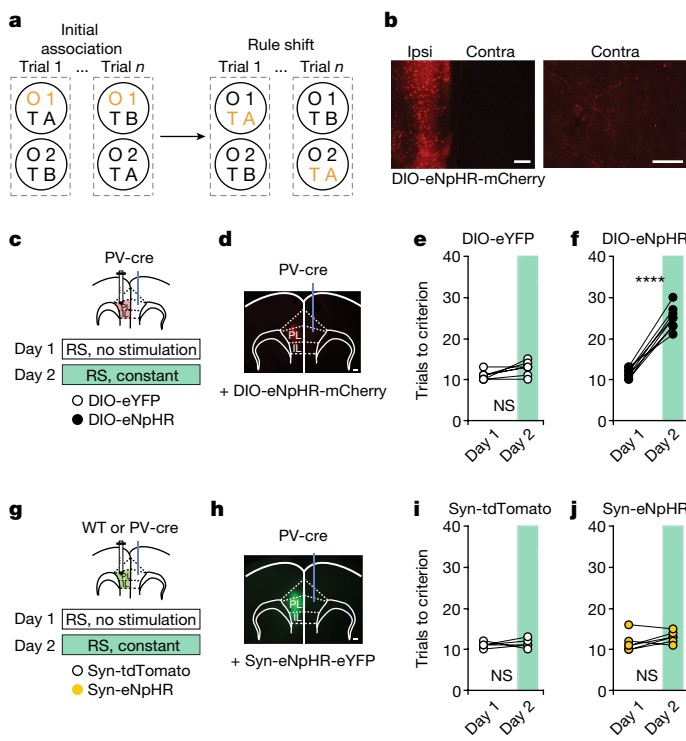

**Fig. 2 | Optogenetic inhibition of callosal mPFC PV projections impairs cognitive flexibility. a**, Schematic of the rule-shift task. On each trial, a mouse chooses one of two bowls, each scented with a different odour (O1 or O2) and filled with a different textured digging medium (TA or TB), to find a food reward. Mice first learn an initial association between one of these sensory cues (for example, O1) and food reward (the cue associated with reward is indicated in orange). Once mice reach the learning criterion (8 out of 10 consecutive trials correct), this association undergoes an extra-dimensional rule shift (RS; for example, from O1 to TA being rewarded). **b**, Representative images showing AAV-DIO-eNpHR-mCherry (DIO-eNpHR) expression in ipsilateral mPFC and callosal PV terminals in contralateral mPFC. Scale bars: 100 μm (left) and 50 μm (right). **c,g**, Experimental design with DIO-eYFP or DIO-eNpHR (**c**) or Syn-tdTomato or Syn-eNpHR (**g**) mice. Day 1, no light delivery; day 2, continuous light for inhibition during the rule shift. **d**, Representative image showing DIO−eNpHR expression in one mPFC and a fibre-optic cannula implanted in contralateral mPFC. Scale bar, 100 μm. **e,f**, Optogenetic inhibition of mPFC callosal PV terminals impairs rule-shift performance in DIO-eNpHR mice ($n = 8$) compared with controls ($n = 7$) (two-way ANOVA (task day × virus); interaction: $P < 0.0001$). **e**, Performance of DIO-eYFP controls did not change ($P = 0.11$). **f**, Inhibition disrupts rule-shift performance in DIO-eNpHR mice ($P < 0.0001$). **h**, Representative image showing Syn−eNpHR expression in one mPFC and a fibre-optic cannula implanted in contralateral mPFC. Scale bar, 100 μm. **i,j**, Performance of Syn-tdTomato controls ($n = 4$) is similar to Syn-eNpHR mice ($n = 6$; two-way ANOVA (task day × virus); interaction: $P = 0.38$). Two-way ANOVA with Bonferroni post hoc comparisons. ****$P < 0.0001$; NS, not significant.

optogenetic manipulations on day 4 of testing (which occurred at least 1 week later). In ChR2-negative controls (PV-cre mice injected with Flp- and Cre-dependent NpHR but mCherry instead of ChR2), rule-shift deficits induced by inhibiting callosal PV[+] projections persisted across days (Extended Data Fig. 7e−g). These results identify callosal PV[+] projections as powerful bi-directional drivers of network plasticity. Notably, long-range GABAergic projections have been hypothesized to have a role in the temporal coordination of neuronal activity[24,25]. On this basis, we hypothesized that the behavioural deficits that we observed may reflect the fact that when callosal PV[+] projections are selectively inhibited, callosal communication (mediated largely by excitatory connections) becomes uncoordinated, disrupting normal information processing.

## Gamma synchrony

To test this possibility, we studied how callosal PV[+] projections influence the cross-hemispheric gamma synchrony in PV interneurons that we previously found to be essential for re-appraising the behavioural salience of sensory cues during rule shifts. For this, we injected AAV-EF1α-DIO-eNpHR-BFP or control virus into one mPFC of PV-cre Ai14 mice (Fig. 3a,b and Extended Data Fig. 8a,b). To track transmembrane voltage activity patterns of PV neurons, we also injected virus expressing the Cre-dependent, genetically encoded voltage indicator Ace-mNeon (AAV-DIO-Ace2N-4AA-mNeon) into both mPFCs. We then implanted multimode optical fibres to excite and measure fluorescence from both Ace-mNeon and a control fluorophore (tdTomato) expressed in PV interneurons in the left and right mPFC as well as to optogenetically silence PV[+] callosal terminals (Fig. 3a,b and Extended Data Fig. 2c). Using this method, we had previously found that gamma synchrony between PV interneurons in the left and right mPFC increases during rule shifts when mice make errors—that is, when they do not receive an expected reward and therefore receive feedback that the previously learned association is no longer valid[8]. We confirmed this result on day 1 (Fig. 3c–g), in the absence of any optogenetic silencing. On day 2, optogenetic inhibition of callosal PV[+] terminals disrupted performance in the rule-shift task, compared with PV-cre Ai14 mice given control virus (Fig. 3c–e and Extended Data Fig. 8a–h). This behavioural deficit, which was consistent with our earlier experiments, correlated with deficits in cross-hemispheric gamma synchrony: specifically, the increase in cross-hemispheric PV neuron gamma synchrony observed after rule-shift errors on day 1 was abolished on day 2 (Fig. 3f–g). Moreover, the impairments in both rule-shift performance and gamma synchrony persisted on day 3, in the absence of any additional optogenetic inhibition (Fig. 3e,g).

Given that inhibiting all callosal projections, using the synapsin promoter, does not affect rule-shift performance, we tested how it would affect gamma synchrony. Again, we observed that this manipulation did not affect rule-shift performance (Fig. 3h–l and Extended Data Fig. 8i–p). Notably, silencing all callosal communication did disrupt normal increases in interhemispheric gamma synchrony following error trials on day 2 (Fig. 3m–n). However, this deficit did not persist in the absence of further optogenetic inhibition on day 3 (Fig. 3m–n). This is consistent with our model, in which the function of long-range GABA projections and gamma synchrony is not to facilitate essential inter-regional communication, but rather to prevent such communication from occurring in an aberrant manner that deleteriously impacts the normal evolution of prefrontal network activity during rule shifts. In this model, the transient loss of gamma synchrony that occurs during nonspecific inhibition of all callosal projections is not problematic because all callosal communication is suppressed. Furthermore, in this case, deficits in gamma synchrony do not persist, indicating that these persistent deficits reflect circuit plasticity that is triggered by inhibiting callosal PV[+] projections but requires the presence of activity in other callosal projections. In addition, inhibiting callosal PV[+] projections also disrupts increases in gamma synchrony occurring during inter-trial intervals (ITI) following rule-shift errors (Extended Data Fig. 9e).

## Updating of activity patterns

We specifically hypothesized that the inhibition of callosal PV[+] projections and consequent deficits in gamma synchrony would disrupt the changes in prefrontal activity patterns that normally serve to update behavioural strategies[3]. In particular, during normal rule-shift learning, gamma synchrony increases specifically following rule-shift errors. Therefore, we hypothesized that prefrontal activity patterns might begin to diverge from previously established representations specifically during this period, and that inhibiting callosal PV[+] projections would disrupt this process. First, using an alternative method

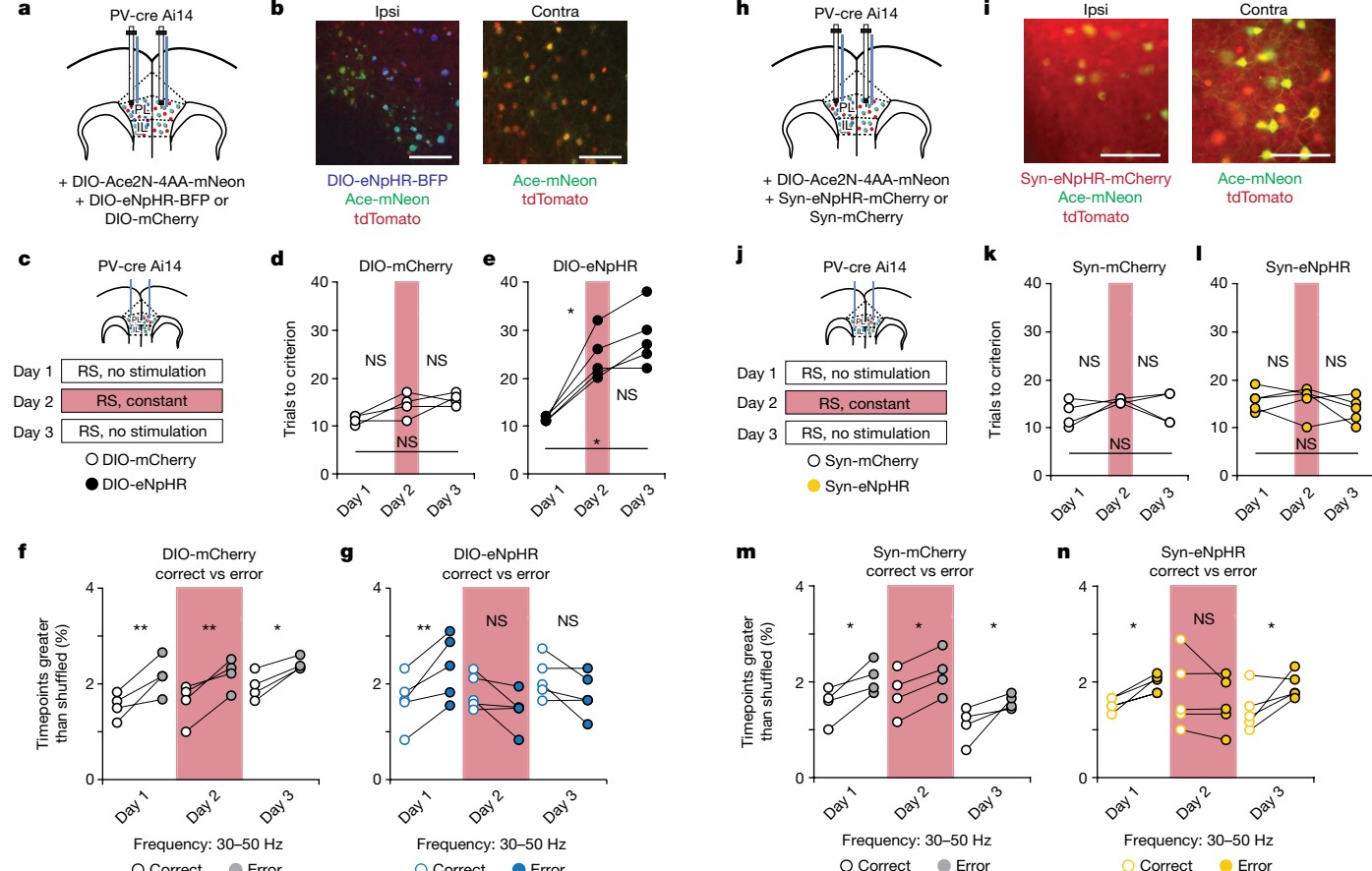

**Fig. 3 | Optogenetic inhibition of callosal mPFC PV projections disrupts interhemispheric gamma synchrony. a**, PV-cre Ai14 mice had bilateral AAV-DIO-Ace2N-4AA-mNeon injections, an ipsilateral AAV-DIO-eNpHR-BFP or control virus (AAV-DIO-mCherry) injection and fibre-optic implants in both prefrontal cortices. **b**, Representative images of DIO-eNpHR, Ace-mNeon and tdTomato expression in the ipsilateral (ipsi) versus contralateral (contra) mPFC. Scale bars, 50 μm. **c**, Experimental design: day 1, no light; day 2, continuous light (in contra) for inhibition during the rule shift; day 3, no light. **d**, Performance of DIO-mCherry mice (*n* = 4) did not change. **e**, Performance was impaired in DIO-eNpHR mice (*n* = 5). **f**, Gamma synchrony was higher after rule-shift errors than after rule-shift correct decisions across days in controls (*n* = 4 mice). **g**, Gamma synchrony was higher after rule-shift errors for DIO-eNpHR mice on day 1 (no light) but not on day 2 or day 3 (*n* = 5 mice). **h**, PV-cre

Ai14 mice had bilateral Ace-mNeon injections, an ipsilateral injection of AAV-Syn-eNpHR-mCherry or AAV-Syn-mCherry (control), and fibre-optic implants in both prefrontal cortices. **i**, Representative images of Syn-eNpHR, Ace-mNeon and tdTomato expression in ipsilateral and contralateral mPFC. Scale bars, 25 μm. **j**, Experiments using Syn-eNpHR with the same design as **c**. **k,l**, Performance of controls (*n* = 4 mice) and Syn-eNpHR mice (*n* = 5) did not change across days. **m**, Gamma synchrony was higher after rule-shift errors across days in controls (*n* = 4 mice). **n**, Gamma synchrony was higher after rule-shift errors for Syn-eNpHR mice on day 1 (no light). This was abolished with light on day 2, then restored on day 3 (no light) (*n* = 5 mice). Full statistics are provided in the Methods. Two-way ANOVA followed by Bonferroni post hoc comparisons. *$P < 0.05$, **$P < 0.01$.

to quantify gamma synchrony within TEMPO measurements on shorter time frames (Methods), we confirmed that interhemispheric gamma synchrony between mPFC PV neurons is higher specifically after digging on error trials compared with correct trials, and that inhibiting callosal PV$^+$ projections specifically disrupts increases in gamma synchrony during this period (Extended Data Fig. 9f–i). Then, to quantify the divergence of activity patterns during and immediately after this period of increased gamma synchrony, we measured the similarity between activity patterns during the 10 s following a choice on correct versus error trials. We measured these activity patterns using microendoscopic calcium imaging in PV-cre mice injected with AAV-synapsin-GCaMP7f (Syn-GCaMP7f) and implanted with a gradient-index (GRIN) lens in one mPFC, and injected in the other mPFC with either AAV-EF1α-DIO-eNpHR-mCherry (DIO-eNpHR), AAV-synapsin-eNpHR-mCherry (Syn-eNpHR) or a control virus (AAV-DIO-mCherry in PV-cre mice or AAV-synapsin-mCherry in wild-type mice) (Fig. 4a–c and Extended Data Fig. 10). We measured Syn-GCaMP7f fluorescence while mice performed a rule shift on three

consecutive days (Fig. 4a). On the second day, we also delivered red light to activate eNpHR in callosal terminals (Extended Data Fig. 2d). As in previous experiments, specifically inhibiting callosal PV$^+$ terminals significantly increased the number of trials needed to learn a rule shift in eNpHR-expressing mice (whereas there was no effect in controls) (Fig. 4d,e and Extended Data Fig. 10f–i). This was accompanied by a significant increase in the similarity between activity patterns occurring after error trials and those after correct trials (by contrast, no such increase occurred in controls) (Fig. 4d,e,h,i). To confirm that this reflects a suppression of changes in activity patterns that normally occur after rule-shift errors, we also computed the similarity between activity patterns after rule-shift errors and those occurring after errors during learning of the initial association (Fig. 4g,k–m and Extended Data Figs. 3e,f and 10t,u). Activity patterns following rule-shift errors also became more similar to those observed during the initial association. Finally, analysing activity on faster timescales (1 s), we found that inhibiting callosal PV$^+$ projections also makes fast-timescale patterns occurring after incorrect decisions during the rule shift more similar to

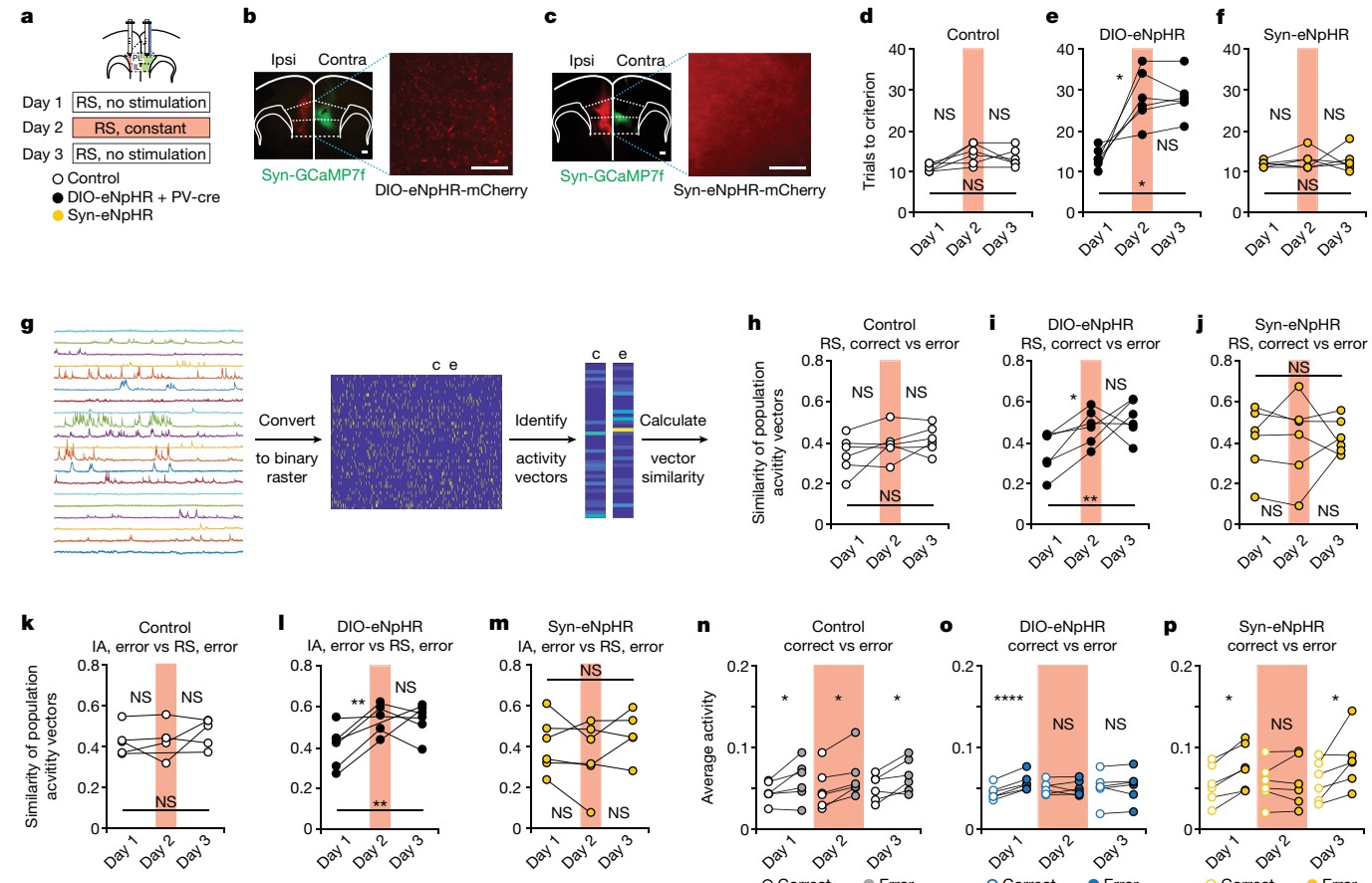

**Fig. 4 | Optogenetic inhibition of callosal PV projections disrupts prefrontal activity patterns. a**, Mice had AAV-DIO-eNpHR-mCherry (in PV-cre mice) or AAV-Synapsin-eNpHR-mCherry (in wild-type mice) or control viruses AAV-DIO-mCherry (PV-cre mice) or AAV-Syn-mCherry (wild-type mice) injected in the ipsilateral mPFC, AAV-synapsin-GCaMP7f injected in the contralateral mPFC, and a GRIN lens (for microendoscope imaging) implanted in the contralateral mPFC. Experimental design: day 1, no light; day 2, continuous light for inhibition during the rule shift; day 3, no light. **b**, Left, representative images showing DIO-eNpHR in ipsilateral mPFC and Syn-GCaMP7f in contralateral mPFC. Right, DIO-eNpHR in callosal PV⁺ axonal fibres. **c**, Left, representative images of Syn-eNpHR in ipsilateral mPFC and Syn-GCaMP7f in contralateral mPFC. Right, Syn-eNpHR in callosal axonal fibres. **d,f**, Performance of controls and Syn-eNpHR mice (*n* = 6 per group) did not change across days. **e**, Performance was impaired in DIO-eNpHR mice (*n* = 6) on day 2 and day 3. **g**, Overview of Ca²⁺ imaging analysis and example data from one experiment. d*F/F* traces for a subset of neurons and 10 min. We detected events in d*F/F* traces to generate a binary activity raster, identify periods immediately following correct choices (c) or errors (e), compute population activity vectors, and calculate the similarity between vectors. **h,j**, For controls and Syn-eNpHR mice (*n* = 6 per group), the similarity of activity vectors following rule-shift errors and those occurring after correct decisions during the rule shift

did not change across days. **i**, In DIO-eNpHR mice (*n* = 6 mice), the similarity between activity vectors following rule-shift errors and those following rule-shift correct trials was higher on days 2 and day 3 than on day 1. **k,m**, In controls and Syn-eNpHR mice (*n* = 6 per group) the similarity of activity vectors following initial association errors and those following rule-shift errors did not change across days. **l**, In DIO-eNpHR mice (*n* = 6), the similarity of activity vectors following initial association errors and those following rule-shift errors was higher on day 2 and day 3 than on day 1. **n**, Controls (*n* = 6) had higher average activity (fraction of active frames, averaged across neurons) during the 10 s following rule-shift errors compared with the 10 s following rule-shift correct decisions on all days. Two-way ANOVA with Bonferroni post hoc comparisons. **o**, In DIO-eNpHR mice (*n* = 6), the increase in average activity after rule-shift errors depends on the day. Average activity increased after rule-shift errors on day 1, but not on days 2 or day 3. Two-way ANOVA with Bonferroni post hoc comparisons. **p**, In Syn-eNpHR mice (*n* = 6), there is an overall increase in average activity following rule-shift errors. This occured on day 1 and day 3, but not on day 2. Two-way ANOVA with Bonferroni post hoc comparisons. Full statistics are provided in the Methods. Two-way ANOVA followed by Tukey's post hoc comparisons unless otherwise noted. **P* < 0.05, ***P* < 0.01, ***P* < 0.0001. **b,c**, Scale bars: 100 μm (left) and 50 μm (right).

those occurring after correct decisions during the rule shift (Extended Data Fig. 10v–x). Furthermore, inhibiting callosal PV⁺ projections also disrupts the evolution of fast-timescale activity patterns that normally occurs over the course of a single error trial (Extended Data Fig. 10q–s). This suggests that inhibiting callosal PV⁺ projections disrupts learning that occurs, at least in part, during each trial. Consistent with this, inhibiting callosal PV+ projections only during ITIs did not affect rule-shift learning (Extended Data Fig. 2l–o).

As in previous experiments, deficits in learning a rule shift observed after inhibiting callosal PV⁺ projections persisted on the next day (day 3) without any additional light delivery to activate eNpHR (Fig. 4e).

The increased similarity between activity patterns following rule-shift errors and those following correct trials and between rule-shift errors and initial association errors also persisted at this later time point (Fig. 4i,l). By contrast, there was no change in rule-shift performance nor in the similarity of activity patterns after correct versus error rule-shift trials across days in eNpHR-negative controls (Fig. 4d,h,k).

Consistent with our earlier findings, nonspecifically inhibiting all callosal projections (using Syn-eNpHR) did not disrupt rule-shift learning (Fig. 4f). Nonspecifically inhibiting all callosal projections also had no effect on the similarity of activity after rule-shift errors to activity after rule-shift correct trials (Fig. 4j) or initial association errors (Fig. 4m).

Thus, callosal inputs can inappropriately maintain previously established neural representations, specifically when callosal PV$^+$ projections and gamma synchrony are suppressed. Furthermore, callosal PV$^+$ projections and interhemispheric PV gamma synchrony seem to be specifically involved in regulating callosal communication—when all callosal communication is suppressed, there is no longer a behavioural consequence of inhibiting callosal PV$^+$ projections and suppressing gamma synchrony.

## Discussion

Changes in patterns of activity within the prefrontal cortex are presumed to drive behavioural adaptation. However, the neural mechanisms that switch the mPFC from maintaining previously learned representations and strategies to updating them have not been known. Previous work had suggested the involvement of PV neurons and long-distance gamma synchrony. Here we characterize a novel callosal projection from prefrontal PV neurons, and reveal the multifaceted mechanism whereby this specific synapse modulates particular targets (that is, MD-projecting pyramidal neurons) and network dynamics (gamma oscillations) to regulate circuit interactions (callosal communication) that shape emergent patterns of activity (population activity vectors following error trials) and guide behaviour. Three aspects of this mechanism are particularly notable.

First, the fact that neocortical GABAergic neurons give rise to long-range connections capable of eliciting physiologically meaningful postsynaptic responses has only recently become appreciated, and the behavioural functions of these long-range GABAergic projections are not well understood. Here we show that in mPFC, callosal GABAergic projections from PV$^+$ neurons promote interhemispheric gamma synchrony, which has previously been shown to be necessary for learning rule shifts. This directly supports a hypothesized role of long-range GABAergic projections in synchronizing rhythmic activity interhemispherically[24], and contrasts with earlier findings from hippocampal slices which suggested that excitatory synapses on PV interneurons synchronize gamma oscillations across long distances[26]. One limitation is that we cannot directly infer the number of PV interneurons driving these increases in synchrony, because they are based on bulk measurements of fluorescence. However, these measures of synchrony increase by around 50%, making it unlikely that they are driven solely by a small fraction of cells. Whereas many long-range GABAergic projections target mainly downstream GABAergic neurons[21], callosal PV$^+$ projections innervate a large fraction of layer 5 and layer 6 pyramidal neurons. Furthermore, callosal PV$^+$ synapses have the same subtype-specificity previously observed for local PV interneuron synapses[27], preferentially inhibiting thalamically projecting rather than callosally projecting pyramidal neurons. This specificity is particularly notable given that we previously found that this mPFC output pathway to MD thalamus is essential for behavioural flexibility[28].

Second, in this task, gamma synchrony does not appear to act via the most basic form of the 'communication through coherence' mechanism[29]. The communication through coherence mechanism proposes that when two regions are synchronized in a coherent manner, they benefit from enhanced effective connectivity. However, in this case, connectivity between the two hemispheres is not actually necessary for rule-shift learning, as activity patterns evolve appropriately and learning is unperturbed when we nonspecifically inhibit all callosal projections. By contrast, when PV$^+$ callosal projections are selectively inhibited, sparing excitatory callosal communication, mice become perseverative and previously learned representations are inappropriately maintained. In this scenario, excitatory communication between the two hemispheres is intact, but gamma synchrony is lost (Fig. 3g). We previously showed that optogenetically perturbing gamma synchrony (using out-of-phase stimulation across hemispheres) disrupts rule-shift learning[8]. Thus, inter-regional communication is not necessary for normal behaviour; however, when gamma synchrony is lost, inter-regional communication occurs in an aberrant manner that interferes with normal learning. This could occur because, during a gamma cycle, high levels of inhibition gradually fall, producing corresponding changes in excitatory neuron firing[8]. In this way, rhythmic inhibition periodically resets network activity; this reset could allow the emergence of new representations that diverge from the previously established ones that are stored within patterns of recurrent connections. By contrast, when excitatory input from the contralateral cortex is not appropriately synchronized by PV-driven inhibition, it could arrive prematurely, inappropriately reinforcing previously learned representations and thereby disrupting normal learning. Appropriately timed callosal inhibition might also counterbalance the influence of callosal and recurrent excitation on a specific neuronal population (for example, mPFC–MD projection neurons) that otherwise maintains previously learned associations.

A third surprising feature is that inhibiting callosal PV$^+$ projections leads to persistent changes in gamma synchrony, neural representations and rule-shift performance. Furthermore, stimulating the same projections can then reverse this behavioural impairment. This reveals a novel form of network plasticity, whereby transiently disrupting connectivity between two brain regions leads to persistent deficits in their ability to synchronize, process information and contribute to behaviour. Notably, the resulting network state, characterized by cognitive inflexibility and deficient task-evoked gamma synchrony, captures key cognitive and circuit endophenotypes of schizophrenia. In many respects, this deleterious network plasticity represents the inverse of our earlier finding that transiently enhancing gamma synchrony (using synchronizing optogenetic stimulation) can persistently rescue rule-shift learning in mutant (heterozygous for *Dlx5* and *Dlx6*) mice[6,8]. Our new findings show that prefrontal circuit plasticity resulting from transient modulations of gamma synchrony is bi-directional, occurs even in previously normal mice, and can be triggered by a previously unappreciated long-range GABAergic projection. It will be important to explore how this plasticity depends on changes in callosal PV$^+$ projections, callosally projecting PV$^+$ neurons, and/or additional circuit elements. Future studies could also explore the role of callosal PV$^+$ projections (and potential plasticity) using behavioural paradigms that differ from those used here in key respects (for example, automated lever-press tasks).

In summary, our results show how a novel connection switches the prefrontal cortex from maintaining to updating behavioural strategies by gating the ability of callosal communication to maintain previously established activity patterns. Furthermore, this connection can trigger a novel bi-directional form of network plasticity. Thus, callosal connections originating from prefrontal PV neurons represent a critical circuit locus for understanding and potentially correcting deficits in gamma synchrony and behavioural flexibility that are major features of schizophrenia[9,10].

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

# Methods

## Mice

All animal care, procedures and experiments were conducted in accordance with NIH guidelines and approved by the Administrative Panels on Laboratory Animal Care at the University of California, San Francisco as well as followed French and European guidelines for animal experimentation and in compliance with the institutional animal welfare guidelines of the Paris Brain Institute. Mice were group housed (2–5 siblings) in a temperature-controlled environment (22–24 °C), had ad libitum access to food and water, and reared in normal lighting conditions (12-h light-dark cycle), until rule-shift experiments began. All experiments were done using PV-cre, wild-type C57/Bl6, and PV-cre Ai14 lines (The Jackson Laboratory). Both male and female adult mice (8–10 weeks and 10–20 weeks old at time of experiment) were used for slice electrophysiology and behavioural experiments, respectively.

## Surgery

Male and female mice were anaesthetized using isoflurane (2.5% induction, 1.2–1.5% maintenance, in 95% oxygen) and placed in a stereotaxic frame (David Kopf Instruments). Body temperature was maintained using a heating pad. An incision was made to expose the skull for stereotaxic alignment using bregma and lambda as vertical references. The scalp and periosteum were removed from the dorsal surface of the skull and scored with a scalpel to improve implant adhesion. Viruses were infused at 100–150 nl min$^{-1}$ through a 35-gauge, beveled injection needle (World Precision Instruments) using a microsyringe pump (World Precision Instruments, UMP3 UltraMicroPump). After infusion, the needle was kept at the injection site for 5–10 min and then slowly withdrawn. After surgery, mice were allowed to recover until ambulatory on a heated pad, then returned to their home cage.

For slice electrophysiology experiments using optogenetic opsins, mice were injected unilaterally in the mPFC, near the border between the prelimbic and infralimbic cortices (1.7 anterior-posterior (AP), + or −0.3 mediolateral (ML), and −2.75 dorsoventral (DV) mm relative to bregma) with 0.4 µl of AAV5-EF1α-DIO-ChR2-eYFP (UNC Vector Core) to selectively target neurons expressing Cre. To allow for virus expression, experiments began at least 3 weeks after injection. For retrograde labelling of mPFC neurons that project callosally versus to mediodorsal (MD) thalamus, cholera toxin subunit B (CTb) conjugated with Alexa Fluor 488 (CTb-488, Invitrogen; 0.2% w/v, 400 nl) was injected in the PFC ipsilateral to the previous AAV-DIO-ChR2 injection (1.7 AP, −0.4 ML, and −2.5 DV mm relative to bregma) and Alexa Fluor 594-conjugated CTb (CTb-594, Invitrogen; 0.2% w/v, 400 nl) was injected in the MD thalamus (contralateral to the AAV-DIO-ChR2 injection) (−1.7 AP, +0.35 ML, −3.5 DV mm relative to bregma). For retrograde labelling of mPFC neurons projecting to the NAc versus dorsal striatum, CTb-488 (0.2% w/v, 400 nl) was injected in the NAc (contralateral to the AAV-DIO-ChR2 injection) (1.34 AP, −1.0 ML, and −4.6 DV mm relative to bregma) and CTb-594 (0.2% w/v, 600 nl) was injected in the dorsal striatum (also contralateral to the AAV-DIO-ChR2 injection) (+0.9 AP, +1.16 ML, −3.0 DV mm relative to bregma). To allow time for retrograde transport of CTb, experiments began one week after CTb injection.

For behavioural experiments using eNpHR and control optogenetics, mice were injected unilaterally in the mPFC, near the border between the prelimbic and infralimbic cortices (1.7 AP, + or −0.3 ML, and −2.75 DV mm relative to bregma) with 1 µl of AAV2-EF1α-DIO-eNpHR3.0-mCherry (UNC Vector Core), 1 µl of AAV5-EF1α-DIO-eYFP (UNC Vector Core), AAV5-hSynapsin-eNpHR3.0-eYFP (UNC Vector Core) or AAV5-hSynapsin-tdTomato (UNC Vector Core), to selectively target Cre-expressing cells or non-selectively target prefrontal neurons. After injecting virus, a 200/240 µm (core/outer) diameter, NA = 0.22, mono fibre-optic cannula (Doric Lenses, MFC_200/240-0.22_2.3mm_FLT) was slowly inserted into mPFC until the tip of the fibre reached a DV depth of −2.25. Implants were affixed onto the skull using Metabond Quick Adhesive Cement

(Parkell). To allow for virus expression, behavioural experiments began at least five weeks after injection.

For behavioural experiments using both NpHR and ChR2 optogenetics, mice were injected unilaterally in the mPFC, near the border between the prelimbic and infralimbic cortices (1.7 AP, −0.3 ML, and −2.75 DV mm relative to bregma) with 0.75 µl of AAV8-nEF-Con/Fon-NpHR3.3-eYFP (Addgene) and 0.75 µl of AAV8-nEF-Con/Fon-ChR2-mCherry (Addgene) or 0.75 µl of AAV8-EF1α-Con/Fon-mCherry (Addgene) and contralaterally in PFC (1.7 AP, +0.3 ML, −2.75 DV mm relative to bregma) with 0.7 µl AAVrg-EF1α-FlpO (Addgene), to selectively target Cre-expressing cells in ipsilateral PFC that send projections to contralateral PFC. After injecting virus, a 200/240 µm (core/outer) diameter, NA = 0.22, mono fibre-optic cannula (Doric Lenses, MFC_200/240-0.22_2.3mm_FLT) was slowly inserted into mPFC until the tip of the fibre reached a DV depth of −2.25. Implants were affixed onto the skull using Metabond Quick Adhesive Cement (Parkell). To allow for virus expression, behavioural experiments began at least five weeks after injection.

For behavioural experiments that combined dual-site voltage indicator imaging with optogenetics, mice were injected bilaterally at 3 depths (DV: −2.5, −2.25 and −2.0) at the following AP/ML for mPFC: 1.7 AP, ±0.3 ML with 3 × 0.2 µl of AAV1-CAG-DIO-Ace2N-4AA-mNeon (Virovek). Mice were also injected unilaterally in the mPFC (1.7 AP, −0.3 ML, and −2.75 DV mm relative to bregma) with either 1 µl of AAV2-EF1α-DIO-eNpHR3.0-BFP (Virovek) or 1 µl of AAV5-EF1α-DIO-mCherry (UNC Vector Core), 1 µl of AAV2-hSynapsin-eNpHR3.0-mCherry (UNC Vector Core) or 1 µl of AAV2-hSynapsin-mCherry (UNC Vector Core). After injection of virus, two 400/430 µm (core/outer) diameter, NA = 0.48, multimode fibre implants (Doric Lenses, MFC_400/430-0.48_2.8mm_ZF1.25_FLT) were slowly inserted into the mPFC at a ±12° angle using the following coordinates: 1.7 (AP), ±0.76 (ML), −2.14 (DV). To allow for virus expression, behavioural experiments began at least five weeks after injection.

For behavioural experiments that combined in vivo calcium imaging with optogenetics, mice were injected unilaterally at 4 depths (DV: −2.75, −2.5, −2.25, −2.0) at the following AP/ML for mPFC: 1.7 AP, +0.4 ML with diluted (1:3; Addgene) 4 × 0.15 µl of AAV9-synapsin-jGCaMP7f-WPRE (Addgene). Mice were also injected unilaterally in the contralateral mPFC (1.7 AP, −0.3 ML, and −2.75 DV mm relative to bregma) with either 1 µl of AAV2-EF1α-DIO-eNpHR3.0-mCherry (UNC Virus Core), 1 µl of AAV5-EF1α-DIO-mCherry (UNC Virus Core), 1 µl of AAV2-hSynapsin-eNpHR3.0-mCherry (UNC Virus Core) or 1 µl of AAV2-hSynapsin-mCherry (UNC Virus Core). After injection of virus, a 0.5 mm × 4.0 mm-long integrated GRIN lens (Inscopix) was slowly advanced into the mPFC until the tip was placed at 1.7 AP, +0.4 ML, DV −2.25 and cemented in place with Metabond Quick Adhesive Cement (Parkell). To allow for virus expression, behavioural experiments began at least five weeks after injection.

## Slice preparation and analysis

Adult mice were anaesthetized using isoflurane and decapitated, and their brains were rapidly removed. Coronal slices (250 µm thick) were cut from adult mice of either sex using a vibratome (VT1200S Leica Microsystems) and a chilled slicing solution in which Na$^+$ was replaced by sucrose, then incubated in warmed artificial cerebrospinal fluid (ACSF) at 30-31 °C for 15 min and then at least 1 h at room temperature before being used for recordings. ACSF contained (in mM): 126 NaCl, 26 NaHCO$_3$, 2.5 KCl, 1.25 NaH$_2$PO$_4$, 1 MgCl$_2$, 2 CaCl and 10 glucose. Slices were secured by placing a harp along the midline between the two hemispheres.

Somatic whole-cell patch recordings were obtained from ChR2-negative neurons amid ChR2-positive axonal fibres in the mPFC contralateral to the site of virus injection, on an upright microscope (BX51WI; Olympus). Recordings were made using a Multiclamp 700A (Molecular Devices). Patch electrodes (tip resistance = 2–6 MOhms) were filled with the following (in mM): 130 potassium gluconate, 10 KCl, 10 HEPES, 10 EGTA, 2 MgCl$_2$, 2 MgATP, and 0.3 NaGTP (pH adjusted to

7.3 with KOH). All recordings were at 32.5 ± 1 °C. Series resistance was usually 10–20 MΩ, and experiments were discontinued above 30 MΩ.

Intrinsic properties were calculated based on the current-clamp responses to a series of 250 msec current pulse injections from −200 to 450 pA (50 pA per increment). Spiking properties were calculated based on the response to a current pulse that was 100 pA above the minimal level that elicited spiking. Neurons were classified as fast spiking if they met 3 out of the 4 following criteria: action potential (AP) half-width was < 0.5 ms, firing frequency > 50 Hz, fast after hyperpolarization (fAHP) amplitude > 14 mV, and spike-frequency accommodation (SFA) index < 2.

## In vitro ChR2 stimulation

Stimulation of channelrhodopsin (ChR2) in callosal PV terminals was performed using ~4–5 mW flashes of light generated by a Lambda DG-4 high-speed optical switch with a 300 W Xenon lamp (Sutter Instruments) or by an LED (Cairn Research OptoLED Lite), and an excitation filter set centred around 470 nm, delivered to the slice through a 40× objective (Olympus). Illumination was delivered across a full high-power (40×) field. To measure IPSPs, current-clamp recordings were performed while stimulating ChR2 using trains of light pulses (5 ms light pulses at 5 Hz, 20 Hz, and 40 Hz). In experiments in which glutamatergic and GABA$_A$ receptors were blocked, drugs were bath applied at the following concentrations (in µM): 10 CNQX (Tocris) or 20 DNQX (Tocris), 50 APV (Tocris), and 10 gabazine (Sigma). Drugs were prepared as concentrated stock solutions and were diluted in ACSF on the day of the experiment.

## Rule-shift task

This cognitive flexibility task has been described previously[6]. In brief, mice are singly housed and habituated to a reverse light/dark cycle, and food intake is restricted until the mouse is 80–85% of the ad libitum feeding weight. After mice reached their target weight, they underwent one day of habituation. On this day, mice were given ten consecutive trials with the baited food bowl to ascertain that they could reliably dig and that only one bowl contained food reward. All mice were able to dig for the reward, and started the task the next day. At the start of each trial, the mouse was placed in its home cage to explore two bowls, each containing one odour and one digging medium, until it dug in one bowl, signifying a choice. As soon as a mouse began to dig in the incorrect bowl, the other bowl was removed, so there was no opportunity for 'bowl switching' (digging is defined as the sustained displacement of the medium within a bowl). The bait was a piece of a peanut butter chip (approximately 5–10 mg in weight) and the cues, either olfactory (odour) or somatosensory and visual (texture of the digging medium which hides the bait), were altered and counterbalanced. All cues were presented in small animal food bowls (All Living Things Nibble bowls, PetSmart) that were identical in colour and size. Digging media were mixed with the odour (0.01% by volume) and peanut butter chip powder (0.1% by volume). All odours were ground dried spices (McCormick or Alpi Nature garlic and McCormick or Albert Ménès coriander), and unscented digging medium (Mosser Lee White Sand Soil Cover or Scalare Sable de Rivière, Natural Integrity Clumping Clay or Monoprix cat litter).

After mice reached their target weight, they underwent one day of habituation. On this day, mice were given ten consecutive trials with the baited food bowl to ascertain that they could reliably dig and that only one bowl contained food reward. Specifically, the habituation trials are used to train the mouse on the mechanics of the task, there is no association made between food reward and cue. All mice were able to dig for the reward. Mice do not undergo any other specific training before being tested on the task. Then, on days 1 and 2 (and in some cases, on additional days as well), mice performed the task (this was the testing done for experiments). After the task was done for the day, the bowls were filled with different odour–medium combinations

and food was evenly distributed among these bowls and given to the mouse so that no specific cue was rewarded greater than the other cues present. The same stimuli were used across days—only the cue that is associated with the food reward changed.

Mice were tested through a series of trials. The determination of which odour and medium to pair and which side (left or right) contained the baited bowl was randomized (subject to the requirement that the same combination of pairing and side did not repeat on more than three consecutive trials) using https://www.random.org. On each trial, while the particular odour–medium combination present in each of the two bowls may have changed, the particular stimulus (for example, a particular odour or medium) that signalled the presence of food reward remained constant over each portion of the task (initial association and rule shift). If the initial association paired a specific odour with food reward, then the digging medium would be considered the irrelevant dimension. The mouse is considered to have learned the initial association between stimulus and reward if it makes eight correct choices during ten consecutive trials. Each portion of the task ended when the mouse met this criterion. Following the initial association, the rule-shift portion of the task began, and the particular stimulus associated with reward underwent an extra-dimensional shift. For example, if an odour had been associated with reward during the initial association, then a digging medium was associated with reward during the rule-shift portion of the task. The mouse is considered to have learned this extra-dimensional rule shift if it makes 8 correct choices during ten consecutive trials. When a mouse makes a correct choice on a trial, it is allowed to consume the food reward before the next trial. Following correct trials, the mouse is transferred from the home cage to a holding cage for about 10 s while the new bowls were set up (ITI). After making an error on a trial, a mouse was transferred to the holding cage for about 2 min (ITI). For Extended Data Fig. 4p–z, the ITI following errors was 30 s in the holding cage. All animals performed the initial association in a similar number of trials (average: 10–15 trials). Experiments were performed blind to the virus injected. Videos were manually scored with a temporal resolution of 1 s.

For analyses (described below), the onset of digging was chosen as the time of a decision for two reasons. First, as noted above, once a mouse began to dig in the incorrect bowl, the other (correct) bowl was removed. Second, only upon the commencement of digging could a mouse determine whether reward was present in the chosen bowl and obtain feedback about whether or not it had made a correct choice. The time windows used for analysis excluded periods when the mouse moved from the home cage to the holding cage and vice versa.

## Rule-reversal task

This cognitive flexibility task was described previously[6,8]. Similarly to the mechanics of the rule-shift task described above, after the initial association, the rule-reversal portion of the task began, and the particular stimulus associated with reward underwent an intra-dimensional reversal. For example, if an odour had been associated with reward during the initial association, then the previously unrewarded odour became associated with reward during the rule-reversal portion of the task. The mouse was considered to have learned the intra-dimensional rule reversal when it made eight correct choices out of ten consecutive trials.

## In vivo optogenetic stimulation

For behavioural experiments using optogenetic eNpHR stimulation: A 532 nm green laser (OEM Laser Systems) was coupled to the mono fibre-optic cannula (Doric Lenses) with a zirconia sleeve (Doric Lenses) through a 200-µm-diameter mono fibre-optic patch cord (Doric Lenses) and adjusted such that the final light power was 2.5 mW. For behavioural experiments using optogenetic ChR2 stimulation: A 473 nm blue laser (OEM Laser Systems) was coupled to the mono fibre-optic cannula (Doric Lenses) with a zirconia sleeve (Doric Lenses) through

a 200 μm diameter mono fibre-optic patch cord (Doric Lenses, Inc.) and adjusted such that the final light power was 0.5 mW. A function generator (Agilent 33500B Series Waveform Generator) connected to the laser generated a 40-Hz train of 5-ms pulses.

Extended Data Fig. 2 shows experiments designed to control for potential behavioural effects of scattered light from one hemisphere activating eNpHR in PV neuron cell bodies in the contralateral hemisphere. These experiments used a final light power of 0.1 mW when connected to the 532 nm green laser (OEM Laser Systems) or 638 nm red laser (Doric Lenses). To determine the appropriate light power for these experiments, a mouse was implanted with a dual fibre-optic cannula (Doric Lenses; DFC_200/240-0.22_2.3mm_GS0.7_FLT) without virus injection in order to measure light scattering from one mPFC to the contralateral hemisphere. Using a dual fibre-optic patch cord (Doric Lenses; DFP_200/240/900-0.22_2m_GS0.7-2FC), light was delivered to the mPFC on one side, and the light coming through the other side was measured using a light meter (ThorLabs, PM100D). The final light power delivered to one mPFC was 2.5 mW, across wavelengths 532 nm, 594 nm, and 638 nm—similar to what was used in the optogenetic and optogenetic and dual-site voltage indicator experiments. Measurements of 40 nW, 20 nW, and 50 nW at the contralateral fibre were observed, respectively. Accounting for transmission loss of the patch cord (for example, only 80% of the light is transmitted from end to end), the scattered light power entering the fibre tip would be 50 nW, 25 nW and 62.5 nW respectively. This measurement likely overestimates the actual light entering via the fibre tip located in brain parenchyma since it will include both scattered light within the brain and contamination from ambient room light. Conversely, this measurement only includes light located in the vicinity of the fibre tip that is traveling at an appropriate angle to enter the fibre (which has numerical aperture 0.22 implying a 25.4° acceptance angle). Therefore, to be extremely conservative, experiments in Extended Data Fig. 2 utilized a final light power of 0.1 mW, which is >1,000 times stronger than the measured scattered light power. Both 532 nm and 638 nm at this final light power ipsilateral to the viral injection site was used.

Experiments in Extended Data Fig. 2l–o were similar to other experiments using optogenetic eNpHR stimulation except in the following respects: mice were injected with 0.7 μl of AAV2-EF1α-DIO-eNpHR3.0-mCherry (UNC Vector Core) and 0.4 μl of AAV5-EF1α-DIO-ChR2-eYFP (UNC Vector Core), to selectively target Cre-expressing cells (although ChR2 stimulation was not ultimately used in these experiments). Light for optogenetic eNpHR stimulation was specifically switched on on day 2 during ITIs (when the mouse was in the holding cage) and switched off during the trial in the home cage.

For all optogenetic experiments, light stimulation began once mice reached the 80% criterion during the initial association portion of the task. Mice then performed three additional initial association trials with the light stimulation before the rule-shift portion of the task began. The light stimulation did not alter the performance or behaviour of the mice during these three extra trials of the initial association. Experiments were performed blind to virus injected.

### Combined dual-site voltage indicator imaging and optogenetic eNpHR stimulation
High-bandwidth, time-varying bulk fluorescence signals were measured at each recording site using the dual-site voltage-indicator technique[8], with some modifications as described below.

### Optical apparatus
A fibre-optic stub (400 μm core, NA = 0.48, low-autofluorescence fibre; Doric Lenses, MFC_400/430-0.48_2.8mm_ZF1.25_FLT) was stereotaxically implanted in each targeted brain region. A matching fibre-optic patch cord (Doric Lenses, MFP_400/430/1100-0.48_2m_FC-ZF1.25) provided a light path between the animal and a miniature, permanently aligned optical bench, or 'mini-cube'

(Doric Lenses, FMC5_E1(460-490)_F1(500-540)_E2(555-570)_F2(580-600)_S). One fibre on the ipsilateral side of the viral injection of either 1 μl of AAV2-EF1α-DIO-eNpHR3.0-BFP (Virovek) or 1 μl of AAV5-EF1α-DIO-mCherry (UNC Vector Core) was used to both deliver excitation light to and collect emitted fluorescence from that recording site. The fibre contralateral to the viral injection site was connected to a separate mini-cube (Doric Lenses, FMC6_E1(460-490)_F1(500-540)_E2(555-570)_F2(580-600)_O(628-642)_S) that was attached to a 638 nm laser (Doric Lenses) and was used to deliver excitation light and optogenetic stimulation to and collect emitted fluorescence from that recording site. The far end of the patch cord and each 1.25 mm diameter zirconia optical implant ferrule were cleaned with isopropanol before each recording, then securely attached via a zirconia sleeve.

For the first mini-cube sans laser port, optics allow for the simultaneous monitoring of two spectrally separated fluorophores, with dichroic mirrors and cleanup filters chosen to match the excitation and emission spectra of the voltage sensor and reference fluorophores in use ('mNeon' voltage sensor channel: excitation 460–490 nm, emission 500–540 nm; 'red' control fluorophore: excitation 555–570 nm, emission 580–600 nm). The mini-cube optics are sealed and permanently aligned and all five ports (sample to animal, two excitation lines and two emission lines) are provided with matched coupling optics and FC connectors to allow for a modular system design.

For the second mini-cube that contains a port for optogenetic stimulation, a 565 nm LED and 555–570 nm filter is used to excite tdTomato, a 580–600 nm filter is used for tdTomato emission, and a 638 nm laser with 628–642 nm filter is used to excite eNpHR. The light used to excite eNpHR does not interfere with the genetically encoded voltage indicator based measurements of synchrony, and the light used to excite tdTomato does not activate eNpHR enough to affect rule-shift performance. This is due to two factors. First, the eNpHR and tdTomato excitation spectra are offset—for example, at 605 nm, eNpHR activation is near its peak whereas relative excitation of tdTomato is <1%. Second, the intensity used to excite tdTomato is just ~0.1 mW, which is ~50-fold less than what was used to activate eNpHR and disrupt rule-shift performance. The mini-cube optics are sealed and permanently aligned and all six ports (sample to animal, three excitation lines and two emission lines) are provided with matched coupling optics and FC connectors to allow for a modular system design.

To perform dual-site voltage indicator recordings, excitation light for each of the two colour channels was provided by a fibre-coupled LED (Center wavelengths 490 nm and 565 nm, Thorlabs M490F3 and M565F3) connected to the mini-cube by a patch cord (200 μm, NA = 0.39; Thorlabs M75L01). Using a smaller diameter for this patch cord than for the patch cord from the cube to the animal is critical to reduce the excitation spot size on the output fibre face and thus avoid cladding autofluorescence. LEDs were controlled by a 4-channel, 10-kHz-bandwidth current source (Thorlabs DC4104). LED current was adjusted to give a final light power at the animal (averaged during modulation, see below) of approximately 200 μW for the mNeon channel (460–490 nm excitation), and 100 μW for the Red channel (555–570 nm excitation).

Each of the two emission ports on the mini-cube was connected to an adjustable-gain photoreceiver (Femto, OE-200-Si-FC; Bandwidth set to 7 kHz, AC-coupled, 'low'gain of ~5 × 10⁷ V W⁻¹) using a large-core high-NA fibre to maximize throughput (600 μm core, NA = 0.48 (Doric lenses, MFP_600/630/LWMJ-0.48_0.5m_FC-FC).

Note that, for dual-site voltage indicator recordings and optogenetics experiments, two completely independent optical setups were employed, with separate implants, patch cords, mini-cubes, LEDs, a separate laser, photoreceivers, and lock-in amplifiers.

### Modulation and lock-in detection
At each recording site, each of the two LEDs was sinusoidally modulated at a distinct carrier frequency to reduce crosstalk due to

overlap in fluorophore spectra. The corresponding photoreceiver outputs were then demodulated using lock-in amplification techniques. A single instrument (Stanford Research Systems, SR860) was used to generate the modulation waveform for each LED and to demodulate the photoreceiver output at the carrier frequency. To further reduce crosstalk between recording sites, distinct carrier frequencies (2, 2.5, 3.5 and 4 kHz) were used across sites. Low-pass filters on the lock-in amplifiers were selected to reject noise above the frequencies under study (cascade of 4 Gaussian FIR filters with 84 Hz equivalent noise bandwidth; final attenuation of signals are approximately −1dB (89% of original magnitude) at 20 Hz, −3dB (71% of original magnitude) at 40 Hz, and −6dB (50% of original magnitude) at 60 Hz).

### In vivo dual-site voltage indicator imaging

Analogue signals were digitized by a multichannel real-time signal processor (Tucker-Davis Technologies; RX-8). The commercial software Synapse (Tucker-Davis Technologies) running on a PC was used to control the signal processor, write data streams to disk, and to record synchronized video from a generic infrared USB webcam (Ailipu Technology, ELP-USB100W05MT-DL36). Lock-in amplifier outputs were digitized at 3 kHz.

### Combined dual-site voltage indicator imaging and optogenetics analysis

Analysis of voltage indicator data was described previously[8] and was facilitated using the signal processing toolbox and MATLAB (Mathworks), using the following functions: fir1, filtfilt, and regress. All four signals during the entire time series of the experiment (left mNeon, left tdTomato, right mNeon, right tdTomato) were first filtered around a frequency of interest. To quantify zero-phase lag cross-hemispheric synchronization between left and right mNeon signals, a linear regression analysis was performed to predict the right mNeon signal using the following inputs: left mNeon signal, left tdTomato signal, and right tdTomato signal. The goodness of fit is compared to how well the regression works if the left mNeon signal is shuffled, i.e., if a randomly chosen segment of the original left mNeon signal is used, instead of the segment recorded at the same time as the right mNeon signal. $R^2$ values are calculated as a function of time using one second segments and compared to the 99th percentile of the distribution of $R^2$ values obtained from 100 fits to randomly shuffled data. The fraction of time points at which the $R^2$ obtained from actual data exceeds the 99th percentile of the $R^2$ values obtained from shuffled data was used to measure zero-phase lag synchronization between the left and right mNeon signals.

This analysis was performed at the time of the decision (for example, immediately following the beginning of digging in one bowl, until the end of digging), and smoothed measurements over a 5-min time window following the time point of interest. The first five trials of the rule shift were analysed. Experiments were performed, scored and analysed blind to virus injected.

### Shorter-timeframe resolution dual-site voltage indicator imaging and optogenetics analysis

For high temporal resolution quantification of synchrony between signals from dual-site voltage indicator recordings, we first bandpass-filtered Ace-mNeon and tdTomato signals as described above. Then, we 'corrected' the filtered Ace-mNeon signal (to minimize artefacts and noise) by fitting the ipsilateral filtered tdTomato signal via robust linear regression using the robustfit function in Matlab and a time windows of 250 ms. Then, we subtracted off this fit of the tdTomato signal from the filtered Ace-mNeon signal to obtain a corrected Ace-mNeon signal. We then calculated zero-lag cross-correlation between the corrected Ace-mNeon signals from the left and right mPFC across the whole session using 1-s windows.

### Combined calcium imaging and optogenetics

Imaging data were collected using a miniaturized one-photon microscope (nVoke2; Inscopix). GCaMP7f signals (calcium activity) were detected using 435–460 nm excitation LED (0.1–0.2 mW), and optogenetic stimulation of eNpHR-expressing axons was performed using a second 590–650 nm excitation LED (1–2 mW light power). nVoke2 software (Inscopix) was used to control the microscope and collect imaging data. Images were acquired at 20 frames per second, spatially downsampled (4×), and were stored for offline data processing. An input TTL from a separate ANY-maze computer (Stoelting Europe) to the nVoke2 acquisition software were used to synchronize calcium imaging and mouse behaviour movies.

### Combined calcium-imaging and optogenetics analysis

Calcium-imaging movies were preprocessed using Inscopix Data Processing Software (IDPS; Inscopix). The video frames were spatially filtered (bandpass) with cut-offs set to 0.008 pixel$^{-1}$ (low) and 0.3 pixel$^{-1}$ (high) followed by frame-by-frame motion correction for removing movement artefacts associated with respiration and head-unrestrained behaviour. The mean image over the imaging session was computed, and the d$F/F$ was computed using this mean image. The resultant preprocessed movies were then exported into MATLAB, and cell segmentation was performed using an open-source calcium-imaging software (CIAPKG)[30]. Specifically, a principal component analysis/independent component analysis (PCA/ICA) approach was used to detect and extract regions of interest (presumed neurons) per field of view[31]. For each movie, the extracted output neurons were then manually sorted to remove overlapping neurons, neurons with low signal-to-noise ratio, and neurons with aberrant shapes. Accepted neurons and their calcium activity traces were exported to MATLAB for further analysis using custom scripts as previously described[32]. In brief, the s.d. ($\sigma$) of the calcium movie was calculated and this was used to perform threshold-based event detection on the traces by first detecting increases in d$F/F$ exceeding 2 (over 1 s). Subsequently, events were detected that exceeded 10 for over 2 s and had a total area under the curve higher than 150$\sigma$. The peak of the event was estimated as the local maximum of the entire event. For an extracted output neuron, active frames were marked as the period from the beginning of an event until the calcium signal decreased 30% from the peak of the event (up to a maximum of 2 s).

We calculated the similarity of population activity vectors using the 'cosine similarity', which is equivalent to computing the normalized dot product between the two vectors, that is:

$$\left( \sum x_i y_i \right) / \left[ \left( \sum x_i^2 \right) \left( \sum y_i^2 \right) \right]^{1/2}$$

where $x_i$ and $y_i$ represent the average activity of the $i$th neuron in the two population activity vectors. Each vector was the average of activity during the 10 s immediately following a choice. We computed the similarity between each pair of vectors, then averaged this similarity across all the pairs from one mouse.

The first five trials of the rule shift were analysed. For Extended Data Fig. 10t–u, the last five initial association trials and the additional initial association trials during optogenetic inhibition were analysed.

### Histology and imaging

All mice used for behavioural and imaging experiments were anaesthetized with Euthasol and transcardially perfused with 30 ml of ice-cold 0.01 M PBS followed by 30 ml of ice-cold 4% paraformaldehyde in PBS. Brains were extracted and stored in 4% paraformaldehyde for 24 h at 4 °C before being stored in PBS. Slices 70–100 μm thick were obtained on a Leica VT100S and mounted on slides. All imaging was performed on an Olympus MVX10, Nikon Eclipse 90i, Zeiss LSM510, Zeiss Axioskop2, Zeiss ApoTome.2, and Keyence BZ-X All-in-One Fluorescence

Microscope. All mice were verified to have virus-driven expression and optical fibres located in the mPFC. For mice used in parvalbumin immunohistochemistry, 60-μm slices from PV-cre mice injected with AAV2-EF1α-DIO-ChR2-eYFP unilaterally in PFC were obtained on a Leica VT100S and were rinsed twice at room temperature (10 min each) in PBS and incubated overnight at 4 °C with 0.3% Triton X-1000, 0.1% normal donkey serum (NDS) and monoclonal anti-PV antibody (1:1,000; Sigma). Slices were then rinsed twice in PBS (10 min each) at room temperature and incubated with Alexa 688 goat anti-mouse antibody (1:500; Invitrogen) for 3 h at room temperature. Slices were then rinsed twice in PBS (10 min each) at room temperature and coverslipped in mounting medium. Immunofluorescence was then observed with Zeiss ApoTome.2 and images were acquired.

### Data analyses and statistics

Statistical analyses were performed using Prism 8 (GraphPad) and detailed in the corresponding figure legends. Quantitative data are expressed as the mean and error bars represent the s.e.m. Group comparisons were made using two-way ANOVA followed by Bonferroni post hoc tests to control for multiple comparisons unless otherwise noted. Paired and unpaired two-tailed Student's $t$-tests were used to make single-variable comparisons. Similarity of variance between groups was confirmed by the $F$ test. Measurements were taken from distinct samples and from samples that were measured repeatedly. *$P < 0.05$, **$P < 0.01$, ***$P < 0.001$, ****$P < 0.0001$. Comparisons with no asterisk or 'NS' had $P > 0.05$ and were considered not significant. No statistical methods were used to pre-determine sample sizes but our sample size choice was based on previous studies[6,8] and are consistent with those generally employed in the field. Data distribution was assumed to be normal, but this was not formally tested.

For Fig. 3d,e,k,l, full statistics are: optogenetic inhibition of callosal mPFC PV terminals impairs rule shifts in DIO-eNpHR mice ($n = 5$) compared to DIO-mCherry controls ($n = 4$) on days 2 and 3 (two-way ANOVA (task day × virus); interaction: $P = 0.0009$). **d**, DIO-mCherry performance did not change (day 1 to 2: $P = 0.28$; day 1 to 3: $P = 0.094$; day 2 to 3: $P > 0.99$). **d,e,k,l**, Two-way ANOVA (task day × virus) comparing rule-shift (RS) performance across groups (interaction: $P < 0.0001$), showed no group difference on day 1 but significant impairment on days 2 and 3 for DIO-eNpHR compared to DIO-mCherry and Syn-tdTomato. **e**, Optogenetic inhibition of mPFC callosal PV terminals impairs rule shifts in DIO-eNpHR mice compared to DIO-mCherry controls (day 1 to 2: $P = 0.014$; day 1 to 3: $P = 0.012$; day 2 to 3: $P = 0.075$). For Fig. 3f,g, full statistics are: 30–50 Hz synchronization is higher after RS errors than after RS correct decisions across task days in controls ($n = 4$ mice; two-way ANOVA; main effect of trial type: $P = 0.0056$; day 1: $P = 0.004$; day 2: $P = 0.005$; day 3: $P = 0.022$). **g**, Gamma synchrony is higher after RS errors than after RS correct decisions for DIO-eNpHR mice on day 1 (no light), but not days 2 and 3 ($n = 5$ mice; two-way ANOVA (trial type × task day); interaction: $P = 0.0039$; day 1: $P = 0.0056$; day 2: $P = 0.16$; day 3: $P = 0.23$). Differences in gamma synchrony between RS errors and correct trials are also significantly lower in DIO-eNpHR mice compared to controls on days 2 (two-way ANOVA (trial type × virus); interaction: $P = 0.018$) and 3 (two-way ANOVA (trial type × virus); interaction: $P = 0.034$). We then performed two-way ANOVA (task day × virus, type of error) comparing gamma synchrony across groups with appropriate post hoc tests corrected for multiple comparisons; gamma synchrony is lower for DIO-eNpHR mice compared to matched controls on error (interaction: $P = 0.049$), but not correct trials (interaction: $P = 0.93$) on both days 2 ($P = 0.0089$) and 3 ($P = 0.016$). For Fig. 3k–n, full statistics are: inhibiting callosal terminals does not affect rule shifts in Syn-eNpHR mice ($n = 5$) compared to controls ($n = 4$) across days (two-way ANOVA (task day × virus); $P = 0.37$). **k**, Performance of controls did not change across days (day 1 to 2: $P = 0.48$; day 1 to 3: $P > 0.99$; day 2 to 3: $P > 0.99$). **l**, Performance of Syn-eNpHR mice did not change across days (day 1 to 2: $P > 0.99$; day 1 to 3: $P = 0.96$; day

2 to 3: $P = 0.85$). **m**, Interhemispheric 30–50 Hz synchronization is higher after RS errors than RS correct decisions across days in controls ($n = 4$ mice; two-way ANOVA; main effect of trial type: $P = 0.020$; day 1: $P = 0.015$; day 2: $P = 0.049$; day 3: $P = 0.025$). **n**, Gamma synchrony is higher after RS errors than RS correct decisions for Syn-eNpHR mice on day 1 (no light). This was abolished with light on day 2, then restored in the absence of light on day 3 ($n = 5$ mice; two-way ANOVA; (trial type × task day); interaction: $P = 0.011$; day 1: $P = 0.037$; day 2: $P = 0.46$; day 3: $P = 0.021$). Gamma synchrony is lower for Syn-eNpHR mice compared to matched controls on error (two-way ANOVA; interaction: $P = 0.045$) but not correct trials (interaction: $P = 0.74$) on day 2 ($P = 0.043$), but not on day 3 ($P = 0.58$). Two-way ANOVA followed by Bonferroni post hoc comparisons was used.

For Fig. 4d–f, full statistics are: optogenetic inhibition of callosal PV terminals impairs rule shifts in DIO-eNpHR mice ($n = 6$) compared to controls ($n = 6$) and Syn-eNpHR mice ($n = 6$) on days 2 and 3 (two-way ANOVA (task day × virus); interaction: $P < 0.0001$). **d**, Performance of controls did not change (day 1 to 2: $P = 0.057$; day 1 to 3: $P = 0.12$; day 2 to 3: $P = 0.67$). **e**, Optogenetic inhibition of callosal PV terminals impairs rule shifts in DIO-eNpHR mice (day 1 to 2: $P = 0.020$; day 1 to 3: $P = 0.010$; day 2 to 3: $P > 0.99$). **f**, Performance of Syn-eNpHR mice did not change (day 1 to 2: $P = 0.58$; day 1 to 3: $P = 0.82$; day 2 to 3: $P > 0.99$). For Fig. 4h,j, full statistics are: optogenetic inhibition changes the similarity of activity vectors specifically in DIO-eNpHR mice (two-way ANOVA, main effect of task day: $P = 0.017$; task day × group interaction: $P = 0.32$). **h**, For controls ($n = 6$ mice), there is no change across days in the similarity between activity vectors after RS errors and those after correct decisions (day 1 to 2: $P = 0.57$; day 1 to 3: $P = 0.32$; day 2 to 3: $P = 0.90$). **i**, In DIO-eNpHR mice ($n = 6$ mice), there is an increase in the similarity between activity vectors after RS errors and those after correct decisions from day 1 (before optogenetic inhibition) to days 2 and 3 (during and after inhibition, respectively) (day 1 to 2: $P = 0.044$; day 1 to 3: $P = 0.0067$; day 2 to 3: $P = 0.71$). **j**, For Syn-eNpHR mice ($n = 6$ mice), there is no change across days in the similarity between activity vectors after RS errors and those after correct decisions (day 1 to 2: $P = 0.99$; day 1 to 3: $P = 0.93$; day 2 to 3: $P = 0.98$). Two-way ANOVA to compare the change in activity vector similarity from day 1 to 2 in the DIO-eNpHR group to a group consisting of the eNpHR-negative and synapsin-eNpHR mice, using mouse, day, and day × group interaction as factors revealed a significant day × group interaction ($P = 0.040$). **k–m**, The similarity between activity vectors after decisions on error trials during the initial association (IA) and those during the RS. There is increased similarity across days in DIO-eNpHR mice ($n = 6$; two-way ANOVA (task day × virus); interaction: $P = 0.025$). **k**, There is no change in the similarity of population activity vectors after IA errors and RS errors across days in controls ($n = 6$; day 1 to 2: $P = 0.99$; day 1 to 3: $P = 0.52$; day 2 to 3: $P = 0.50$). **l**, During and following optogenetic inhibition on days 2 and 3 in DIO-eNpHR mice, there is an increase in the similarity of population activity vectors following IA versus RS error trials (day 1 to 2: $P = 0.0038$; day 1 to 3: $P = 0.0045$; day 2 to 3: $P = 0.95$). **m**, In Syn-eNpHR mice, there is no change in the similarity of population activity vectors following IA versus RS errors trials across days ($n = 6$; day 1 to 2: $P = 0.41$; day 1 to 3: $P = 0.90$; day 2 to 3: $P = 0.24$). **n**, Controls have increases in average activity (fraction of frames in which a neuron is active, averaged across all neurons) during the 10 s following RS errors compared to the 10 s following RS correct decisions on all days ($n = 6$; two-way ANOVA, main effect of trial type: $P = 0.010$; day 1: Bonferroni $P = 0.022$; day 2: Bonferroni $P = 0.015$; day 3: Bonferroni $P = 0.016$). **o**, DIO-eNpHR mice have an increase in average activity after errors that depends on the day ($n = 6$; two-way ANOVA (task day × trial type); interaction: $P = 0.0003$). This difference occurs on day 1 (Bonferroni $P < 0.0001$), but is abolished with light delivery on day 2 (Bonferroni $P > 0.99$), and continues to be absent even without further light delivery on day 3 (Bonferroni $P > 0.99$). **p**, In Syn-eNpHR mice, there is an overall increase in average activity following error trials ($n = 6$; two-way ANOVA,

main effect of trial type: $P = 0.0088$). This occurs on days 1 (Bonferroni $P = 0.038$) and 3 (Bonferroni $P = 0.016$), but not with light delivery on day 2 (Bonferroni $P > 0.99$). Two-way ANOVA followed by Tukey post hoc comparisons was used unless otherwise noted.

## Reporting summary

Further information on research design is available in the Nature Portfolio Reporting Summary linked to this article.

## Data availability

The data that support the findings of this study are available from the corresponding author upon reasonable request. The underlying physiological data (trial-by-trial measurements of synchrony and population activity vectors) and associated MATLAB code are available on Zenodo at https://doi.org/10.5281/zenodo.7709805. Source data are provided with this paper.

## Code availability

Custom codes for analysis and modelling were written in MATLAB and are available on Zenodo (https://doi.org/10.5281/zenodo.7709805) as well as from the corresponding author upon request.

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

**Acknowledgements** V.S.S. is supported by grants from the US National Institute of Health (R01MH121342 and R01NS116594), the McKnight Endowment Fund for Neuroscience (Memory and Cognitive Disorders award), and the Brain Research Foundation (Scientific Innovations Award BRF-SIA-2019-02). K.K.A.C. is supported by the Institut National de la Santé et de la Recherche Médicale (Inserm) and the Marie Skłodowska-Curie Individual Fellowship (MSCA-IF). The authors thank O. Lavielle for providing PV-cre heterozygous mice; L. Frank and M. Brainard for feedback on earlier versions of this manuscript; members of the Alberto Bacci laboratory for discussions; and the ICM technical staff of the facilities PHENO-ICMice and Histomics. This work was also supported by 'Investissements d'avenir' ANR-10-IAIHU-06.

**Author contributions** K.K.A.C. and V.S.S. designed the study. K.K.A.C. conducted all experiments and analyses, with the exception of Flp- and Cre-dependent AAV-NpHR experiments with or without ChR2, synapsin-eNpHR dual-site voltage indicator experiments and controls, synapsin-eNpHR and synapsin-mCherry microendoscopic calcium-imaging experiments, and experiments in Extended Data Fig. 2e–h, which were performed by J.S. A.J.P. analysed the shorter-timeframe dual-site voltage indicator recordings in Extended Data Fig. 9f–l. M.L.T. performed surgeries for synapsin–eNpHR and synapsin–mCherry microendoscopic calcium-imaging experiments. K.K.A.C. and V.S.S. wrote the paper.

**Competing interests** The authors declare no competing interests.

**Additional information**
**Correspondence and requests for materials** should be addressed to Kathleen K. A. Cho or Vikaas S. Sohal.

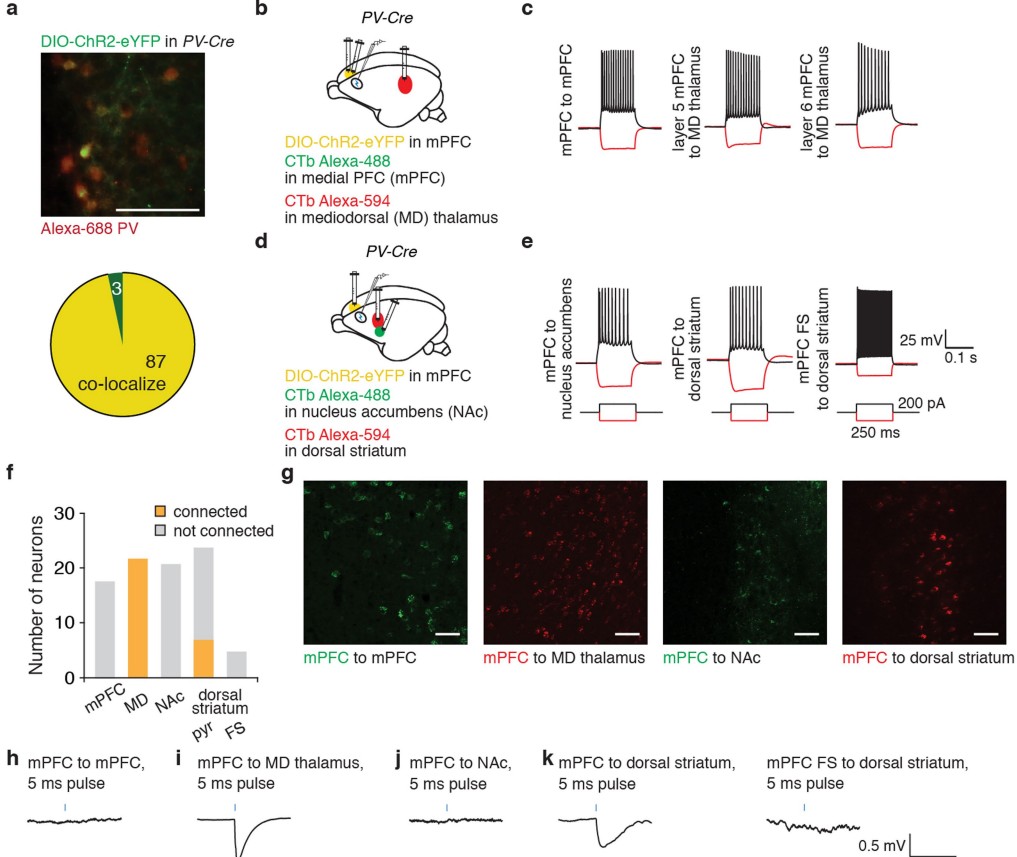

**Extended Data Fig. 1 | Callosal PV+ projections innervate mPFC neurons projecting to mediodorsal thalamus and dorsal striatum. a**, Example immunohistochemistry image showing staining for parvalbumin (PV, red) and eYFP expression after a unilateral injection of AAV-DIO-ChR2-eYFP in the mPFC of *PV-Cre* mice. ChR2-eYFP expression is 96.7% specific for PV colabeled neurons. **b**, Experimental design: AAV-DIO-ChR2-eYFP was injected into one mPFC of *PV-Cre* mice to express ChR2 in callosal PV terminals. To retrogradely-label neurons that project to the contralateral mPFC or mediodorsal (MD) thalamus, cholera toxin subunit B conjugated with Alexa Fluor 488 (CTb Alexa-488) was injected in the mPFC ipsilateral to the AAV-DIO-ChR2 injection, and Alexa Fluor 594-conjugated cholera toxin subunit B (CTb Alexa-594) was injected in MD thalamus (contralateral to the site of AAV-DIO-ChR2 injection) (*n* = 3 mice). Whole-cell recordings were made from labeled neurons (contralateral to the AAV-DIO-ChR2 injection) within prefrontal brain slices. **c**, Example current-clamp responses to hyperpolarizing or depolarizing current pulses in retrogradely-labeled pyramidal neurons projecting to contralateral mPFC and MD thalamus. **d**, Experimental design: Injection of AAV-DIO-ChR2-eYFP in one mPFC of *PV-Cre* mice to express channelrhodopsin in callosal PV terminals. To retrogradely-label neurons that project to nucleus

accumbens (NAc) or dorsal striatum, CTb Alexa-488 was injected in NAc and CTb Alexa-594 was injected in dorsal striatum (both contralateral to the AAV-DIO-ChR2 injection) (*n* = 3 mice). Whole-cell recordings were made from labeled neurons within prefrontal brain slices. **e**, Example current-clamp responses to hyperpolarizing or depolarizing current pulses in retrogradely-labeled neurons projecting to NAc or dorsal striatum. A small number of fast-spiking (FS) neurons were present among retrogradely-labeled neurons projecting to dorsal striatum. **f**, During optogenetic stimulation of callosal PV terminals, we observed consistent synaptic responses in all MD-projecting neurons and a fraction of dorsal striatum-projecting pyramidal neurons. **g**, Representative images showing CTb labeling of mPFC neurons projecting to the contralateral mPFC, ipsilateral MD thalamus, ipsilateral NAc, or ipsilateral dorsal striatum. **h**–**k**, A blue light flash (5 ms, 470 nm; denoted as a blue bar) delivered through the 40x objective were used to optogenetically stimulate callosal PV terminals. Example recordings from mPFC pyramidal neurons projecting to MD thalamus or dorsal striatum, showing optogenetically-evoked inhibitory postsynaptic potentials (IPSPs) in the presence of DNQX and APV. Scale bars, 100 μm, unless otherwise indicated.

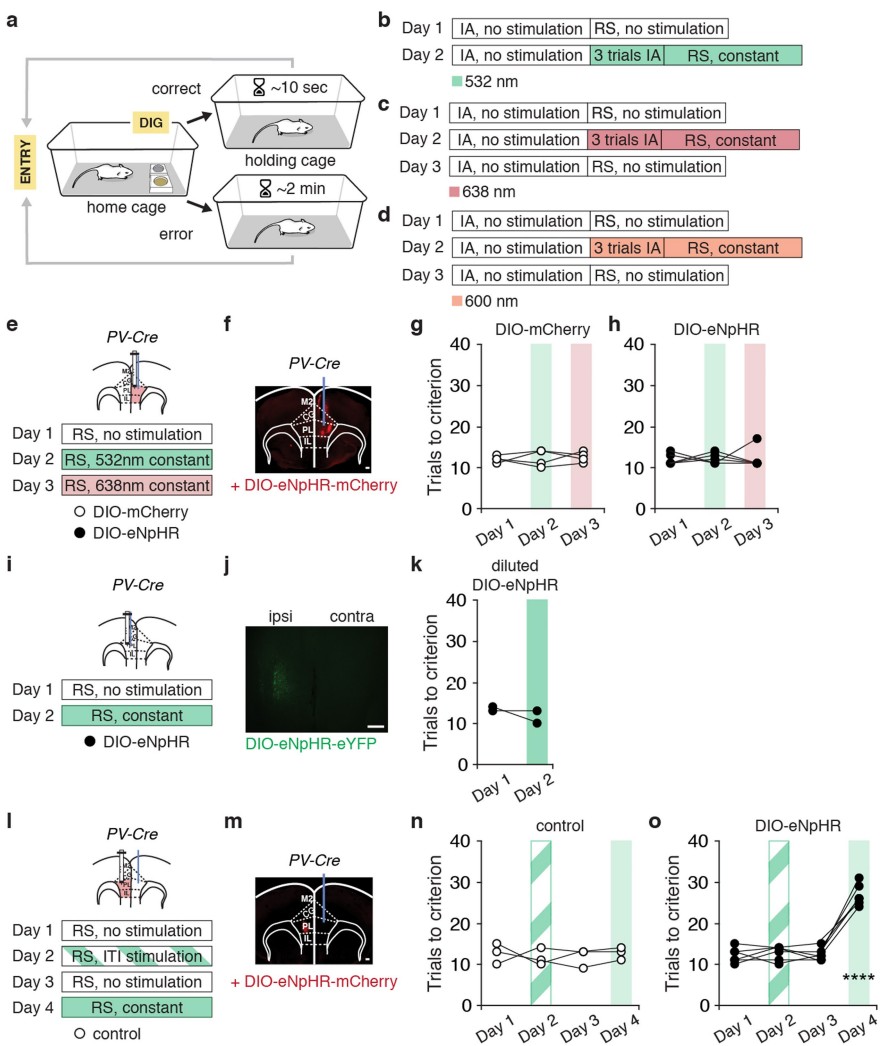

**Extended Data Fig. 2 | Rule shift task description and control experiments using weak light delivery or weak viral expression. a**, Trial timeline. A mouse begins each trial when it is placed in the home cage, then makes a decision, indicated by digging in one bowl. If the mouse is correct, the food reward is consumed. The mouse is then transferred to the holding cage until the next trial. The intertrial interval (ITI) is longer after errors. **b**, For optogenetic inhibition behavior experiments in Fig. 2, Day 1: no light delivery during the initial association (IA), nor during the rule shift (RS); Day 2: no light is delivered while mice learn the IA, but once mice meet the criterion for learning (8/10 consecutive trials correct), we begin delivering continuous 532 nm light and test the animal for 3 additional IA trials, before switching to the RS portion of the task. **c**, For optogenetic inhibition dual-site voltage indicator experiments in Fig. 3, Day 1: no light delivery during the IA, nor during the RS; Day 2: no light delivery during the IA, but continuous light delivery of 638 nm begins during 3 additional IA trials, followed by the RS; Day 3: no light delivery during the IA, nor during the RS. **d**, For optogenetic inhibition + microendoscopic calcium imaging experiments in Fig. 4, Day 1: no light delivery during the IA, nor during the RS; Day 2: no light delivery during the IA, but continuous light delivery of 600 nm begins during 3 additional IA trials, followed by the RS; Day 3: no light delivery during the IA, nor during the RS. **e**, Control experiments to verify that weak light delivery does not affect RS learning. Experimental design: Day 1, no light delivery; Day 2, continuous 0.1 mW 532 nm light is delivered during the RS; Day 3, continuous light at 0.1 mW 638 nm for inhibition during the RS. **f**, Representative image showing AAV-DIO-eNpHR-mCherry (DIO-eNpHR) expression and a fiber-optic cannula in the same mPFC hemisphere in a *PV-Cre*

mouse. Scale bar, 100 μm. **g**, **h**, Performance in the rule shift task did not vary across days in either controls ($n = 4$), or DIO-eNpHR-injected mice ($n = 5$; two-way ANOVA (task day × virus); interaction: $F_{(2,14)} = 0.01721$, $P = 0.983$). **i**, Experimental design: Day 1, no light delivery; Day 2, continuous 2.5 mW 532 nm light is delivered during the RS. **j**, Representative image showing diluted (200 nL in 800 nL saline) DIO-eNpHR expression and a fiber-optic cannula in the same mPFC hemisphere in a *PV-Cre* mouse. Scale bar, 250 μm. **k**, Light stimulation of PV cells infected with lower virus titer did not alter RS performance across days ($n = 2$ mice; two-tailed paired *t*-test, $P = 0.5000$). **l**, Experimental design: Day 1, no light delivery; Day 2, light delivery during the inter-trial interval (ITI) during the RS for optogenetic inhibition of callosal PV terminals; Day 3, no light was delivered; Day 4, continuous light during the RS. **m**, Representative image showing DIO-eNpHR-mCherry (DIO-eNpHR) expression in one mPFC and a fiber-optic cannula implanted in the contralateral mPFC. Scale bar, 100 μm. **n**, **o**, Optogenetic inhibition of mPFC callosal PV terminals impairs rule shift performance in DIO-eNpHR-expressing mice only when delivered during both the trial and inter-trial intervals of the RS ($n = 6$) compared to controls ($n = 3$; two-way ANOVA (task day × virus); interaction: $F_{(3,28)} = 23.31$, $P < 0.0001$). **n**, Performance of controls was not different from eNpHR-expressing mice on Day 1 (post hoc $t_{(28)} = 0.3489$, $P > 0.9999$), Day 2 (post hoc $t_{(28)} = 0.6978$, $P > 0.9999$), nor Day 3 (post hoc $t_{(28)} = 0.4652$, $P > 0.9999$). **o**, Inhibition disrupts rule shift performance in DIO-eNpHR-expressing mice compared to controls when light is delivered continuously throughout the RS on Day 4 (post hoc $t_{(28)} = 9.886$, $P < 0.0001$). Two-way ANOVA followed by Bonferroni post hoc comparisons was used. ****$P < 0.0001$.

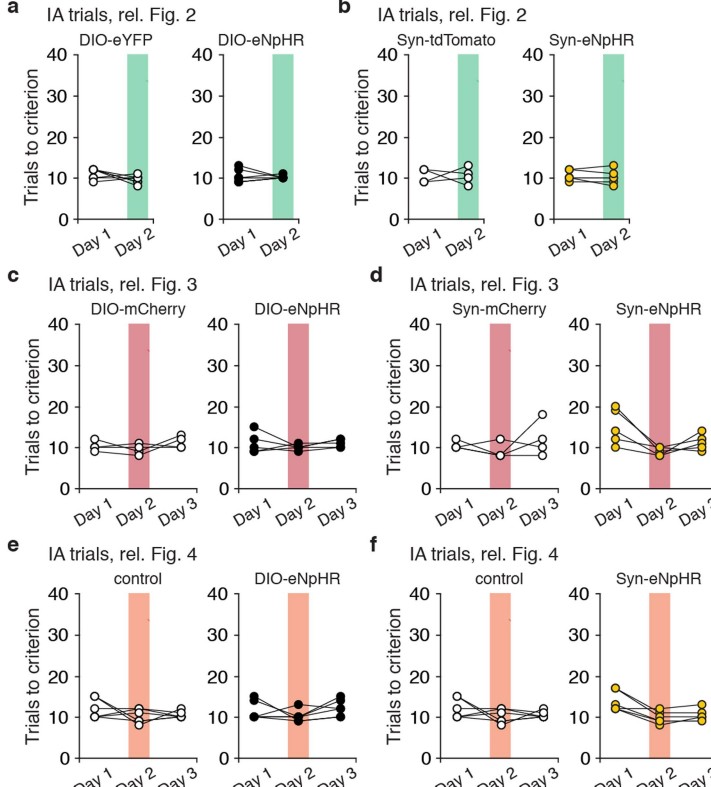

**Extended Data Fig. 3 | Initial association performance. a**, Related to the experiments in Fig. 2, there is no difference in the number of trials needed to reach the criterion for learning the initial association (IA) for DIO-eYFP controls ($n = 7$) vs. DIO-eNpHR-expressing mice ($n = 8$; two-way ANOVA (task day × virus); interaction: $F_{(1,13)} = 1.936$, $P = 0.1874$). **b**, Related to the experiments in Fig. 2, there is no difference in the number of IA trials to reach the learning criterion for Syn-tdTomato controls ($n = 4$) vs. Syn-eNpHR-expressing mice ($n = 6$; two-way ANOVA (task day × virus); interaction: $F_{(1,8)} = 0.05424$, $P = 0.8217$). **c**, Related to the experiments in Fig. 3, there is no difference in the number of IA trials needed to reach the learning criterion for DIO-mCherry controls ($n = 4$) vs. DIO-eNpHR-expressing mice ($n = 5$; two-way ANOVA (task day × virus); interaction: $F_{(2,14)} = 0.3027$, $P = 0.7435$). **d**, Related to the experiments in Fig. 3,

there is no difference in the number of IA trials needed to reach the learning criterion for Syn-mCherry controls ($n = 4$) vs. Syn-eNpHR-expressing mice ($n = 5$; two-way ANOVA (task day × virus); interaction: $F_{(2,14)} = 2.131$, $P = 0.1557$). **e**, Related to the experiments in Fig. 4, there is no difference in the number of IA trials needed to reach the learning criterion for controls (DIO-mCherry or Syn-mCherry) ($n = 6$) vs. DIO-eNpHR-expressing mice ($n = 6$; two-way ANOVA (task day × virus); interaction: $F_{(2,20)} = 0.8831$, $P = 0.4290$). **f**, Related to the experiments in Fig. 4, there is no difference in the number of IA trials needed to reach the learning criterion for controls (DIO-mCherry or Syn-mCherry; $n = 6$) vs. Syn-eNpHR-expressing mice ($n = 6$; two-way ANOVA (task day × virus); interaction: $F_{(2,20)} = 1.372$, $P = 0.2764$). Two-way ANOVA followed by Bonferroni post hoc comparisons was used.

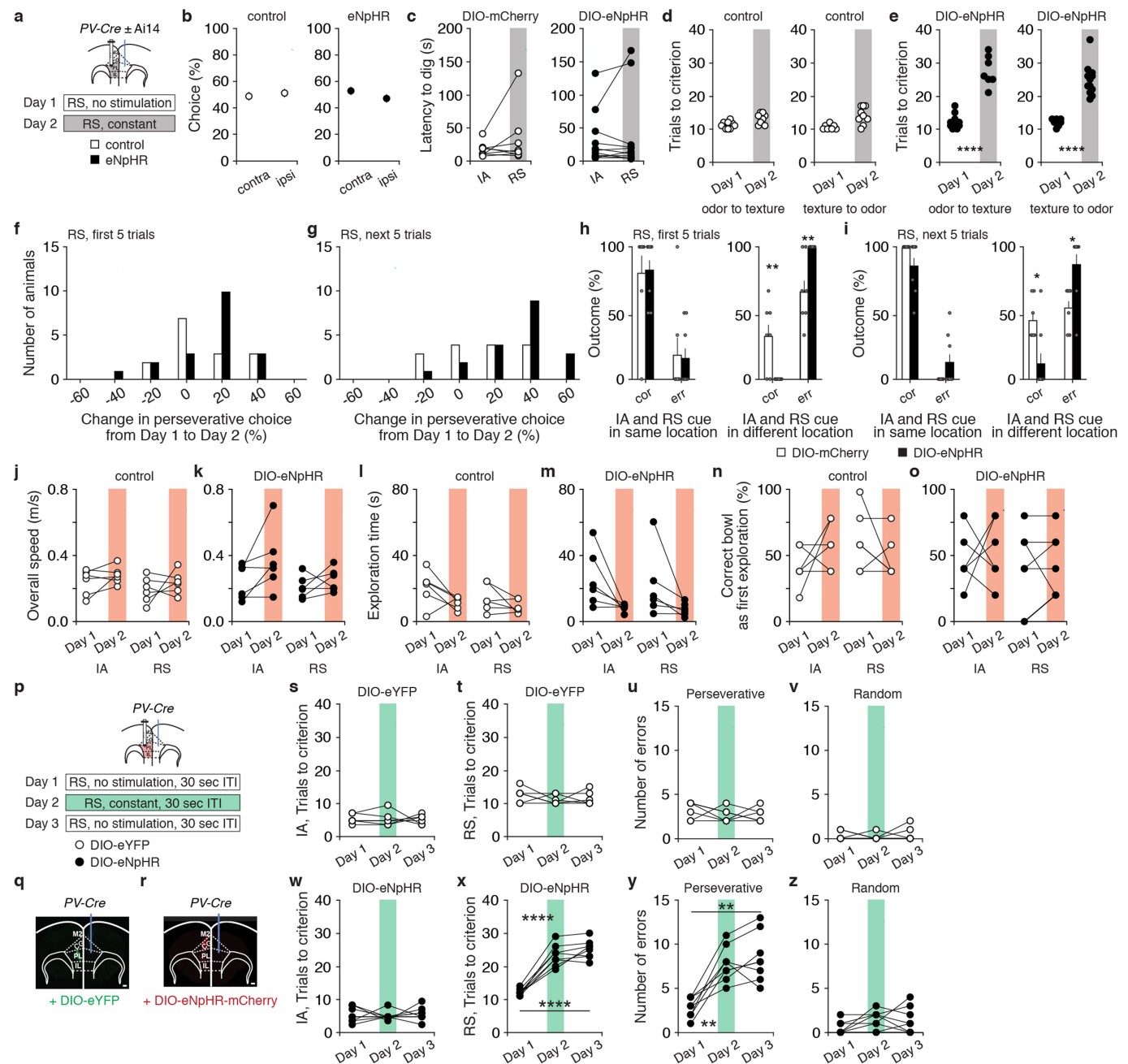

**Extended Data Fig. 4** | See next page for caption.

**Extended Data Fig. 4 | Motor and animal behavior during optogenetic manipulations. a**, We compared behavior between *PV-Cre* ± Ai14 mice injected with control virus (AAV-DIO-eYFP, AAV-DIO-mCherry, AAV-Syn-tdTomato, AAV-Syn-mCherry) vs. experimental eNpHR-expressing mice (injected with AAV-DIO-eNpHR-BFP, AAV-DIO-eNpHR-mCherry, AAV-Syn-eNpHR3.0-eYFP, or AAV-Syn-eNpHR-mCherry), for experiments in which light for optogenetic inhibition was delivered on Day 2. **b**, There is no bias in the fraction of choices made that were ipsilateral versus contralateral to the side of optogenetic inhibition ($n = 28$ DIO- and Syn-controls, $n = 41$ DIO- and Syn-eNpHR mice; two-way ANOVA (side bias × virus); interaction: $F_{(1,67)} = 0.7813, P = 0.3799$). For this, we specifically analyzed the final three initial association (IA) trials on Day 2. (On Day 2, after mice reach the learning criterion for the IA, we began delivering light for optogenetic inhibition and performed three additional IA trials, followed by the rule shift). **c**, In experiments that included time-stamped behavior (dual-site voltage indicator and microendoscopic calcium imaging experiments), there is no difference in the latency to dig (aka, 'time to choice') when mice receive optogenetic inhibition during the first 5 rule shift (RS) trials vs. no optogenetic inhibition during the first 5 IA trials ($n = 8$ DIO-mCherry controls, $n = 11$ DIO-eNpHR mice; two-way ANOVA (type of task × virus); interaction: $F_{(1,17)} = 0.4236, P = 0.5239$). **d–e**, RS performance as a function of the type of cue shifts (odor to texture, texture to odor), task day (Day 1: no light; Day 2: light on for optogenetic inhibition), and group ($n = 17$ eNpHR-negative controls, $n = 19$ DIO-eNpHR mice) (two-way ANOVA (virus × task day); interaction: $F_{(3,64)} = 24.00, P < 0.0001$; there was no difference in performance for control mice on Day 1 vs. 2 for odor to texture shifts (Tukey's post hoc $q_{(64)} = 1.943, P = 0.5203$); there was not a significant difference in performance for control mice on Day 1 vs. 2 for texture to odor shifts (Tukey's post hoc $q_{(64)} = 3.559, P = 0.0668$); there was a marked significant difference in performance for DIO-eNpHR mice on Day 1 vs. 2 for odor to texture shifts (Tukey's post hoc $q_{(64)} = 16.16, P < 0.0001$); there was a marked significant difference in performance for DIO-eNpHR mice on Day 1 vs. 2 for texture to odor shifts (Tukey's post hoc $q_{(64)} = 13.3, P < 0.0001$). **f–g**, To determine whether the effects of optogenetic inhibition on the RS manifest immediately vs. only accrue after prolonged light delivery, we calculated the percentage of perseverative errors separately for the first 5 RS trials versus the next 5 RS trials, across control and experimental cohorts. Optogenetic inhibition of callosal PV projections causes mice to perseverate more on Day 2 compared to Day 1 for both the first 5 RS trials ($n = 19$ DIO-eNpHR mice; two-way ANOVA (trial type × task day); interaction: $F_{(1,18)} = 7.496, P = 0.0135$; post hoc $t_{(18)} = 2.581, P = 0.0376$) and the next 5 RS trials (post hoc $t_{(18)} = 6.453, P < 0.0001$). By contrast, there was no change from Day 1 to Day 2 in the percentage of perseverative choices within either the first 5 ($n = 15$ controls; two-way ANOVA (type of trial type × task day); interaction: $F_{(1,14)} = 0.09894, P = 0.7577$; post hoc $t_{(14)} = 1.557, P = 0.2836$) or next 5 RS trials in controls (post hoc $t_{(14)} = 2.002, P = 0.1302$). **h**, The proportion of correct vs. error decisions is plotted as a function of whether the cues that would be rewarded in the IA and RS are located in the same vs. different bowls, during the first 5 RS trials ($n = 7$ DIO-mCherry controls, $n = 8$ DIO-eNpHR mice; same location two-way ANOVA (trial type × virus); interaction: $F_{(1,13)} = 0.02208, P = 0.8842$; same and correct post hoc $t_{(26)} = 0.1486, P > 0.9999$; same and incorrect post hoc $t_{(26)} = 0.1486, P > 0.9999$; different location two-way ANOVA (trial type × virus); interaction: $F_{(1,13)} = 13.87, P = 0.0026$; different and correct post hoc $t_{(26)} = 3.724, P = 0.0019$; different and incorrect post hoc $t_{(26)} = 3.724, P = 0.0019$). **i**, Same as **h** but for the next 5 RS trials ($n = 7$ DIO-mCherry control mice; same location two-way ANOVA (trial type × virus); interaction: $F_{(1,13)} = 3.214, P = 0.0963$; same and correct post hoc $t_{(26)} = 1.793, P = 0.1693$; same and incorrect post hoc $t_{(26)} = 1.793, P = 0.1693$;

different location two-way ANOVA (trial type × virus); interaction: $F_{(1,13)} = 8.948, P = 0.0104$; different and correct post hoc $t_{(26)} = 2.991, P = 0.0120$; different and incorrect post hoc $t_{(26)} = 2.991, P = 0.0120$). **j–k**, The overall speed (meters per second) of mice during the first 5 IA and RS trials across days in the cohort of mice used for microendoscopic $Ca^{2+}$ imaging ($n = 6$ eNpHR-negative controls; $n = 6$ DIO-eNpHR mice; two-way ANOVA (IA vs. RS × task day) for control mice: interaction: $F_{(1,10)} = 0.00271, P = 0.9595$; two way-ANOVA (IA vs. RS × task day) for DIO-eNpHR mice: interaction: $F_{(1,10)} = 1.378, P = 0.2677$). **l–m**, There is no difference in the amount of time (seconds) mice spent exploring bowls before making a decision during the first 5 IA and RS trials across days in the microendoscope experimental dataset ($n = 6$ eNpHR-negative controls; $n = 6$ DIO-eNpHR mice; two-way ANOVA (IA vs. RS × task day) for control mice: interaction: $F_{(1,10)} = 0.5053, P = 0.4934$; two way-ANOVA (IA vs. RS × task day) for DIO-eNpHR mice: interaction: $F_{(1,10)} = 0.1147, P = 0.7419$). **n–o**, The first move of the mouse toward the correct bowl (percent) during the first 5 IA and RS trials across days in the $Ca^{2+}$ imaging experimental dataset ($n = 6$ eNpHR-negative controls; $n = 6$ DIO-eNpHR mice; two-way ANOVA (IA vs. RS × task day) for control mice: interaction: $F_{(1,10)} = 3.347, P = 0.0973$; two way-ANOVA (IA vs. RS × task day) for DIO-eNpHR mice: interaction: $F_{(1,10)} = 0.1316, P = 0.7244$). **p–z**, Effects of inhibiting callosal PV+ projections during a version of the RS task using a shorter (30 second) intertrial interval (ITI). **p**, Experimental design: Day 1, no light delivery; Day 2, continuous light during the RS for optogenetic inhibition of callosal PV terminals ; Day 3, no light was delivered. **q**, Representative image showing DIO-eYFP expression in one mPFC and a fiber-optic cannula implanted in the contralateral mPFC in a *PV-Cre* mouse. **r**, Representative image showing DIO-eNpHR-mCherry (DIO-eNpHR) expression in one mPFC and a fiber-optic cannula implanted in the contralateral mPFC in a *PV-Cre* mouse. **s,w**, IA performance with a 30 s ITI in eNpHR-negative mice ($n = 6$) and eNpHR-expressing mice ($n = 8$; two-way ANOVA (task day × virus); interaction: $F_{(2,24)} = 0.1585, P = 0.8543$). **t,x**, Optogenetic inhibition of mPFC callosal PV terminals with a 30 s ITI impairs rule shift performance in DIO-eNpHR mice ($n = 8$) compared to controls ($n = 6$; two-way ANOVA (task day × virus); interaction: $F_{(2,24)} = 50.79, P < 0.0001$). **t**, Performance of DIO-eYFP controls did not change from Day 1 to Day 2 (Tukey's post hoc $q_{(5)} = 1.606, P = 0.5356$), Day 1 to Day 3 (Tukey's post hoc $q_{(5)} = 1.035, P = 0.7567$), nor Day 2 to Day 3 (Tukey's post hoc $q_{(5)} = 0.4344, P = 0.9498$). **x**, Inhibition disrupts rule shift performance in DIO-eNpHR mice from Day 1 to Day 2 (Tukey's post hoc $q_{(7)} = 15.34, P < 0.0001$), Day 1 to Day 3 (Tukey's post hoc $q_{(7)} = 21.75, P < 0.0001$), but not Day 2 to Day 3 (Tukey's post hoc $q_{(7)} = 2.679, P = 0.2101$). **u, y**, Optogenetic inhibition of callosal PV terminals increases perseverative errors in DIO-eNpHR mice ($n = 8$ mice) compared to DIO-eYFP controls ($n = 6$ mice; two-way ANOVA (task day × virus) interaction: $F_{(2,24)} = 19.79, P < 0.0001$). **v, z**, Optogenetic inhibition of callosal PV terminals has no effect on random errors in DIO-eNpHR mice ($n = 8$) compared to DIO-eYFP controls ($n = 6$; two-way ANOVA (task day × virus); interaction: $F_{(2,24)} = 1.079, P = 0.3559$). **u, v**, Light delivery does not affect the number of perseverative (post hoc $t_{(5)} = 0.000 – 1.000, P > 0.9999$) or random (post hoc $t_{(5)} = 0.4152 – 0.7906, P > 0.9999$) errors in DIO-eYFP controls across days. **y, z**, Optogenetic inhibition of callosal PV terminals on Day 2 increased the number of perseverative (post hoc $t_{(7)} = 6.008, P = 0.0016$ from Day 1 to Day 2; post hoc $t_{(7)} = 5.844, P = 0.0019$ from Day 1 to Day 3; but not from Day 2 to Day 3: post hoc $t_{(7)} = 1.111, P = 0.9093$) but not random (post hoc $t_{(7)} = 0.000 – 2.198, P = 0.1918 – > 0.9999$) errors compared to no stimulation. Two-way ANOVA followed by Bonferroni post hoc comparisons was used unless otherwise noted. Data were expressed as mean ± s.e.m. **$P < 0.01$, ****$P < 0.0001$; scale bar, 100 μm.

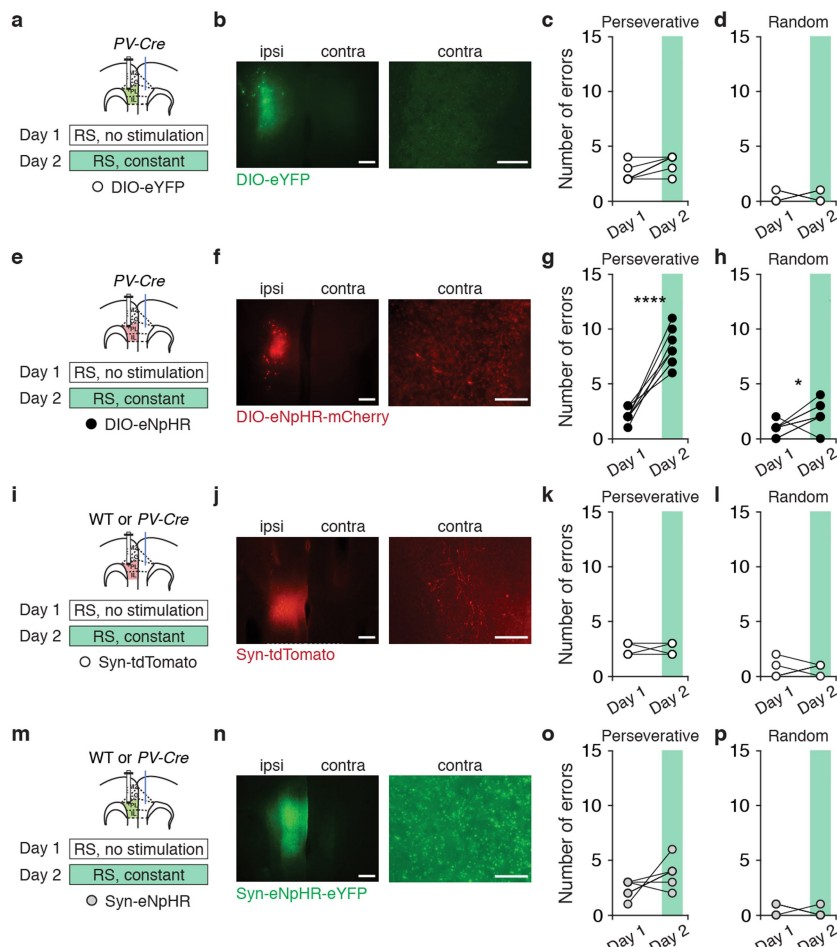

**Extended Data Fig. 5 | Optogenetic inhibition of callosal PV terminals (but not nonspecific inhibition of all callosal terminals) increases errors during rule shifts. a**, **e**, **i**, **m**, Experimental design: Day 1, no light delivery; Day 2, continuous light for optogenetic inhibition of callosal PV terminals during the rule shift (RS). **b**, **f**, **j**, **n**, Representative image showing viral expression in the mPFC ipsilateral to the injection (ipsi), and labeled callosal terminals in the contralateral mPFC (contra). **c**, **g**, Optogenetic inhibition of callosal PV terminals increases perseverative errors in DIO-eNpHR mice ($n = 8$ mice) compared to DIO-eYFP controls ($n = 7$ mice; two-way ANOVA (task day × virus); interaction: $F_{(1,13)} = 35.71$, $P < 0.0001$). **d**, **h**, Optogenetic inhibition of callosal PV terminals has a marginal effect on random errors in DIO-eNpHR mice ($n = 8$ mice) compared to DIO-eYFP controls ($n = 7$ mice; two-way ANOVA (task day × virus);

interaction: $F_{(1,13)} = 4.617$, $P = 0.0511$). **c**, **d**, Light delivery does not affect the number of perseverative (post hoc $t_{(13)} = 1.877$, $P = 0.1662$) or random (post hoc $t_{(13)} = 0.0$, $P > 0.9999$) errors in DIO-eYFP controls. **g**, **h**, Optogenetic inhibition of callosal PV terminals on Day 2 increased the number of perseverative (post hoc $t_{(13)} = 10.75$, $P < 0.0001$) and random (post hoc $t_{(13)} = 3.145$, $P = 0.0155$) errors compared to no stimulation on Day 1. **k**, **l**, **o**, **p**, Optogenetic inhibition of all callosal projections has no effect on perseverative errors in Syn-eNpHR mice ($n = 6$) compared to controls ($n = 4$; two-way ANOVA (task day × virus); interaction: $F_{(1,8)} = 0.0$, $P > 0.9999$) nor on random errors (two-way ANOVA (task day × virus); interaction: $F_{(1,8)} = 0.07805$, $P = 0.787$). Two-way ANOVA followed by Bonferroni post hoc comparisons was used. $*P < 0.05$, $****P < 0.0001$; scale bars, 250 μm and 100 μm, respectively.

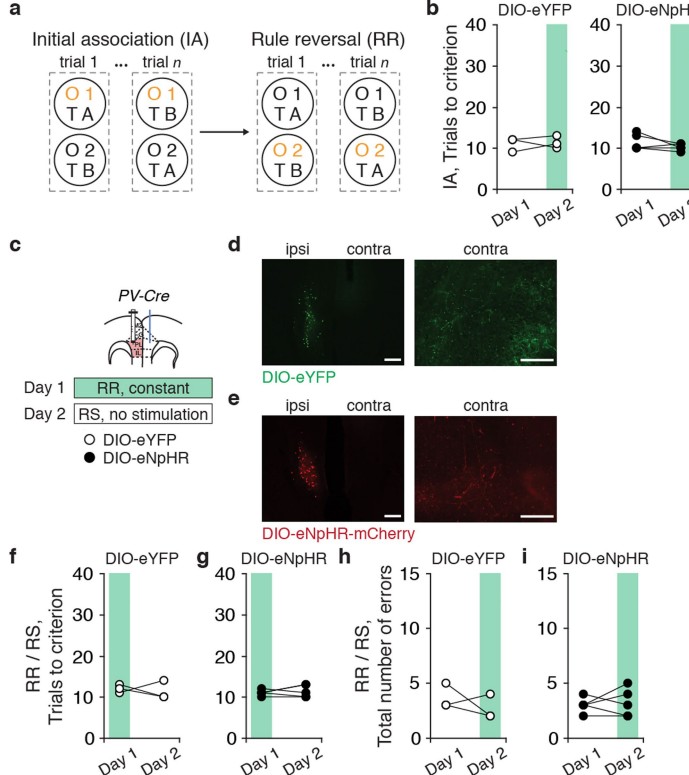

**Extended Data Fig. 6 | Optogenetic inhibition of callosal PV terminals during a rule reversal does not impair reversal learning or induce persistent deficits in subsequent rule shifts. a**, Schematic illustrating the rule reversal task, in which mice chose one of two bowls, each baited by an odor (O1 or O2) and texture (TA or TB) cue, to find a hidden food reward (the stimulus associated with reward is indicated in orange). Mice first learn an initial association (IA) between either an odor or texture cue (odor O1 in this case) and food reward. Once mice reach the learning criterion (eight correct out of ten consecutive trials), this association undergoes an intra-dimensional rule reversal (RR; e.g., from O1 to O2). **b**, There is no difference in the number of IA trials needed to reach the learning criterion for control (n = 3 mice; AAV-DIO-eYFP injected; DIO-eYFP) vs. eNpHR-expressing (n = 5 mice; AAV-DIO-eNpHR-mCherry injected; DIO-eNpHR) mice across days (two-way ANOVA (task day × virus); interaction: $F_{(1,6)}$ = 1.127, P = 0.3292).

**c**, Experimental design: Day 1, continuous light for optogenetic inhibition of callosal PV terminals during the RR; Day 2, no light delivery during the rule shift (RS). **d**, **e**, Representative images showing viral expression in the mPFC ipsilateral to the injection (ipsi), and labeled callosal terminals in the contralateral mPFC (contra). Scale bars, 250 μm and 100 μm, respectively. **f**, **g**, Performance of DIO-eYFP controls (n = 3, **f**) is similar to DIO-eNpHR mice (n = 5, **g**) from Day 1 to Day 2 (two-way ANOVA (task day × virus); interaction: $F_{(1,6)}$ = 0.4286, P = 0.5370). **h**, **i**, Optogenetic inhibition of callosal PV terminals does not change the total number of errors (perseverative and random) in DIO-eNpHR mice (n = 5 mice, **i**) compared to DIO-eYFP controls across days (n = 3 mice, **h**); two-way ANOVA (task day × virus); interaction: $F_{(1,6)}$ = 1.095, P = 0.3358. Two-way ANOVA followed by Bonferroni post hoc comparisons was used.

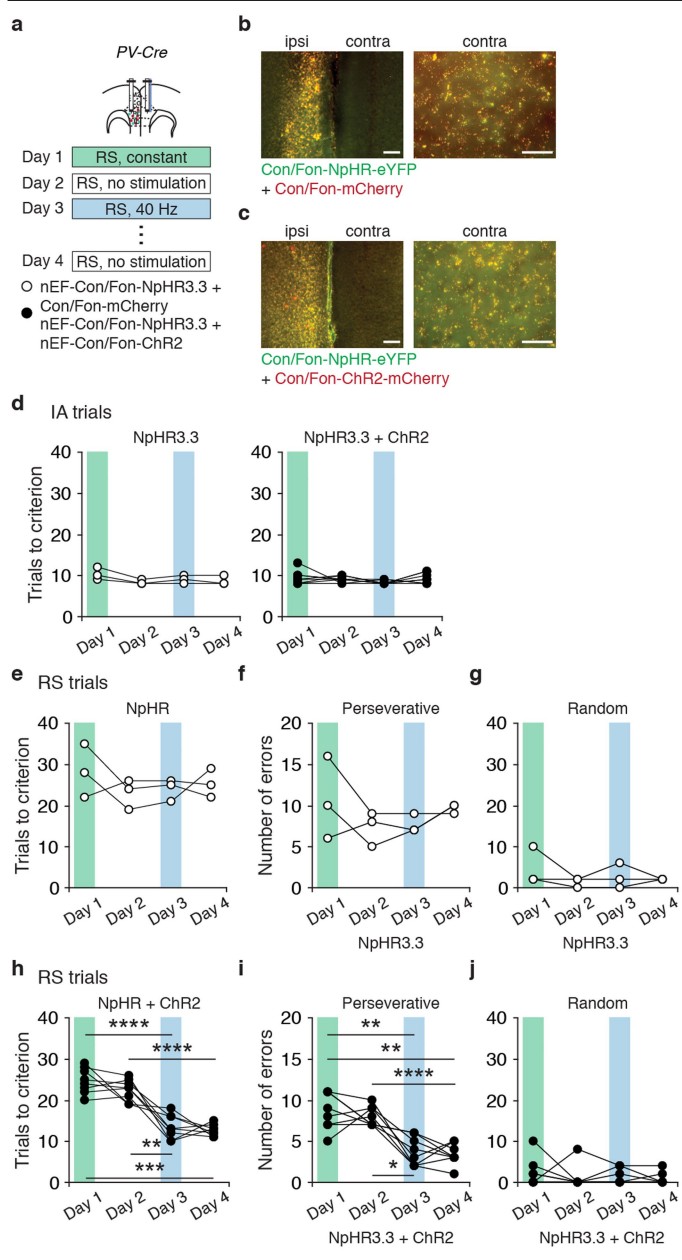

**Extended Data Fig. 7 | Optogenetic inhibition of callosal PV terminals induces persistent perseveration during rule shifts that is reversed by 40 Hz stimulation of the same terminals. a**, Experimental design: Day 1, continuous green light was delivered for terminal inhibition during the rule shift (RS); Day 2, no light delivery; Day 3, 40 Hz blue light was delivered for terminal stimulation during the RS; on Day 4, which occurred 1 week later, no light was delivered. **b**, Representative images showing AAV-nEF-Con/Fon-NpHR3.3-eYFP (Con/Fon-NpHR-eYFP) and AAV-Con/Fon-mCherry (Con/Fon-mCherry) expression in one mPFC (ipsi) and callosal PV⁺ fibers and terminals in the contralateral mPFC (contra). **c**, Representative images showing AAV-nEF-Con/Fon-NpHR3.3-eYFP (Con/Fon-NpHR-eYFP) and AAV-nEF-Con/Fon-ChR2-mCherry (Con/Fon-ChR2-mCherry) expression in one mPFC (ipsi) and callosal PV terminals in the contralateral mPFC (contra). **d**, There is no difference in the number of initial association (IA) trials needed to reach the learning criterion between mice which express only NpHR ($n = 3$) and those expressing both NpHR and ChR2 ($n = 8$; two-way ANOVA (task day × virus); interaction: $F_{(3,27)} = 1.317$, $P = 0.2892$). **e**,**h**, In mice which express NpHR only (**e**; $n = 3$), optogenetic inhibition on Day 1 causes mice to take a large number of trials to learn the rule shift, and this does not change across subsequent days (post hoc $t_{(2)} = 0.3941–1.732$, $P > 0.9999$). By contrast, rule shift performance in mice expressing both NpHR and ChR2 (**h**; $n = 8$) is significantly different across days than that of mice which express NpHR-only (two-way ANOVA (task day × virus); interaction: $F_{(3,27)} = 6.747$, $P = 0.0015$). Optogenetic inhibition of mPFC callosal PV terminals causes NpHR+ChR2-expressing mice ($n = 8$) to take a large number of trials to learn rule shifts on Day 1 and this does not change on Day 2 (no light) (post hoc $t_{(7)} = 1.446$, $P > 0.9999$). However, RS learning is rescued by 40 Hz optogenetic stimulation on Day 3 (post hoc $t_{(7)} = 10.91$, $P < 0.0001$) and this improvement does not change on 'Day 4' of testing which occurs one week later (post hoc $t_{(7)} = 0.6394$, $P > 0.9999$). **f**, **i**: Optogenetic inhibition followed by stimulation of callosal PV terminals changes the number of perseverative errors in mice expressing both NpHR and ChR2 ($n = 8$ mice) compared to controls expressing only NpHR ($n = 3$ mice; two-way ANOVA; main effect of task day: $F_{(2.015,18.13)} = 7.167$, $P = 0.0050$; main effect of virus: $F_{(1,9)} = 14.31$, $P = 0.0043$; interaction: $F_{(3,27)} = 5.324$, $P = 0.0052$). By contrast, there is no difference in numbers of random errors (two-way ANOVA; main effect of task day: $F_{(1.491,13.42)} = 1.706$, $P = 0.2189$; main effect of virus: $F_{(1,9)} = 1.523$, $P = 0.2483$; interaction: $F_{(3,27)} = 0.2901$, $P = 0.8322$). **f**, Once mice expressing NpHR only receive optogenetic inhibition on Day 1, the number of perseverative errors is stable across days ($n = 3$ mice; Day 1 to Day 2: post hoc $t_{(2)} = 1.222$, $P > 0.9999$; Day 1 to Day 3: post hoc $t_{(2)} = 1.299$, $P > 0.9999$; Day 1 to Day 4: post hoc $t_{(2)} = 0.3111$, $P > 0.9999$; Day 2 to Day 3: post hoc $t_{(2)} = 0.3780$, $P > 0.9999$; Day 2 to Day 4: post hoc $t_{(2)} = 1.606$, $P > 0.9999$; Day 3 to Day 4: post hoc $t_{(2)} = 2.000$, $P > 0.9999$). **g**, In mice that express NpHR only ($n = 3$ mice), numbers of random errors are also stable across days (Day 1 to Day 2: post hoc $t_{(2)} = 1.387$, $P > 0.9999$; Day 1 to Day 3: post hoc $t_{(2)} = 1.732$, $P > 0.9999$; Day 1 to Day 4: post hoc $t_{(2)} = 1.000$, $P > 0.9999$; Day 2 to Day 3: post hoc $t_{(2)} = 1.000$, $P > 0.9999$; Day 2 to Day 4: post hoc $t_{(2)} = 1.000$, $P > 0.9999$; Day 3 to Day 4: post hoc $t_{(2)} = 0.3780$, $P > 0.9999$). **i**, 40 Hz stimulation of callosal PV terminals on Day 3 reduces the number of perseverative errors in mice expressing both NpHR and ChR2 ($n = 8$; Day 1 to Day 2: post hoc $t_{(7)} = 0.6494$, $P > 0.9999$; Day 1 to Day 3: post hoc $t_{(7)} = 5.218$, $P = 0.0074$; Day 1 to Day 4: post hoc $t_{(7)} = 6.416$, $P = 0.0022$; Day 2 to Day 3: post hoc $t_{(7)} = 4.822$, $P = 0.0115$; Day 2 to Day 4: post hoc $t_{(7)} = 12.33$, $P < 0.0001$; Day 3 to Day 4: post hoc $t_{(7)} = 0.4971$, $P > 0.9999$). **j**, 40 Hz stimulation on Day 3 does not affect the number of random errors in mice expressing both NpHR and ChR2 ($n = 8$ mice; Day 1 to Day 2: post hoc $t_{(7)} = 0.7061$, $P > 0.9999$; Day 1 to Day 3: post hoc $t_{(7)} = 0.6417$, $P > 0.9999$; Day 1 to Day 4: post hoc $t_{(7)} = 1.210$, $P > 0.9999$; Day 2 to Day 3: post hoc $t_{(7)} = 0.3140$, $P > 0.9999$; Day 2 to Day 4: post hoc $t_{(7)} = 0.4237$, $P > 0.9999$; Day 3 to Day 4: post hoc $t_{(7)} = 0.7977$, $P > 0.9999$). Two-way ANOVA followed by Bonferroni post hoc comparisons was used. *$P < 0.05$, **$P < 0.01$, ***$P < 0.001$, ****$P < 0.0001$; scale bars, 100 μm and 50 μm, respectively.

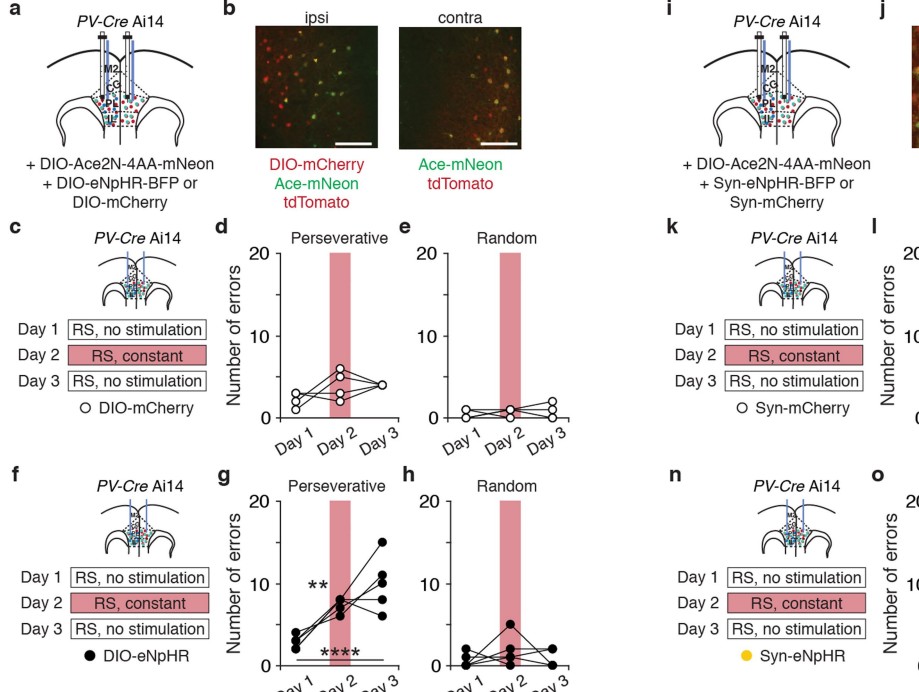

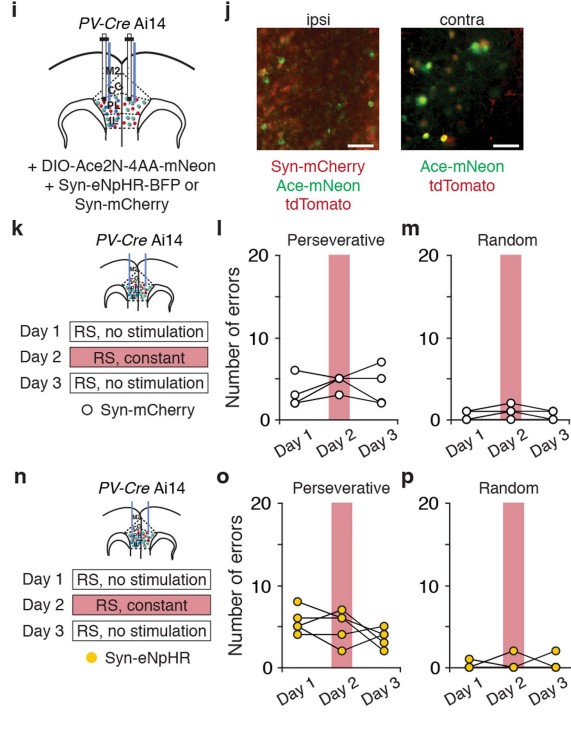

**Extended Data Fig. 8 | Optogenetic inhibition of callosal PV terminals, delivered while measuring signals from voltage indicators, increases errors during rule shifts (RS). a**, *PV-Cre* Ai14 mice had bilateral AAV-DIO-Ace2N-4AA-mNeon (Ace-mNeon) injections, an ipsilateral AAV-DIO-eNpHR-BFP (DIO-eNpHR) or AAV-DIO-mCherry injection and multimode fiber-optic implants in both prefrontal cortices. **b**, Representative images from mice injected with a control virus (DIO-mCherry), showing mCherry, Ace-mNeon, and tdTomato expression in the mPFC ipsilateral to the virus injection (ipsi), and Ace-mNeon and tdTomato in the contralateral hemisphere (contra). **c, f**, Experimental design: Day 1, no light delivery; Day 2, continuous light for inhibition during the rule shift (RS); Day 3, no light delivery. **d, e, g, h**, Optogenetic inhibition of callosal PV terminals increases perseverative errors in DIO-eNpHR-expressing mice (*n* = 5 mice, **g-h**) compared to DIO-mCherry-expressing controls (*n* = 4 mice, **d-e**; two-way ANOVA (task day × virus); interaction: $F_{(2,14)}$ = 5.226, *P* = 0.0202), but has no effect on random errors (two-way ANOVA (task day × virus); interaction: $F_{(2,14)}$ = 0.4552, *P* = 0.6434). **d, e**, Light delivery does not affect the number of perseverative (Day 1 to Day 2: post hoc $t_{(14)}$ = 1.392, *P* = 0.5566; Day 1 to Day 3: post hoc $t_{(14)}$ = 1.392, *P* = 0.5566; Day 2 to Day 3: post hoc $t_{(14)}$ = 0.0, *P* > 0.9999) or random (Day 1 to Day 2: post hoc $t_{(14)}$ = 0.284, *P* > 0.9999; Day 1 to Day 3: post hoc $t_{(14)}$ = 0.284, *P* > 0.9999; Day 2 to Day 3: post hoc $t_{(14)}$ = 0.0, *P* > 0.9999) errors in controls across days. **g, h**, Optogenetic inhibition of callosal PV terminals induces perseveration on Day 2 and Day 3 compared to no stimulation on Day 1 (Day 1 to Day 2: post hoc $t_{(14)}$ = 4.092, *P* = 0.0033; Day 1 to Day 3: post hoc $t_{(14)}$ = 6.405, *P* < 0.0001; Day 2 to Day 3: post hoc $t_{(14)}$ = 2.313, *P* = 0.1094), but has no effect on random errors (Day 1 to Day 2: post hoc $t_{(14)}$ = 1.524, *P* = 0.4493; Day 1 to Day 3: post hoc $t_{(14)}$ = 0.254, *P* > 0.9999; Day 2

to Day 3: post hoc $t_{(14)}$ = 1.27, *P* = 0.6744). **i**, *PV-Cre* Ai14 mice had bilateral AAV-DIO-Ace2N-4AA-mNeon (Ace-mNeon) injections, an ipsilateral AAV-Synapsin-eNpHR-BFP (Syn-eNpHR) or AAV-Synapsin-mCherry (Syn-mCherry) injection and multimode fiber-optic implants in both prefrontal cortices. **j**, Representative images from mice injected with a control virus (Syn-mCherry), showing mCherry, Ace-mNeon, and tdTomato expression in the mPFC ipsi to the virus injection, and Ace-mNeon and tdTomato in the contra hemisphere. **k, n**, Experimental design: Day 1, no light delivery; Day 2, continuous light for inhibition during the R; Day 3, no light delivery. **l, m, o, p**, Optogenetic inhibition of callosal terminals does not change the number of perseverative errors in Syn-eNpHR mice (*n* = 5 mice, **l-m**) compared to Syn-mCherry controls (*n* = 4 mice, **l-m**; two-way ANOVA (task day × virus); interaction: $F_{(2,14)}$ = 1.933, *P* = 0.1814), and has no effect on random errors (two-way ANOVA (task day × virus); interaction: $F_{(2,14)}$ = 0.3789, *P* = 0.6914). **l, m**, Light delivery does not affect the number of perseverative (Day 1 to Day 2: post hoc $t_{(3)}$ = 1.464, *P* = 0.7183; Day 1 to Day 3: post hoc $t_{(3)}$ = 0.8783, *P* > 0.9999; Day 2 to Day 3: post hoc $t_{(3)}$ = 0.4804, *P* > 0.9999) or random (Day 1 to Day 2: post hoc $t_{(3)}$ = 1.732, *P* = 0.5451; Day 2 to Day 3: post hoc $t_{(3)}$ = 1.732, *P* = 0.5451) errors in controls across days. **o, p**, Nonspecific optogenetic inhibition of all callosal projections does not affect the number of perseverative (*n* = 5 mice; Day 1 to Day 2: post hoc $t_{(4)}$ = 0.6882, *P* > 0.9999; Day 1 to Day 3: post hoc $t_{(4)}$ = 2.108, *P* = 0.3081; Day 2 to Day 3: post hoc $t_{(4)}$ = 1.121, *P* = 0.9752) or random (Day 1 to Day 2: post hoc $t_{(4)}$ = 0.4082, *P* > 0.9999; Day 1 to Day 3: post hoc $t_{(4)}$ = 1.000, *P* > 0.9999; Day 2 to Day 3: post hoc $t_{(4)}$ = 0.000, *P* > 0.9999) errors in Syn-eNpHR mice across days. Two-way ANOVA followed by Bonferroni post hoc comparisons was used. ***P* < 0.01, *****P* < 0.0001; scale bars, 50 μm.

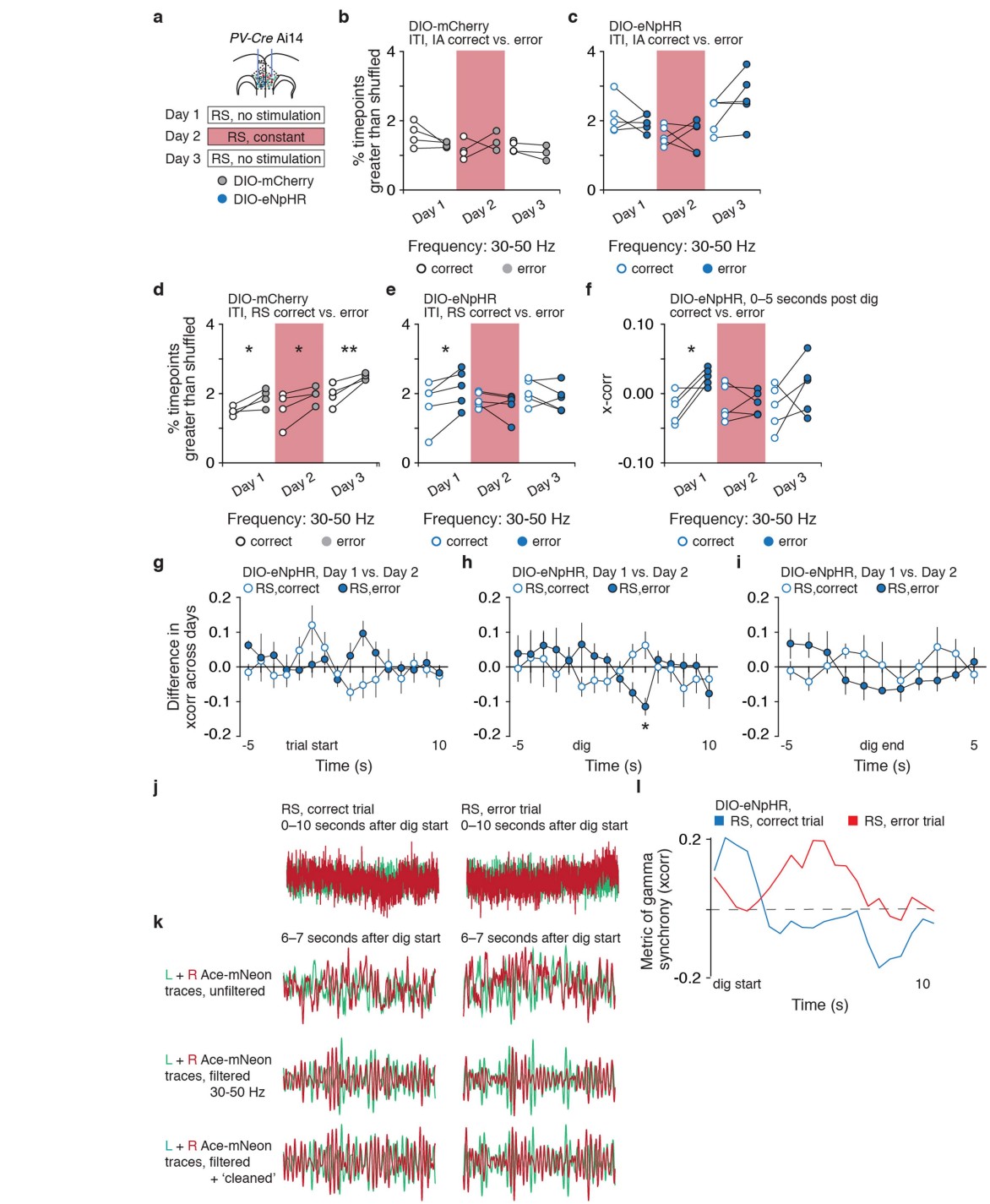

**Extended Data Fig. 9** | See next page for caption.

**Extended Data Fig. 9 | Additional analyses of gamma synchrony using dual-site voltage indicators. a**, Experimental design: Day 1, no optogenetic inhibition; Day 2, continuous light for optogenetic inhibition during the rule shift (RS); Day 3, no optogenetic inhibition (light for TEMPO was delivered on all days). **b**–**e**, Gamma synchrony during the intertrial interval (ITI). **b**–**c**, Gamma synchrony during ITIs does not change following correct or error trials during the initial association (IA) in either control ($n = 4$ mice; two-way ANOVA (trial type × task day); interaction: $F_{(1,7)} = 0.1143$, $P = 0.2111$) or DIO-eNpHR mice ($n = 5$ mice; two-way ANOVA (trial type × task day); interaction: $F_{(2,12)} = 1.903$, $P = 0.1915$) across days. **d**, Gamma synchrony during ITIs is higher after rule shift (RS) errors than after RS correct choices across days in control mice ($n = 4$ mice; Day 1: post hoc $t_{(9)} = 2.977$, $P = 0.047$; Day 2: post hoc $t_{(9)} = 2.969$, $P = 0.0472$; Day 3: post hoc $t_{(9)} = 4.039$, $P = 0.0088$). **e**, In DIO-eNpHR mice, gamma synchrony during ITIs is higher after RS errors than after RS correct choices on Day 1 ($n = 5$ mice; Day 1: post hoc $t_{(12)} = 2.914$, $P = 0.039$), but not on Day 2 when mice receive optogenetic inhibition of callosal PV+ terminals (post hoc $t_{(12)} = 1.041$, $P = 0.9545$) nor on Day 3 (post hoc $t_{(12)} = 1.153$, $P = 0.8136$). **f**, We performed a shorter timeframe re-analysis of TEMPO data collected from *PV-Cre* Ai14 injected in one mPFC with AAV-DIO-eNpHR. This was originally collected for Fig. 3. Gamma synchrony was calculated as the zero-phase lag cross-correlation ('x-corr') between 'corrected' mNeon signals that had been filtered in the 30–50 Hz as described in the Methods. Synchrony was averaged over the 5 s following correct choices or errors during the rule shift (RS). This measure of interhemispheric gamma synchrony is higher after RS errors than RS correct choices on Day 1, but was not different after errors vs. correct choices on Days 2 and 3 ($n = 5$ mice; two-way ANOVA; main effect of type of decision: $F_{(1,8)} = 5.349$,

$P = 0.0495$; Day 1: post hoc $t_{(24)} = 2.619$, $P = 0.0451$; Day 2: post hoc $t_{(24)} = 0.04492$, $P > 0.9999$; Day 3: post hoc $t_{(24)} = 1.763$, $P = 0.2717$). **g**–**i**, We also re-analyzed the TEMPO data collected from *PV-Cre* Ai14 mice injected in one mPFC with DIO-eNpHR to identify specific times when optogenetic inhibition of callosal PV+ projections disrupts gamma synchrony. We measured the change in gamma synchrony (calculated in 1 s windows for various time points relative to behavioral events) between Day 1 (control) and Day 2 (optogenetic inhibition) for both errors (filled circles) and correct choices (open circles) during the first 5 RS trials in *PV-Cre* mice expressing DIO-eNpHR. **g**, This change in gamma synchrony (change = Day 2 – Day 1) is not significantly different for error vs. correct trials around the time of trial start ($n = 5$ mice; two-way ANOVA (time point × error vs. correct); interaction: $F_{(15,64)} = 1.642$, $P = 0.0874$). **h**, This change in gamma synchrony is more negative for error vs. correct trials around the start of digging ($n = 5$ mice; two-way ANOVA (time point × error vs. correct); interaction: $F_{(15,64)} = 1.931$, $P = 0.0360$). **i**, This change in gamma synchrony is more negative for error vs. correct trials around the end of digging ($n = 5$ mice; two-way ANOVA (time point × error vs. correct); interaction: $F_{(10,44)} = 2.096$, $P = 0.0454$). **j**, Example traces of left (L) and right (R) Ace2n-4AA-mNeon (Ace-mNeon) traces (green and red, respectively), from 0–10 s after dig start on RS correct and error trials in a DIO-eNpHR-expressing mouse and **k**, zoomed in on the period 6–7 s after dig start. **l**, The time course of gamma synchrony (correlation values calculated from filtered and cleaned Ace-mNeon signals) from 0–10 s after dig start for these two example trials. Two-way ANOVA followed by Bonferroni post hoc comparisons was used. Data were expressed as mean ± s.e.m. *$P < 0.05$, **$P < 0.01$.

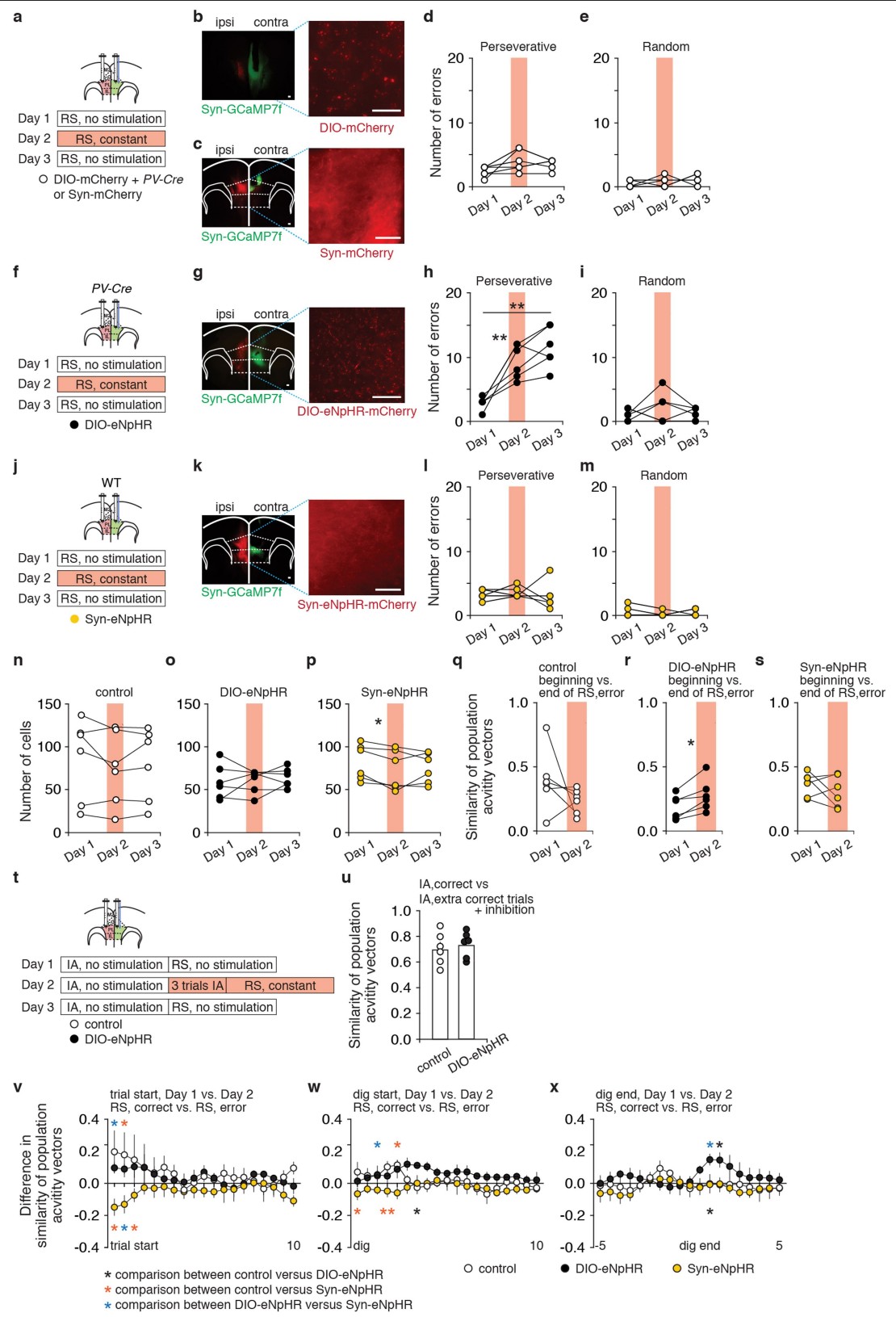

**Extended Data Fig. 10 |** See next page for caption.

**Extended Data Fig. 10 | Optogenetic inhibition of callosal terminals, delivered during concomitant calcium imaging, increases errors during rule shifts (RS).** Finer timescale analysis of how optogenetic inhibition affects changes in activity patterns during RS. **a**, **f**, **j**, *PV-Cre* or WT mice had AAV-DIO-eNpHR-mCherry (DIO-eNpHR) or AAV-Syn-eNpHR-mCherry (Syn-eNpHR) virus or a control virus (AAV-DIO-mCherry in *PV-Cre* mice or AAV-Synapsin-mCherry) injected in one mPFC and AAV-Synapsin-GCaMP7f (Syn-GCaMP7f) injected in the contralateral hemisphere. A GRIN lens, connected to a microendoscope, was implanted in contralateral to the site of eNpHR / control virus injection. Experimental design: Day 1, no light delivery for optogenetic inhibition; Day 2, continuous light for inhibition during the rule shift (RS); Day 3, no light delivery for optogenetic inhibition (light was delivered for calcium imaging on all days). **b**, Left: Representative image showing AAV-DIO-mCherry (DIO-mCherry) injected in the ipsilateral mPFC (ipsi) and Syn-GCaMP7f in the contralateral hemisphere (contra). Right: DIO-mCherry expression in callosal PV axonal fibers. **c**, Left: Representative image showing AAV-Syn-mCherry (Syn-mCherry) injected in the ipsilateral mPFC (ipsi) and Syn-GCaMP7f in the contralateral hemisphere (contra). Right: Syn-mCherry expression in callosal axonal fibers. **d**, **e**, **h**, **i**, **l**, **m**, Optogenetic inhibition of callosal PV terminals increases perseverative errors in DIO-eNpHR-expressing mice ($n$ = 6 mice, panel **h**) compared to controls ($n$ = 6 mice, panel **d**) and Syn-eNpHR-expressing mice ($n$ = 6 mice, panel **l**) (two-way ANOVA (task day × virus); interaction: $F_{(4,30)}$ = 11.74, $P$ < 0.0001). However there is no effect on random errors (compare panels **e**, **i**, **m**) (two-way ANOVA (task day × virus); interaction: $F_{(4,30)}$ = 1.377, $P$ = 0.2654). **d**, Light delivery does not affect the number of perseverative errors in control mice across days (Day 1 to Day 2: post hoc $q_{(5)}$ = 2.936, $P$ = 0.1896; Day 1 to Day 3: post hoc $q_{(5)}$ = 3.457, $P$ = 0.1235; Day 2 to Day 3: post hoc $q_{(5)}$ = 1.257, $P$ = 0.6700). **e**, Similar to **d**, there is no change in random errors in control mice ($n$ = 6 mice; Day 1 to Day 2: post hoc $q_{(5)}$ = 1.118, $P$ = 0.7245; Day 1 to Day 3: post hoc $q_{(5)}$ = 2.828, $P$ = 0.2072; Day 2 to Day 3: post hoc $q_{(5)}$ = 0.7670, $P$ = 0.8547). **g**, Left: Representative image showing AAV-DIO-eNpHR-mCherry (DIO-eNpHR) injected in the ipsi mPFC and Syn-GCaMP7f in the contra hemisphere. Right: DIO-eNpHR-mCherry expression in callosal PV axonal fibers. **h**, Optogenetic inhibition of callosal PV terminals increases the number of perseverative errors on Day 2 and Day 3 compared to no stimulation on Day 1 (Day 1 to Day 2: post hoc $q_{(5)}$ = 7.156, $P$ = 0.0090; Day 1 to Day 3: post hoc $q_{(5)}$ = 8.181, $P$ = 0.0050; Day 2 to Day 3: post hoc $q_{(5)}$ = 3.240, $P$ = 0.1476). **i**, Optogenetic inhibition of callosal PV terminals has no effect on random errors ($n$ = 6 mice; Day 1 to Day 2: post hoc $q_{(5)}$ = 1.225, $P$ = 0.6825; Day 1 to Day 3: post hoc $q_{(5)}$ = 0.3147, $P$ = 0.9732; Day 2 to Day 3: post hoc $q_{(5)}$ = 1.414, $P$ = 0.6083). **k**, Left: Representative image showing AAV-Synapsin-eNpHR-mCherry (Syn-eNpHR) injected in the ipsi mPFC and Syn-GCaMP7f in the contra hemisphere. Right: Syn-eNpHR-mCherry expression in callosal axonal fibers. **l**, In Syn-eNpHR mice, optogenetic inhibition of all callosal terminals does not affect the number of perseverative errors across days (Day 1 to Day 2: post hoc $q_{(5)}$ = 2.739, $P$ = 0.2231; Day 1 to Day 3: post hoc $q_{(5)}$ = 0.4939, $P$ = 0.9358; Day 2 to Day 3: post hoc $q_{(5)}$ = 0.8687, $P$ = 0.8190). **m**, Similar to **l**, in Syn-eNpHR mice, optogenetic inhibition of all callosal terminals does not affect random errors ($n$ = 6 mice; Day 1 to Day 2: post hoc $q_{(5)}$ = 2.236, $P$ = 0.3350; Day 1 to Day 3: post hoc $q_{(5)}$ = 1.257, $P$ = 0.6700; Day 2 to Day 3: post hoc $q_{(5)}$ = 1.651, $P$ = 0.5192). **n**, The number of cells in control (DIO-mCherry and Syn-mCherry) mice did not differ across days ($n$ = 6 mice; one-way ANOVA; $P$ = 0.3100; Day 1 to Day 2: post hoc $q_{(5)}$ = 2.068, $P$ = 0.3818; Day 1 to Day 3: post hoc $q_{(5)}$ = 1.155, $P$ = 0.7101; Day 2 to Day 3: post hoc $q_{(5)}$ = 1.421, $P$ = 0.6057).

**o**, The number of cells in DIO-eNpHR-expressing mice did not differ across days ($n$ = 6 mice; one-way ANOVA; $P$ = 0.7050; Day 1 to Day 2: post hoc $q_{(5)}$ = 0.1767, $P$ = 0.9914; Day 1 to Day 3: post hoc $q_{(5)}$ = 0.7532, $P$ = 0.8594; Day 2 to Day 3: post hoc $q_{(5)}$ = 1.131, $P$ = 0.7192). **p**, The number of cells in Syn-eNpHR mice decreased slightly from Day 1 (no optogenetic inhibition) to Day 2 (optogenetic inhibition of all callosal projections) ($n$ = 6 mice; one-way ANOVA; $P$ = 0.0921; Day 1 to Day 2: post hoc $q_{(5)}$ = 4.836, $P$ = 0.0419; Day 1 to Day 3: post hoc $q_{(5)}$ = 4.254, $P$ = 0.0653; Day 2 to Day 3: post hoc $q_{(5)}$ = 0.8008, $P$ = 0.8431). **q–s**, To quantify how activity evolves over the course of each RS error trial, we measured the similarity between population activity vectors (computed using a 2.5 s window) occurring at the beginning of the 10 s period following the start of digging, and those at the end of this period on the same trial, for each mouse on Day 1 versus Day 2. **q**, The similarity of activity patterns from the beginning to end of the post-dig period on RS error trials does not change from Day 1 to 2 in controls (two-tailed paired $t$-test, $t_{(5)}$ = 1.332, $P$ = 0.2403; $n$ = 6); **r**, however this similarity increases from Day 1 to 2 for DIO-eNpHR mice (two-tailed paired $t$-test, $t_{(5)}$ = 3.921, $P$ = 0.0112; $n$ = 6 mice); **s**, this similarity does not differ from Day 1 to 2 for Syn-eNpHR mice (two-tailed paired $t$-test, $t_{(5)}$ = 0.9724, $P$ = 0.3755; $n$ = 6 mice). **t**, Optogenetic inhibition of callosal PV terminals does not affect the similarity between population activity vectors measured during correct IA trials preceding the rule shift. Mice had AAV-DIO-eNpHR-mCherry (DIO-eNpHR in *PV-Cre* mice) or a control virus (AAV-DIO-mCherry in *PV-Cre* mice or AAV-Syn-mCherry in WT mice) injected into one mPFC, and AAV-Synapsin-GCaMP7f injected and a GRIN lens implanted in the contralateral hemisphere. Experimental design: Day 1, no light for optogenetic inhibition; Day 2, continuous light for optogenetic inhibition was delivered during the rule shift (RS); Day 3, no light delivery for optogenetic inhibition (light was delivered for calcium imaging on all days). **u**, On Day 2, after mice reach the learning criterion for the IA, we began delivering light for optogenetic inhibition and performed three additional IA trials, followed by the rule shift. Optogenetic inhibition of callosal PV terminals did not affect the similarity between activity vectors recorded after correct choices during these three additional IA trials, and those recorded during the preceding five IA trials ($n$ = 6 mice in each group; two-tailed, unpaired $t$-test, $t_{(10)}$ = 0.4958, $P$ = 0.6307). Thus, the effect of inhibiting callosal PV terminals to suppress changes in activity does not occur prior to the rule shift. **v–x**, The change from Day 1 to 2 (change = Day 2 value - Day 1 value) in the similarity of population activity vectors between correct trials and errors during the first 5 RS trials calculated using 1 s windows, for various time points relative to trial start (**v**), dig start (**w**), or dig end (**x**). **v**, There is a main effect of virus (two-way ANOVA; $F_{(2,285)}$ = 16.95; $P$ < 0.0001) between groups in the change in the population activity vector similarity from Day 1 to Day 2 around the time of trial start ($n$ = 6 mice in each group). Tukey's post hoc tests reveal statistically significant differences at specific time points. **w**, There is a difference between groups (main effect of virus) for the change in population vector similarities following the start of digging ($n$ = 6 mice in each group; two-way ANOVA; $F_{(2,285)}$ = 23.86, $P$ < 0.0001). Tukey's post hoc tests reveal statistically significant differences at specific time points. **x**, There is a difference between groups ($n$ = 6 mice in each group; main effect of virus, two-way ANOVA; $F_{(2,285)}$ = 11.63, $P$ < 0.0001) for the change in population vector similarities around the end of digging. Tukey's post hoc tests reveal statistically significant differences at specific time points. Two-way ANOVA followed by Tukey post hoc comparisons was used unless other noted. Data were expressed as mean ± s.e.m. *$P$ < 0.05, **$P$ < 0.01; scale bars, 100 μm and 50 μm, respectively.

# nature research

| | |
|---|---|

# Reporting Summary

Nature Research wishes to improve the reproducibility of the work that we publish. This form provides structure for consistency and transparency in reporting. For further information on Nature Research policies, see Authors & Referees and the Editorial Policy Checklist.

## Statistics

For all statistical analyses, confirm that the following items are present in the figure legend, table legend, main text, or Methods section.

| n/a | Confirmed | |
|---|---|---|
| ☐ | ☒ | The exact sample size (*n*) for each experimental group/condition, given as a discrete number and unit of measurement |
| ☐ | ☒ | A statement on whether measurements were taken from distinct samples or whether the same sample was measured repeatedly |
| ☐ | ☒ | The statistical test(s) used AND whether they are one- or two-sided *Only common tests should be described solely by name; describe more complex techniques in the Methods section.* |
| ☐ | ☒ | A description of all covariates tested |
| ☐ | ☒ | A description of any assumptions or corrections, such as tests of normality and adjustment for multiple comparisons |
| ☐ | ☒ | A full description of the statistical parameters including central tendency (e.g. means) or other basic estimates (e.g. regression coefficient) AND variation (e.g. standard deviation) or associated estimates of uncertainty (e.g. confidence intervals) |
| ☐ | ☒ | For null hypothesis testing, the test statistic (e.g. *F*, *t*, *r*) with confidence intervals, effect sizes, degrees of freedom and *P* value noted *Give P values as exact values whenever suitable.* |
| ☒ | ☐ | For Bayesian analysis, information on the choice of priors and Markov chain Monte Carlo settings |
| ☒ | ☐ | For hierarchical and complex designs, identification of the appropriate level for tests and full reporting of outcomes |
| ☒ | ☐ | Estimates of effect sizes (e.g. Cohen's *d*, Pearson's *r*), indicating how they were calculated |

*Our web collection on statistics for biologists contains articles on many of the points above.*

## Software and code

Policy information about availability of computer code

| Data collection | Synapse 84 (Tucker-Davis Technologies), nVoke2 (Inscopix, Inc.), ANY-maze (Stoelting Europe), pClamp (Molecular Devices). |
|---|---|
| Data analysis | Statistical analyses were performed in Matlab (Mathworks) and Prism 8 (GraphPad). OpenScope (Tucker-Davis Technologies) was used for behavioral scoring. Image processing was performed using ImageJ. Adobe Illustrator was used for assembling figures. |

For manuscripts utilizing custom algorithms or software that are central to the research but not yet described in published literature, software must be made available to editors/reviewers. We strongly encourage code deposition in a community repository (e.g. GitHub). See the Nature Research guidelines for submitting code & software for further information.

## Data

Policy information about availability of data

All manuscripts must include a data availability statement. This statement should provide the following information, where applicable:
- Accession codes, unique identifiers, or web links for publicly available datasets
- A list of figures that have associated raw data
- A description of any restrictions on data availability

The data that support the findings of this study are available from the corresponding author upon reasonable request. The underlying physiological data (trial-by-trial measurements of synchrony and population activity vectors) and associated MATLAB code are available on Zenodo at https://doi.org/10.5281/zenodo.7709805.

# Field-specific reporting

Please select the one below that is the best fit for your research. If you are not sure, read the appropriate sections before making your selection.

☒ Life sciences ☐ Behavioural & social sciences ☐ Ecological, evolutionary & environmental sciences

For a reference copy of the document with all sections, see nature.com/documents/nr-reporting-summary-flat.pdf

# Life sciences study design

All studies must disclose on these points even when the disclosure is negative.

| | |
|---|---|
| Sample size | No statistical methods were used to predetermine sample size, but sample size choice was based on previous studies (Cho et al., 2015) and are consistent with those generally employed in the field. |
| Data exclusions | In all behavioral experiments (slice electrophysiology, optogenetics, dual-site voltage-indicator photometry, in vivo calcium imaging), we verified that viral expression and/or fiber tip placement was in the target structure. No data was excluded. |
| Replication | All key findings were replicated in new cohorts, and all attempts at replication were successful. |
| Randomization | Animals were randomly assigned numbers and tested blind for the experimental condition. |
| Blinding | All behavioral experiments were performed, scored, and analyzed blind to the virus injected. |

# Reporting for specific materials, systems and methods

We require information from authors about some types of materials, experimental systems and methods used in many studies. Here, indicate whether each material, system or method listed is relevant to your study. If you are not sure if a list item applies to your research, read the appropriate section before selecting a response.

### Materials & experimental systems

| n/a | Involved in the study |
|---|---|
| ☒ | ☐ Antibodies |
| ☒ | ☐ Eukaryotic cell lines |
| ☒ | ☐ Palaeontology |
| ☐ | ☒ Animals and other organisms |
| ☒ | ☐ Human research participants |
| ☒ | ☐ Clinical data |

### Methods

| n/a | Involved in the study |
|---|---|
| ☒ | ☐ ChIP-seq |
| ☒ | ☐ Flow cytometry |
| ☒ | ☐ MRI-based neuroimaging |

## Animals and other organisms

Policy information about studies involving animals; ARRIVE guidelines recommended for reporting animal research

| | |
|---|---|
| Laboratory animals | Male and female mice of the following C57/Bl6 strains were used, at 10-25 weeks of age: PV-Cre, Ai14 (all from The Jackson Laboratory). |
| Wild animals | The study did not involve wild animals. |
| Field-collected samples | The study did not involve samples collected from the field. |
| Ethics oversight | All animal care, procedures, and experiments were conducted in accordance with the NIH guidelines and approved by the Administrative Panels on Laboratory Animal Care at the University of California, San Francisco as well as followed French and European guidelines for animal experimentation and in compliance with the institutional animal welfare guidelines of the Paris Brain Institute. |

Note that full information on the approval of the study protocol must also be provided in the manuscript.

