## [Peer Review File · Nature]

Manuscript Title: Long-range inhibition synchronizes and updates prefrontal task activity

Redactions – unpublished data

Reviewer Comments & Author Rebuttals

Reviewer Reports on the Initial Version:

Referees' comments:

Referee #1 (Remarks to the Author):

Summary: Cho, Shi, & Sohal study the function of an unusual population of neurons, colossal-projecting mPFC PV+ interneurons (INs). They focus on the contribution of these neurons to rule shifting behavior in a compound discrimination task previously tied to mPFC function in rats and mice.

The study builds off of several previous findings: 1) that gamma frequency synchrony across PV+ INs of both hemispheres increases after error trials in the lab's version of the rule-shifting task, and 2) that genetic disruption of inhibition through DLX5/6 gene affects performance in this same task.

The authors first characterize this interesting population of neurons and show they project across the corpus callosum to the contralateral mPFC and preferentially target PT-type neurons.

Then authors then show that NpHR mediated optogenetic inhibition of PV+ INs terminals in the hemisphere contralateral to labeling does three things:

- 1) inhibition of PV+ INs terminals in the hemisphere contralateral to labeling increases trials to criterion in the shifting phase, when inhibition all colossal terminals to the contralateral mPFC does not have any effect on trials to criterion in this phase.
- 2) inhibition of PV+ INs terminals in the hemisphere contralateral to labeling decreases expected gamma synchrony after error trials, and
- 3) inhibition of PV+ INs terminals in the hemisphere contralateral to labeling is associated with an increase in similarity of mPFC pyramidal neuron activity patterns after correct vs. error trials. This loss of distinctiveness is postulated to be a signature of failure of the colossal projecting interneuron function to altering network state to achieve shifting.

A curious finding is that these 3 effects persist 1 day after the stimulation when the behavior is re-run (Day 3).

Overall impact

The cell population under study is novel and exciting these data will be of great interest to the neurobiology field generally. The clinical inspiration is also significant. The data will be informative to schizophrenia theory and research.

My concerns focus on the source and meaning of the long lasting results and what the specifics of the single task and its highly curated data reveals and conceals.

Major issues

Shifting Behavior and its components:

Very little data is provided about the mouse behavioral performance, especially in main figures. IA data should be shown for all phases to establish if it is stable. For RS, is unclear what is driving the additional trials to criterion. Also, is there a change in time to choice? Exploration and motor behavior in the cage?

The narrow focus on shifting after compound discrimination learning (IA) in this highly honed/efficient homecage version of the task has both pluses and minuses.

The literature has several papers that dissociate the mPFC and OFC roles in reversal and shifting therefore it is appropriate to study shifting when studying the role of the mPFC. However, it would be helpful to know if motor aspects of this behavior and alternate, but PFC related behaviors, are affected or intact. Perhaps the left right coordination of hemispheres has effects on motor aspects of the task and left vs right orienting. Controls that would help isolate the exact function of the colossal PV terminal inhibition include rotations and turning, compound vs single stimulus discrimination learning, and reversal learning. It is also of interest to know if these PV colossal projections are also made between the OFC regions and playing a role in reversal learning.

There is also possibility the task taxes working memory as well as shifting. The supplemental info reveals that some part of the post error trial is in a holding cage for "about" 2 minutes. This way to run the task is unusual and taxes working memory more heavily than other more standard methods of running the task. It is also a heavy punishment which likely drives the very low ~10 trials to criteria in controls in rule shift. Thus, the task as run here depends on working memory of ~2 minutes and sensitivity to this waiting time. So, we might also want to know if the effect of terminal stimulation wanes if this additional tax on working memory is lessened by reducing the holding cage time. If this affects performance, then it may be a working memory effect (outcome or choice could be forgotten in stim condition). Alternatively, the controls may show more trials to criterion without this heavy punishment and produce more stochastic choices, again calling into question the reason behind the difference in trials to criterion (possibly an explore exploit tradeoff).

The methods also state that "After the task was done for the day, the bowls were filled with different odor-medium combinations and food was evenly distributed among these bowls and given to the mouse so that the mouse would disregard any associations made earlier in the day." The concept of "disregard" is not fleshed out and other possibilities/effects (consolidation/reconsolidation) should be considered.

The three days of training cannot be considered identical. The task may be solved using a different cognitive model on day 2 and then 3 than on day 1 (particularly after learning more about the long holding time). The lab has honed this task to a level that shifting takes only ~10 trials on all days, but this consistency does not prove that underlying cognitive model of the shift and the 2 minute punishment after errors is not changing/developing.

The digging media are 'unscented' commercially by label but also likely saliently represent 2 odors as well as 2 textured media. Same odor media with two textures (different particle size of same material or textured same material presented on reverse sides) are alternatives that test odor to texture/media shift, rather than odor set A to odor set B. It is unclear to me if the same odors were in all three days and if media (odor set B) was ever the first rewarded category for IA. Can day 2 or 3 also then be considered a compound reversal if the odors are the same?

Stimulation:

It is hard to understand the stimulation timing and duration protocol from the main text and figures. On the figure it says constant. How long did the task take? Should this be considered?

To some extent comparing the PV specific vs non specific terminal stimulation experiments satisfy control experiment requirements (for example for damage by some light). But longer trials to criterion means more light on Day 2. Can this be controlled for? Was a trial structure considered: light applied at time of choice instead.

Additionally, if light duration is ruled out as important for the Day 3 effect, the specific effect of PV terminal stimulation that causes lasting effects on day 3 suggests that this paradigm has powerful effects on plasticity and network function. Therefore it raises the question does terminal stimulation need to be done during behavior or is it sufficient to stimulate in the home cage for the same period of time (in the absence of a task) to produce a disruption of shifting one hour or one day later? If this works changes the interpretation of the manipulation to something more complex resulting from plasticity possibly downstream of the artificial manipulation.

Neural data:

The supplemental info reveals that some part of the post error trial is in a holding cage and also the post correct trial (after time to eat?), if I understand correctly. It is unclear if the neural data from post error and post correct trial are collected in the home/task cage or the holding cage and at the same delay from transfer. How is this transfer out and back in included and considered from the neural data? Is chasing the mouse part of this data and more common after an error? Do the results hold if this unusual capture and transfer of the mouse is not performed?

Minor Comments:

- The methods for running the behavior on three different days is not well explained in the main text. Are the same stimuli or different stimuli used?
- The methods for running the pretraining and IA phase of the behavior are also not well explained

in the main text.

- Figures always show within group change well, but never plot control and experimental groups on same plot.
- Fig 1: What layer(s) did the viral injection reach? What layer(s) are the images here from? The body text states contralateral terminals were labeled particularly in L5-6; the figure does not show any distribution across layers.
- Fig 1: How were PT vs. IT pyramidal neurons distinguished? (is the point of 1c to show that PT vs. IT have signature responses to depolarizing current? If so, that should be made explicit)
- Fig 3: body text states the injected control virus was tdTomato, but Fig 3a and figure legend report mCherry
- Normative language in writing: “proper function” “normal behavior” is inappropriate given the already artificial environment of the lab, inbred mice, and task. Typical or WT might suffice.

Referee #2 (Remarks to the Author):

In this manuscript, Kathleen et al., evaluate the role of prefrontal PV callosal projections on a extradimensional set shifting task. The authors used a combination of behavioral approaches, slices recording, voltage sensor and calcium imaging approaches. The authors first show using slice recordings that activation of PV projections onto the contralateral hemisphere induced inhibitory current in principal cells. Then, by specifically inhibiting this callosal projection they impaired rule shifting. Rule shifting impairment was associated with a decrease interhemispheric gamma synchrony during error trials and an increase in neuronal activity similarity. The authors concluded that callosal PV projections are key for maintaining and updating behavioral strategies.

The concept that specific inhibitory projection pathways are key for the selection of adapted behavioral strategies is very interesting and a promising area of research. The present manuscript attempted to make a strong causal link between a specific inhibitory callosal projection emanating from prefrontal parvalbumin-expressing neurons and extradimensional set shifting. Although I found the concept interesting, I do have some reserve related to the novelty of the results. Moreover, several key experiment and analyses are missing to claim that callosal PV projections are key for maintaining and updating behavioral strategies.

First of all, this manuscript is a follow up of a recent paper from the same group demonstrating that prefrontal gamma synchronization is key for extradimensional set shifting. Second, cortical inhibitory callosal projections are already known (Rock et al., 2017 Cerebral cortex) and gamma synchronization is known to depend on parvalbumin-expressing interneurons. Although this paper demonstrates that callosal PV projections are key in extradimensional set shifting, this represent to my point of view only an incremental advance in the field. Furthermore, the analyses used to demonstrate that the similarity of neuronal representations is increased following inhibition of the PV callosal pathway are preliminary and do not fully support the authors conclusions.

Figure 1: The anatomical and functional demonstration that PV-expressing cells project contralaterally and inhibit principal cells deserve additional analyses. First of all, the authors should

evaluate whether cells expressing eYFP are indeed PV-expressing cells using immunocytochemistry. The authors should also provide controls that the construct is not leaking in the contralateral side. Optogenetic manipulations on retrogradely labeled cells should also be performed. Second, an AHP over 3 mV is not an absolute criterion to define pyramidal tract cells, again retrograde staining should be done in addition. Quantification of the latency of IPSP should be provided (panel e, IPSP on the first stimulation pulse does not look time-locked).

Figure 2: I found overall that the behavioral data should be described in much more details. The fine timescale analysis of the behavioral data should be provided as it may represent a serious confound on the interpretation of the results. Before to be averaged, data should be provided by trial types in order to evaluate the dispersion of the data and potential bias in stimulus selection. Concerning the optogenetic manipulation, the authors must provide control experiments demonstrating that the manipulation does not impact motor behavior and analyze in details the animal behavior during the time course of the manipulation. Another concern is the design of the optogenetic stimulation, once IA criterion is reached (8 correct out of 10 consecutive trials), it is not clear why the authors first inhibited 3 additional IA trials before to inhibit RS trials. It is possible that the manipulation impacts the original association rather than the rule shifting aspect of the task. Also, the fact that the authors inhibited the entire RS sequence does not allow to evaluate if the animal would have succeeded the RS task on this particular trial type.

Figure 3: Gamma synchrony should also be demonstrated using classical LFP recording approaches and dynamically analyzed over the time course of the trials/stimulation. Raw data should be provided as well. The long-lasting effect of the manipulation on both behavior and gamma synchrony is puzzling and the authors should at least control that the callosal PV pathway is still functional on day 3 and that the stimulation does not induce apoptosis of this cell population.

Figure 4: This dataset is very preliminary. The evaluation of the similarity of neuronal activity patterns is done using a vector that is obtained by (1) averaging 10 second of activity after the animal decision and (2) additional trial averaging. The authors should reconsider their analyses as this one does not provide a good estimate of neuronal activity pattern. I suggest the authors to evaluate changes in neuronal activity pattern dynamically rather than statically and use machine learning algorithm to decode neuronal information. Changes in neuronal activity should also be clearly link to behavioral analyses that is a potential confound. Without a deep analysis of neuronal activity pattern with the appropriate controls, the data provided do not support the conclusions. An effort should also be made to provide raw data to see how neurons change their activity during the task.

Referee #3 (Remarks to the Author):

The manuscript of Cho et al builds upon previous studies, including two from the same authors, to address the mechanisms of cognitive flexibility in the mammalian cortex. This previous work established that rule shifts leads to cross-hemispheric gamma synchrony of PV+ neurons in the PFC and that rule shifts results in changes and/or reorganization of PFC activity. The experiments here

are described as an effort to connect this current knowledge at the levels of the circuit, network dynamics and neural representations. To this end, the authors employ optogenetic behavior, ex vivo physiology, in vivo cell-type specific voltage imaging and in vivo endoscopic calcium imaging to conclude that callosal projecting PV neurons are required for a switch from maintenance to updating of PFC neural representations in response to changes in the rules of the task. The paper is well written and includes some interesting findings, however in its current form I don't see a significant shift in our understanding compared to what has been described in the previous work.

Major points:

1) On the level of the circuit, the paper begins with anatomy to establish the postsynaptic targets of the contralaterally projecting PFC PV+ neurons. The data show that these axons synapse on pyramidal neurons, with a significant bias to neurons with a pyramidal tract electrophysiological signature, but stops there. The authors do not investigate input to GABAergic neurons in the contralateral hemisphere, nor do they offer additional evidence on the projection targets of the PV neurons they identify as the major recipient of the PV input; both these shortcomings limit the usefulness and scope of these data. In the discussion of their own previous *Nature Neuroscience* (2020) paper they highlight the importance of identifying the targets of the pyramidal neurons receiving cross-hemispheric input, but do not include those experiments. Combining the approach used here with the injection of retrograde labels in potential targets (ie. mediodorsal thalamus, nucleus accumbens, striatum) would have made these findings much more useful in interpreting how these circuits shift behavior via the communication between multiple regions.

2) On the level of neural representation- the calcium imaging data from the PFC is not described in sufficient detail, with the authors only showing measurements which captures the similarity based on angle between two population vectors while remaining agnostic to their magnitude. The raw data is not described or discussed- it is impossible to even determine how many units were observed, how many active units contribute to the vectors used for analysis, etc- these first order data must be included in extended data to appreciate the strength and validity of the similarity claims. Further, previous physiological recordings in the PFC (for example, Durstewitz et al *Neuron* 2010, Karlsson et al *Science* 2012, Schmitt et al *Nature* 2017 among others the authors cite) provided numerous examples of how the activity patterns could be analyzed to understand the link between population activity and behavioral shifts; employing similar approaches here would make these data much more useful to understanding the role of the PV projections in this reorganization process.

3) Regarding neural dynamics- one key unaddressed question is if it is crucial that the inhibition of the contralaterally projecting PV terminals occurs during the first instance of rule shift learning? In the authors' 2015 *Neuron* paper, bilateral gamma stimulation rescued RS behavior in the *Dlx5/6+/-* mice and this rescue improved behavior on subsequent days even in the absence of optogenetic stimulation, without the reappearance of gamma synchrony. In the 2020 Cho et al *Nature Neuroscience* paper out-of-phase stimulation impaired RS behavior, but this could be reinstated the next day with in-phase stimulation. In the current work the authors show that optogenetic inhibition of the PV terminals impairs behavior in the first rule shift session and leads to continued poor performance on subsequent days, suggesting what they term a novel form of network plasticity. This result needs to be understood and described more carefully. Can this deficit be rescued with gamma

stimulation as in the mutant mice? A key experiment would be to reverse the day 2 and day 3 protocol in figure 3; does successful RS learning on day 2 make this projection dispensable for later RS performance? Does this lead to another dissociation between performance and the increase in gamma synchrony following error trials, as seen in the *Dlx5/6* mice? Is the PV projections and gamma only required for the first instance of rule shift learning and then becomes dispensable?

4) Related to point 3- I am also confused on the interpretation of the data in extended figure 2i-p, the experiments with optogenetic inhibition of all callosal projections. This is an interesting finding, but as this manipulation should inhibit both PV and glutamatergic projections, shouldn't it also disrupt gamma synchrony across the hemispheres- this should be measured. If there is dissociation of initial RS learning and the error-triggered increase in gamma synchrony, do the authors believe a second mechanism is recruited under these conditions? Is the take-home message that just callosal excitatory input is worse for performance than input at all? Related to this is the conceptual question- why is bilateral synchrony required? The authors refer to communication between the hemispheres becoming "mistimed" when the contralateral PV projections are inhibited. I don't think they have done experiments to address this and should consider using terminology that better reflects their results. If it is about the timing of convergent input from the two hemispheres to a putative downstream target, then its possible asynchronous input could be more detrimental to performance than no input from one hemisphere at all- is this the authors working model? I agree that the error triggered synchrony is an interesting and compelling observation and may correlate with a shift in the PFC pyramidal cell activity patterns, but with the data included in the paper fall short of making the connections between the anatomy, the circuit dynamics and neural representations that the authors are striving for.

Minor points:

1) The labels on the y-axes of Extended Data Fig. 2 are confusing. Currently all read "trials to criterion" however what is being reported in these graphs is the number of errors of a given type; shouldn't the labels simply read "# of errors"?

Author Rebuttals to Initial Comments:

We appreciate the Reviewers' feedback that our findings are "**novel and exciting**," "**of great interest to the neurobiology field generally**" and particularly "**significant**" for schizophrenia research (Reviewer 1), "**very interesting and a promising area of research**" (Reviewer 2), and "**well written and includes some interesting findings**" (Reviewer 3). On the following pages, please find our detailed responses to the Reviewer's helpful feedback. As an overview, we have added the following new studies to the revision (all of which are consistent with and provide additional support for our previous findings and interpretations).

1. Inhibiting *all* callosal synapses elicits a transient, but not persistent, deficit in gamma synchrony (Fig. 3h–n).
2. Inhibiting *all* callosal synapses does not suppress the changes in population activity patterns that normally occur after rule shift errors (Fig. 4f,j,m). (Together with the previous experiment, this supports our model that the function of callosal PV+ projections and gamma synchrony is to prevent callosal communication from occurring in an aberrant manner that disrupts the normal reorganization of prefrontal network activity during rule shifts. In this model, callosal PV+ projections and interhemispheric gamma synchrony are not required when all callosal communication is blocked).
3. Selectively inhibiting callosal PV+ projections does not affect learning of intradimensional rule reversals (as opposed to extradimensional rule shifts), and does not cause long-lasting impairments in subsequent rule shift learning (Extended Data Fig. 6).
4. Using an intersectional (dual recombinase) strategy to specifically label callosally-projecting PV neurons in mPFC with both NpHR3.3 and ChR2, inhibiting and then stimulating these callosal PV+ terminals induces persistent deficits in rule shift performance and a long-lasting reversal of these deficits, respectively (Extended Data Fig. 7).
5. To control for potential leakage of virus from the injected mPFC into the contralateral hemisphere, we show that injecting diluted AAV-DIO-eNpHR into one mPFC and then delivering light to that hemisphere does not affect rule shift learning (Extended Data Fig. 2i–k).
6. Immunostaining confirms that the labeling of PV+ neurons is 96.7% specific (Extended Data Fig. 1a).
7. Slice electrophysiology experiments find that callosal PV+ projections innervate 100% of retrogradely-labeled pyramidal neurons that project to mediodorsal thalamus, compared to 0% of such neurons that project to the contralateral cortex or nucleus accumbens, a small fraction of pyramidal neurons that project to dorsal striatum, and 0% of fast-spiking neurons (Extended Data Fig. 1b–k).
8. We performed additional analyses of microendoscopic Ca²⁺ imaging data to examine changes in activity levels across days (Fig. 4o–q), and also found that machine learning / classifier-based approaches are not suitable for analyzing this type of data.
9. New (fast time-resolution) analyses of TEMPO data to show that gamma synchrony increases during the 5 sec immediately following a choice on rule shift error trials (Extended Data Fig. 9) (this is why we analyzed population vectors during the 10 sec following each choice – to examine changes in population activity that immediately follow this period of heightened synchrony).
10. New analyses of rule shift behavior that show that optogenetic inhibition does not cause any changes in gross motor behavior, side biases, etc. (Extended Data Fig. 4a–c).

Referee #1 (Remarks to the Author):

Summary: Cho, Shi, & Sohal study the function of an unusual population of neurons, colossal-projecting mPFC PV+ interneurons (INs). They focus on the contribution of these neurons to rule shifting behavior in a compound discrimination task previously tied to mPFC function in rats and mice.

The study builds off of several previous findings: 1) that gamma frequency synchrony across PV+ INs of both hemispheres increases after error trials in the lab's version of the rule-shifting task, and 2) that genetic disruption of inhibition through DLX5/6 gene affects performance in this same task.

The authors first characterize this interesting population of neurons and show they project across the corpus collosum to the contralateral mPFC and preferentially target PT-type neurons. Then authors then show that NpHR mediated optogenetic inhibition of PV+ INs terminals in the hemisphere contralateral to labeling does three things:

- 1) inhibition of PV+ INs terminals in the hemisphere contralateral to labeling increases trials to criterion in the shifting phase, when inhibition all colossal terminals to the contralateral mPFC does not have any effect on trials to criterion in this phase.*
- 2) inhibition of PV+ INs terminals in the hemisphere contralateral to labeling decreases expected gamma synchrony after error trials, and*
- 3) inhibition of PV+ INs terminals in the hemisphere contralateral to labeling is associated with an increase in similarity of mPFC pyramidal neuron activity patterns after correct vs. error trials. This loss of distinctiveness is postulated to be a signature of failure of the colossal projecting interneuron function to altering network state to achieve shifting.*

A curious finding is that these 3 effects persist 1 day after the stimulation when the behavior is re-run (Day 3).

Overall impact

The cell population under study is novel and exciting these data will be of great interest to the neurobiology field generally. The clinical inspiration is also significant. The data will be informative to schizophrenia theory and research.

My concerns focus on the source and meaning of the long lasting results and what the specifics of the single task and its highly curated data reveals and conceals.

Major issues

Shifting Behavior and its components:

Very little data is provided about the mouse behavioral performance, especially in main figures. IA data should be shown for all phases to establish if it is stable. For RS, is unclear what is driving the additional trials to criterion. Also, is there a change in time to choice? Exploration and motor behavior in the cage?

Our response: First, for every relevant behavioral experiment (related to Figures 2-4), we now show the corresponding IA data in Extended Data Fig. 3. We have performed statistical tests to confirm that IA performance is stable. For each of these datasets, we have used two-way ANOVA to confirm that there is no significant difference in the number of trials needed to learn the initial association on Day 1 vs. Day 2 between control vs. experimental cohorts (i.e., $p > 0.05$ for all task day \times virus interactions). We also show (by plotting the numbers of perseverative and random) errors that increases in the trials to criterion are associated mainly with an increase in perseverative errors (Extended Data Figs. 5, 7, 8, 10).

For behavioral datasets in which we used video recordings to perform time-stamped analyses (e.g., dual-site voltage indicator and calcium imaging experiments), we have computed the latency to dig (aka the 'time-to-choice'). This is shown in Extended Data Fig. 4c. Using two-way ANOVA, there is no significant change in the time-to-choice from the first 5 initial association trials to the first 5 rule shift trials across control and experimental virus cohorts.

Finally, a representative video showing mouse behavior before and after onset of optogenetic manipulation is now shown in Supplementary Video 1 (6969.mp4). This video starts with the mouse completing a correct trial during the initial association. Following transfer to the rest cage, the optogenetic stimulation begins for 3 additional initial association trials before the rule shift portion of the task begins. Light stimulation does not alter the performance or behavior of the mice during these 3 extra trials of initial association.

The narrow focus on shifting after compound discrimination learning (IA) in this highly honed/efficient homecage version of the task has both pluses and minuses.

The literature has several papers that dissociate the mPFC and OFC roles in reversal and shifting therefore it is appropriate to study shifting when studying the role of the mPFC. However, it would be helpful to know if motor aspects of this behavior and alternate, but PFC related behaviors, are affected or intact. Perhaps the left right coordination of hemispheres has effects on motor aspects of the task and left vs right orienting. Controls that would help isolate the exact function of the colossal PV terminal inhibition include rotations and turning, compound vs single stimulus discrimination learning, and reversal learning. It is also of interest to know if these PV colossal projections are also made between the OFC regions and playing a role in reversal learning.

Our response: We did not generally observe any rotations. To quantify possible effects on left vs. right orienting, turning, etc., we quantified the fraction of trials in which the mouse chose the bowl that was ipsilateral vs. contralateral to the side of optogenetic inhibition. This was specifically done during the 3 trials of initial association which occurred after the mouse reached the learning criterion and we began delivering optogenetic inhibition (but before the onset of the rule shift). These data are shown in Extended Data Fig. 4b. Using two-way ANOVA, there is no difference in the ipsi- vs. contralateral choice percentage made across control and experimental virus cohorts (side choice \times virus interaction: $F_{(1,67)} = 0.7813$, $P = 0.3799$).

We have now performed additional experiments which show that inhibiting callosal PV projections has no effect on the learning of intradimensional rule reversals (in a task which is otherwise identical to a rule shift) (Extended Data Fig. 6). This also shows that impairments in rule

shift learning caused by inhibiting callosal PV projections do not simply reflect changes in motor or orienting behaviors, as the overall mechanics of the rule reversal and rule shift tasks are identical.

There is also possibility the task taxes working memory as well as shifting. The supplemental info reveals that some part of the post error trial is in a holding cage for “about” 2 minutes. This way to run the task is unusual and taxes working memory more heavily than other more standard methods of running the task. It is also a heavy punishment which likely drives the very low ~10 trials to criteria in controls in rule shift. Thus, the task as run here depends on working memory of ~2 minutes and sensitivity to this waiting time. So, we might also want to know if the effect of terminal stimulation wanes if this additional tax on working memory is lessened by reducing the holding cage time. If this affects performance, then it may be a working memory effect (outcome or choice could be forgotten in stim condition). Alternatively, the controls may show more trials to criterion without this heavy punishment and produce more stochastic choices, again calling into question the reason behind the difference in trials to criterion (possibly an explore exploit tradeoff).

Our response: Consistent with the Reviewer’s intuition, when initially optimizing our version of the task (a little more than a decade ago), we found that the long duration of the ‘timeout’ following error trials was critical for mice to rapidly learn associations. Changing this duration could obviously open up a can of worms, as some mice would fail to learn the association and the variability of performance would increase dramatically, making it more difficult to discern changes in performance related to optogenetic manipulations. But the heart of the Reviewer’s question is whether the deleterious effects of inhibiting callosal synapses are related to the working memory load during this long intertrial interval (ITI), i.e., does inhibiting callosal PV projections prevent the PFC from maintaining representations and cause mice to forget previously-learned rules during this long ITI? Three sets of data address this concern. First, in the current manuscript we showed that inhibiting callosal PV projections affects neural representations immediately after the choice is made. Furthermore, this manipulation specifically increased the stability of these representations, i.e., caused patterns of activity after RS errors to become *more similar* to the patterns occurring after IA errors or after RS correct trials. This suggests that the function of callosal projections is not to maintain activity patterns during the ITI, but rather to *alter those patterns before the ITI*.

Second, we have older, unpublished data from experiments in which we optogenetically inhibited all prefrontal interneurons (labeled using *Dlx12b-Cre* mice). In previously published work, we found that inhibiting all prefrontal interneurons throughout both trials and ITIs prevents mice from learning rule shifts (but does not interfere with learning of initial associations or rule reversals). By contrast, in subsequent unpublished experiments, we found that selectively inhibiting prefrontal interneurons only during trials or only during ITIs has no effect on the learning of rule shifts. The fact that mice can learn rule shifts normally even when prefrontal interneurons are inhibited throughout the ITI does not square with the idea that these inhibitory networks play an important role in maintaining task information during the ITI. This earlier experiment is consistent with the interpretation that intact inhibitory networks can act, either during trial periods or during the ITI, to facilitate the divergence of neural representations from previously-established patterns. These data are now shown below in **Reviewer Figure 1**.

Redacted

Reviewer Figure 1: Inhibiting prefrontal interneurons selectively within or between trials does not affect rule shift learning. **a**, Experimental design: in one cohort, mice receive light only during (within) trials on Day 1. On Day 2 they receive constant light throughout the task. The other cohort receives light only between trials (i.e., during ITIs) on Day 1, and constant light on Day 2. **b**, Number of trials needed to reach the learning criterion for the rule shift on each day for each control cohort (which express eYFP but not Arch). **c**, Number of trials needed to reach the learning criterion for the rule shift on each day for each experimental cohort (which express Arch in prefrontal interneurons).

Third, as noted above, we found that inhibiting callosal PV projections does not affect rule reversals (Extended Data Fig. 6). This also runs counter to the idea that these projections serve to maintain task information during the ITI (and suggest that these projections facilitate changes in representations specifically when mice need to learn to use a new set of stimuli to make decisions).

The methods also state that “After the task was done for the day, the bowls were filled with different odor-medium combinations and food was evenly distributed among these bowls and given to the mouse so that the mouse would disregard any associations made earlier in the day.” The concept of “disregard” is not fleshed out and other possibilities/effects (consolidation/reconsolidation) should be considered.

Our response: The Reviewer is correct that we do not know that mice ‘disregard’ previously-learned associations. The point is simply that at the end of each task day, mice receive additional food as part of their food restriction. We presented them this food in the same bowls used for the task and in the presence of different odors and textures, however there was no longer an association between specific cues and food, rather food was present in all bowls. We have adjusted this text in the Methods on lines 954-957.

The three days of training cannot be considered identical. The task may be solved using a different cognitive model on day 2 and then 3 than on day 1 (particularly after learning more about the long holding time). The lab has honed this task to a level that shifting takes only ~10 trials on all days, but this consistency does not prove that underlying cognitive model of the shift and the 2 minute punishment after errors is not changing/developing.

Our response: In principle we agree. However, in practice, we found that inhibiting callosal PV projections elicited the identical effect (increased perseveration and more trials to criterion),

regardless of whether inhibition was delivered on Day 1 (Extended Data Fig. 7e–j) vs. Day 2 (Fig. 2f; Extended Data Fig. 5e–h). This indicates that the effects of optogenetically inhibiting callosal PV projections on behavior do not change from Day 1 to Day 2. Furthermore, the similarity between population activity vectors on different trial types does not change from Day 1 to Day 2 in control mice which do not undergo optogenetic inhibition (Figure 4). Thus, neither the behavior itself, nor the associated patterns of neural activity, nor the effects of optogenetic inhibition on behavior change from Day 1 to Day 2. Of course it is a scientific fallacy to ever presume to ‘know’ exactly what the mouse is thinking as it does this task, but we see no evidence for major changes in the basic circuit mechanisms underlying task performance between Days 1 and 2.

The digging media are ‘unscented’ commercially by label but also likely saliently represent 2 odors as well as 2 textured media. Same odor media with two textures (different particle size of same material or textured same material presented on reverse sides) are alternatives that test odor to texture/media shift, rather than odor set A to odor set B. It is unclear to me if the same odors were in all three days and if media(odor set B) was ever the first rewarded category for IA. Can day 2 or 3 also then be considered a compound reversal if the odors are the same?

Our response: We agree – the key point is that we have two distinct sets of cues. The two odors are always mutually exclusive (i.e., cannot be in the same bowl) and the two “textures” (which probably also have some scent) are also mutually exclusive. (Both textures and both odors are present in every trial). Furthermore, only one of these two sets predicts reward locations at a time, and within a set, both cues predict reward location equally. We do not want to dive too deeply into the ‘meaning’ of the task or exactly what terms should be used to describe it as these are all anthropomorphic concepts. Rather, we would emphasize this is a highly naturalistic task – using cues to define the location of food rewards – one which clearly involves learning and perseveration, and which depends on the PFC.

We did use the same cues across days. This is now explicitly stated in the Methods (lines 956-957). We did vary (in an approximately counterbalanced fashion) between using an odor or texture for the rewarded cue in the initial association. Again, there is a theoretical question about whether there are lingering effects of the previously learned associations on behavior on Days 2 and 3, but we see no evidence of this in terms of the behavior, associated neural activity, or effects of optogenetic manipulations on behavior.

Stimulation:

It is hard to understand the stimulation timing and duration protocol from the main text and figures. On the figure it says constant. How long did the task take? Should this be considered?

Our response: The Reviewer seems to be asking whether the effects can be attributed to the prolonged period of stimulation. This does not seem to be the case as the perseveration occurred immediately during the rule shift. Specifically, to investigate the acute effects of the optogenetic stimulation during the rule shift portion of the task, we calculated the percentage of perseverative errors separately for the first 5 rule shift trials vs. the next 5 rule shift trials, across control and experimental virus cohorts and across Day 1 vs. Day 2. Using two-way ANOVA, followed by Bonferroni post hoc comparisons, optogenetic inhibition of callosal mPFC PV terminals causes mice to perseverate more during both the first 5 rule shift trials and the next 5 rule shift trials on Day 2

compared to the no optogenetic inhibition condition on Day 1. By contrast, there was no change from Day 1 to Day 2 in the percentage of perseverative choices within either the first 5 or next 5 rule shift trials in control (eNpHR-negative) mice. We have plotted this in Extended Data Fig. 4d–e.

To some extent comparing the PV specific vs non specific terminal stimulation experiments satisfy control experiment requirements (for example for damage by some light). But longer trials to criterion means more light on Day 2. Can this be controlled for? Was a trial structure considered: light applied at time of choice instead.

Our response: First, in addition to the controls for light duration the reviewer notes, all experiments contain controls which receive light but lack halorhodopsin. And the changes in behavior appear immediately (during the first 5 trials of the rule shift). So the effects cannot be attributed to the duration of light delivery. Second, with respect to the specific experiment suggested by the Reviewer – as described above, we tried something similar previously (unpublished experiments performed several years ago), but found that optogenetic inhibition of prefrontal interneurons during only the trial (or during only the ITI) period was insufficient to disrupt rule shift learning. This is shown above in Reviewer Figure 1.

Additionally, if light duration is ruled out as important for the Day 3 effect, the specific effect of PV terminal stimulation that causes lasting effects on day 3 suggests that this paradigm has powerful effects on plasticity and network function. Therefore it raises the question does terminal stimulation need to be done during behavior or is it sufficient to stimulate in the home cage for the same period of time (in the absence of a task) to produce a disruption of shifting one hour or one day later? If this works changes the interpretation of the manipulation to something more complex resulting from plasticity possibly downstream of the artificial manipulation.

Our response: To determine if optogenetic inhibition needs to be delivered specifically during the rule shift task in order to elicit lasting effects, we examined rule shift behavior in mice which had previously received optogenetic inhibition of callosal PV projections while performing an intradimensional *rule reversal* (instead of an extradimensional rule shift) on Day 1. On Day 2, these mice learned an initial association and rule shift in the absence of any additional optogenetic inhibition. In these mice, rule shift performance was normal, i.e., not different from control (opsin-negative) mice (Extended Data Fig. 6). This confirms that terminal inhibition needs to be done specifically during a rule shift to produce a long-lasting deleterious effect.

Neural data:

The supplemental info reveals that some part of the post error trial is in a holding cage and also the post correct trial (after time to eat?), if I understand correctly. It is unclear if the neural data from post error and post correct trial are collected in the home/task cage or the holding cage and at the same delay from transfer. How is this transfer out and back in included and considered from the neural data? Is chasing the mouse part of this data and more common after an error? Do the results hold if this unusual capture and transfer of the mouse is not performed?

Our response: Our analysis was done during the first 10 seconds following the choice, which

explicitly excludes the period when the mouse was transferred to the holding cage. We have revised the text on page 44 to state this more clearly (lines 984-985).

Minor Comments:

- *The methods for running the behavior on three different days is not well explained in the main text. Are the same stimuli or different stimuli used?*

Our response: The same stimuli are used across days, but the cue that is associated with food reward changes. We have added a sentence stating this on page 44 of the revised manuscript.

- *The methods for running the pretraining and IA phase of the behavior are also not well explained in the main text.*

Our response: On the day before the first task day, mice undergo habituation as described in Cho et al., Neuron, 2015. We have added that description to the Methods on lines 948-952: “After mice reached their target weight, they underwent one day of habituation. On this day, mice were given ten consecutive trials with the baited food bowl to ascertain that they could reliably dig and that only one bowl contained food reward. All mice were able to dig for the reward, and started the task the next day.”

The procedures for the IA phase are identical to the rule shift (except that a different cue is associated with food reward).

- *Figures always show within group change well, but never plot control and experimental groups on same plot.*

Our response: We have chosen to show the within-group change in single plots, and to plot experimental and control groups in separate plots. We believe that this is the most clear way to show the experimental effect, i.e., the change within the experimental group. It is then straightforward to look at the adjacent plot for the control group to confirm that any across-day change observed for the experimental group is not present for the control group. We do it this way because combining the experimental and control groups on the same plot intermingles the individual data points and the lines connecting them, making the specific trends within each group considerably more difficult to discern.

- *Fig 1: What layer(s) did the viral injection reach? What layer(s) are the images here from? The body text states contralateral terminals were labeled particularly in L5-6; the figure does not show any distribution across layers.*

Our response: The right panel of Figure 1a now includes labels for different layers to make this distribution more clear.

- *Fig 1: How were PT vs. IT pyramidal neurons distinguished? (is the point of 1c to show that PT vs. IT have signature responses to depolarizing current? If so, that should be made explicit)*

Our response: We and others have previously shown that PT vs. IT neurons can be distinguished on the basis of their responses to hyperpolarizing current (Gee et al., *J Neurosci*, 2012). However for the revision, we completed new experiments, shown in Extended Data Fig. 1, which recorded from mPFC projection neurons that had been retrogradely labeled by CTb injections into the contralateral mPFC, mediodorsal thalamus (MD), dorsal striatum or nucleus accumbens (NAc). These reveal that callosal PV+ projections exhibit a high level of specificity in their connectivity, e.g., 100% of MD-projecting neurons receive callosal PV+ input, whereas we did not observe any callosal PV+ connections onto callosal or nucleus accumbens-projecting neurons. We have therefore removed references to PT vs. IT neurons and focused instead on these retrogradely-labeled classes of projection neurons (which are more specific than the broad PT / IT classes).

- *Fig 3: body text states the injected control virus was tdTomato, but Fig 3a and figure legend report mCherry*

Our response: Thank you, we have corrected the text and figure legend. We apologize if this was confusing or unclear – mCherry is a control fluorophore (expressed instead of opsins), whereas tdTomato is expressed in all animals and used to provide a non-voltage dependent reference signal for TEMPO analyses.

- *Normative language in writing: “proper function” “normal behavior” is inappropriate given the already artificial environment of the lab, inbred mice, and task. Typical or WT might suffice.*

Our response: We distinguish several terms here: ‘natural,’ ‘normal,’ ‘typical,’ and ‘wild-type.’ Natural refers to behavior that occurs in the natural environment. Typical refers to the statistical properties of the behavior, i.e., it falls within the typical range. Wild-type refers specifically to genetic status of the mice – because all of our mice are transgenic (*PV-Cre*), some reviewers object to the use of ‘wild-type’ to describe them. In contrast to these terms, we specifically use ‘normal’ to refer to behavior that occurs in the absence of experimental perturbations such as optogenetic inhibition. We realize that the Referee considers the task itself to be an experimental perturbation. However, for the reasons laid out we still find ‘normal’ closer to what we are trying to convey than ‘typical’ or ‘wild-type.’ Therefore, we have revised the text by removing several instances of ‘normal,’ and making the context for the remaining uses of normal more clear, e.g., ‘during normal rule shift learning.’

Referee #2 (Remarks to the Author):

In this manuscript, Kathleen et al., evaluate the role of prefrontal PV callosal projections on a extradimensional set shifting task. The authors used a combination of behavioral approaches, slices recording, voltage sensor and calcium imaging approaches. The authors first show using slice recordings that activation of PV projections onto the contralateral hemisphere induced inhibitory current in principal cells. Then, by specifically inhibiting this callosal projection they impaired rule shifting. Rule shifting impairment was associated with a decrease interhemispheric gamma synchrony during error trials and an increase in neuronal activity similarity. The authors concluded that callosal PV projections are key for maintaining and updating behavioral strategies.

The concept that specific inhibitory projection pathways are key for the selection of adapted behavioral strategies is very interesting and a promising area of research. The present manuscript attempted to make a strong causal link between a specific inhibitory callosal projection emanating from prefrontal parvalbumin-expressing neurons and extradimensional set shifting. Although I found the concept interesting, I do have some reserve related to the novelty of the results. Moreover, several key experiment and analyses are missing to claim that callosal PV projections are key for maintaining and updating behavioral strategies.

First of all, this manuscript is a follow up of a recent paper from the same group demonstrating that prefrontal gamma synchronization is key for extradimensional set shifting. Second, cortical inhibitory callosal projections are already known (Rock et al., 2017 Cerebral cortex) and gamma synchronization is known to depend on parvalbumin-expressing interneurons. Although this paper demonstrates that callosal PV projections are key in extradimensional set shifting, this represent to my point of view only an incremental advance in the field. Furthermore, the analyses used to demonstrate that the similarity of neuronal representations is increased following inhibition of the PV callosal pathway are preliminary and do not fully support the authors conclusions.

Our response: We discuss the analysis of neuronal representations at length below, in response to the comments related to Figure 4. Regarding the novelty and potential importance of the results: on the one hand, we appreciate what the Reviewer is saying. We have previously shown that prefrontal interneurons and gamma synchrony are required for this task. Now, at first blush, it might seem like we are just adding that callosal PV projections are required for this task, which feels incremental. However, the key advance of this paper is actually a much more fundamental insight into how prefrontal circuits function: **by transmitting synchrony, long-range inhibition can effectively gate how other (excitatory) inputs impact emergent network activity, and it is this gating function which ends up switching prefrontal circuits between different operating modes (maintenance vs. updating).**

This is the first mechanism to causally link specific synapses with the changes in neural dynamics and prefrontal activity patterns that accompany shifts in behavior. Moreover, this mechanism is extremely novel and has surprising implications. To see this, just consider the entirely unexpected finding that selectively inhibiting *only* PV+ callosal projections disrupts both behavior and the evolution of population-level neural activity, whereas neither effect occurs when *all* callosal projections are inhibited. A critical part of this is the new experiment (done for the revision) showing that inhibiting all callosal projections does not disrupt the reorganization of

network activity after rule shift errors (Fig. 4j,m). If one were to poll a group of systems neuroscientists, I have a hard time imagining that *any* of them would predict this discordance, wherein the dramatic effects of selectively inhibiting only callosal PV+ synapses on both behavior and activity vectors completely disappear when all callosal synapses are inhibited (we certainly did not). Putting the findings of the original version of the manuscript together with the results of our revision experiments (inhibiting all callosal projections transiently suppresses gamma synchrony but does not suppress changes in population activity vectors after rule shift errors) shows that callosal communication has the potential to maintain previously-established patterns of activity; however when callosal PV projections are able to transmit gamma synchrony, this maintenance operation becomes pre-empted, allowing for the reorganization of prefrontal activity patterns that serves to update behavior. The same net effect can be achieved by simply inhibiting all callosal communication, making interhemispheric gamma synchrony dispensable in this case – this explains why selectively inhibiting only callosal PV projections is worse than nonspecifically inhibiting all callosal projections. In summary, **this study doesn't just identify one new player in prefrontal-dependent cognition; rather, it links together 6 discrete levels of analysis (PV neurons / callosal PV projections targeting specific mPFC projection neurons / gamma synchrony / callosal inputs / population-level activity patterns / rule shifting behavior) into an unexpected, complex and multifaceted mechanism underlying the more 'unstable' network state that mediates shifts in behavioral strategies.** Specifically, callosal projections from PV neurons target mPFC neurons projecting to the MD and dorsal striatum and transmit gamma synchrony, thereby gating the influence of callosal communication on the emergent population-level activity patterns that guide behavior.

Furthermore, this study indicates that the function of long-range GABAergic projections and gamma synchrony may not simply be to facilitate communication between brain regions (i.e., 'communication through coherence'), but rather to *prevent* that communication from occurring in an *aberrant* manner that would interfere with proper information processing. Long-range GABAergic projections between cortical areas have only recently been identified and their functions are still poorly understood. Conversely, the potential function of gamma synchrony has been one of the most enduring controversies in systems neuroscience for decades – so much so that Jess Cardin and I had the first "Dual Perspectives" debate on this topic at SfN a few years ago in front of a standing-room only crowd. Many studies including our own have shown that gamma oscillations influence behavior and cognition. Nevertheless, many systems neuroscientists continue to resist this idea because the question of exactly how gamma synchrony contributes to brain function has been unanswered. Resolving this question is critical because for ~30 years, scientists have speculated about the potential function of gamma synchrony without any hard evidence to back up their ideas, leading to a huge amount of pushback. Here, we show exactly what happens to network-level activity patterns when you disconnect the callosal GABAergic projections that maintain interhemispheric gamma synchrony: under these circumstances, callosal input maintains previously-established representations, preventing the reorganization of activity that should accompany changes in cognitive strategies.

Finally, we would just add that the original manuscript showed that inhibiting callosal PV projections could lead to long-lasting changes in physiology and cognition; in the revision, we now show that stimulating these projections actually rescues these deficits. This plasticity is not the focus of our study, but it is extremely unique. Finding that transiently manipulating just this one synapse

elicits enduring bi-directional changes in cognition underscores that this is not ‘just another synapse’ but rather a ‘master regulator’ of prefrontal network dynamics and information processing.

Figure 1: The anatomical and functional demonstration that PV-expressing cells project contralaterally and inhibit principal cells deserve additional analyses. First of all, the authors should evaluate whether cells expressing eYFP are indeed PV-expressing cells using immunocytochemistry. The authors should also provide controls that the construct is not leaking in the contralateral side. Optogenetic manipulations on retrogradely labeled cells should also be performed. Second, an AHP over 3 mV is not an absolute criterion to define pyramidal tract cells, again retrograde staining should be done in addition. Quantification of the latency of IPSP should be provided (panel e, IPSP on the first stimulation pulse does not look time-locked).

Our response: First, regarding the comment that “(panel e, IPSP on the first stimulation pulse does not look time-locked)” – we apologize, we now realize that one of the marks indicating light pulses was not properly aligned to the voltage trace in the original version of this figure. We have corrected this figure. The latency from the start of the light flash to the onset of the first IPSPs was 5.1 ± 0.3 msec.

This is the same Cre-driver line we originally used in Sohal et al., *Nature*, 2009, and which have been validated many times since. That said, we have done immunohistochemistry and confirmed that eNpHR-eYFP expression was 96.7% specific for PV interneurons (Extended Data Fig. 1a). As for ‘controls that the construct is not leaking onto the contralateral side’ – we are extremely sensitive to this issue, and in post-hoc histology for all animals used for behavior and in all of our slice experiments, we have never observed any labeled cell bodies contralateral to the injection site. Moreover, we recorded from 18 fast-spiking neurons contralateral to the injection site, and none of them exhibited any light response.

Extended Data Fig. 2e–h already shows experiments designed to control for potential behavioral effects of scattered light from one hemisphere activating eNpHR in PV neuron cell bodies in the contralateral hemisphere. Here, we find that direct, but lower light stimulation of PV cell bodies did not alter the performance of the mice.

To further control for potential (but not observed) leak of the virus to the contralateral side, we have now performed additional control experiments in which *PV-Cre* mice were injected with a diluted concentration of DIO-eNpHR3.0 (200 nL in 800 nL saline) and implanted ipsilaterally with a fiber-optic cannula (to simulate a theoretical but not-observed leak of a small amount of virus into the mPFC contralateral to the injection site). On Day 1, rule shift performance was observed in the absence of optogenetic manipulation. On Day 2, light stimulation of the PV cells infected with lower virus titer did not alter the performance of the mice. These results are shown in Extended Data Fig. 2i–k.

Re: “*Optogenetic manipulations on retrogradely labeled cells should also be performed.*” For our new experiments in which we both optogenetically inhibit (to induce deficits) and then stimulate (to rescue performance) callosal PV projections, we used an intersectional strategy to only label callosally-projecting PV neurons, i.e., we injected AAVretro-Flp into one mPFC of *PV-Cre* mice and implanted an optical fiber on that side, then injected dual recombinase (Cre+Flp) dependent viruses

encoding ChR2 and NpHR into the contralateral mPFC. Thus in this experiment, we are only manipulating the terminals of retrogradely-labeled cells. These data are shown in Extended Data Fig. 7. Note that for this experiment, we optogenetically inhibit callosal PV+ terminals on Day 1, so these mice become impaired and require 20–35 trials to learn the rule shift, compared to the typical range of 10–15 trials observed in the absence of optogenetic stimulation (Figs. 2f, 3e, 4e).

To further characterize the projection targets of mPFC pyramidal neurons that receive callosal mPFC PV synapses, we retrogradely labeled four brain structures using fluorescent dye-conjugated cholera toxin subunit B (CTb). These structures were the contralateral mPFC (i.e., ipsilateral to the DIO-ChR2 injection but contralateral to the side for recordings), dorsal striatum, mediodorsal thalamus, and nucleus accumbens. These results are shown in Extended Data Fig. 1. Strikingly, callosal PV projections innervated 22/22 MD-projecting pyramidal neurons compared to 0/18 callosally-projecting pyramidal neurons, 0/21 nucleus accumbens-projecting pyramidal neurons, and 7/24 dorsal-striatum projecting pyramidal neurons. These results provide critical information about the ‘wiring diagram’ for these novel callosal GABAergic projections, and show that callosal PV+ projections regulate prefrontal output in a target circuit-specific manner. The striking difference between MD-projecting neurons (100% connectivity) vs. callosal and nucleus accumbens-projecting neurons (0% connectivity) is particularly notable in light of our previous findings that mPFC-MD outputs are critical for learning in a similar extradimensional rule shifting task (Marton et al., *J Neurosci*, 2016).

Because we now present these data using CTb-labeled projection subtypes, we have removed the previous PT vs. IT distinction from Figure 1.

Figure 2: I found overall that the behavioral data should be described in much more details. The fine timescale analysis of the behavioral data should be provided as it may represent a serious confound on the interpretation of the results. Before to be averaged, data should be provided by trial types in order to evaluate the dispersion of the data and potential bias in stimulus selection. Concerning the optogenetic manipulation, the authors must provide control experiments demonstrating that the manipulation does not impact motor behavior and analyze in details the animal behavior during the time course of the manipulation. Another concern is the design of the optogenetic stimulation, once IA criterion is reached (8 correct out of 10 consecutive trials), it is not clear why the authors first inhibited 3 additional IA trials before to inhibit RS trials. It is possible that the manipulation impacts the original association rather than the rule shifting aspect of the task. Also, the fact that the authors inhibited the entire RS sequence does not allow to evaluate if the animal would have succeeded the RS task on this particular trial type.

Our response: First, for behavioral datasets in which we used video recordings to perform time-stamped analyses (e.g., dual-site voltage indicator and calcium imaging experiments), we have computed the latency to dig (aka the ‘time to choice’). This is shown in Extended Data Fig. 4c. Using two-way ANOVA, there is no significant change in the time-to-choice from the first 5 initial association trials to the first 5 rule shift trials across control and experimental virus cohorts.

To quantify possible effects on left vs. right orienting, turning, etc., we quantified the fraction of trials in which the mouse chose the bowl that was ipsilateral vs. contralateral to the side of optogenetic inhibition. This was specifically done during the 3 trials of initial association which

occurred after the mouse reached the learning criterion and we began delivering optogenetic inhibition (but before the onset of the rule shift). These data are shown in Extended Data Fig. 4b. Using two-way ANOVA, there is no difference in the ipsi- vs. contralateral choice percentage made across control and experimental virus cohorts (side choice × virus interaction: $F_{(1,67)} = 0.7813$, $P = 0.3799$).

We have now performed additional experiments which show that inhibiting callosal PV projections has no effect on the learning of intradimensional rule reversals (in a task which is otherwise identical to the rule shift task) (Extended Data Fig. 6). This also shows that the behavior effects of inhibiting callosal PV projections do not reflect changes in motor or orienting behaviors as the overall mechanics of the rule reversal and rule shift tasks are identical.

This is similar to our previous finding that a much stronger manipulation – inhibition of *all* GABAergic neurons in the prefrontal cortex – does not affect learning of an initial association or intradimensional rule reversal (Cho et al., *Neuron*, 2015).

By switching on the inhibition for the last 3 trials of the initial association, we have been able to confirm that callosal PV inhibition does not affect performance during this period indicating that it does not affect decision making non-specifically. (Being able to distinguish performance and motor behavior during these trials is an important control that contrasts with the dramatic effects of callosal PV inhibition during the subsequent rule shift – which is why we deliver light during these 3 IA trials. Another benefit is that we could use these trials to confirm that optogenetic inhibition of callosal PV projections does not affect the similarity of activity patterns occurring before the rule shift, Extended Data Fig. 10q-r).

Showing that inhibiting callosal PV projections does not affect the 1) time to choice / latency to dig, 2) performance of a well-learned initial association, 3) bias for left vs. right choices, and 4) learning of an intradimensional rule reversal, confirms that the effects we observe are not due to motor effects, nonspecific aspects of attention / motivation / decision making, specific cue configurations / trial types, or even learning in general. Rather, the behavioral effects of inhibiting callosal PV projections must be highly specific for learning new rules only when they specifically rely on cues that were previously present but irrelevant to trial outcomes. We are not completely certain what the Reviewer means by *'Before to be averaged, data should be provided by trial types in order to evaluate the dispersion of the data and potential bias in stimulus selection... the fact that the authors inhibited the entire RS sequence does not allow to evaluate if the animal would have succeeded the RS task on this particular trial type,'* but note two things. First, the fact that the behavioral effect was specific for rule shifts (RS) and did not occur for rule reversals (RR) indicates that the induced deficit reflects the nature of the task (extra vs. intradimensional shift) rather than just difficulty in responding to a particular stimulus or stimulus combination. Second, we have now separately analyzed trial outcomes, depending on whether the correct initial association (IA) and correct RS cues were located in the same vs. different bowls, for the first 5 RS trials (Extended Data Fig. 4f) or the next 5 trials (Extended Data Fig. 4g). This shows that the main effect of inhibition is to increase RS errors specifically when the correct IA and RS cues are located in different bowls. (Note that during the rule reversals, the IA and RR cues are always in different bowls, but optogenetic inhibition does not disrupt performance. This further underscores that optogenetic inhibition causes a deficit which strongly depends on the nature of the task rather than just the trial type).

Figure 3: Gamma synchrony should also be demonstrated using classical LFP recording approaches and dynamically analyzed over the time course of the trials/stimulation. Raw data should be provided as well. The long-lasting effect of the manipulation on both behavior and gamma synchrony is puzzling and the authors should at least control that the callosal PV pathway is still functional on day 3 and that the stimulation does not induce apoptosis of this cell population.

Our response: In our previous work (Cho et al., *Nat Neurosci*, 2020), we simultaneously measured fluorescence from voltage indicators and LFPs. While we did see increases in gamma synchrony (specifically zero-phase lag wavelet coherence), this was more modest and less specific than what we saw using genetically encoded voltage indicators (i.e., LFPs exhibited large increases in synchrony at lower frequencies, outside the gamma band). These additional, less specific findings likely reflect the passive propagation of electrical fields across the relatively short interhemispheric distance. In this context, we are concerned about the limitations of LFP recording for this purpose, and are not sure what additional crucial information would be gained by this approach.

That being said, we agree that other parts of the Reviewer's comment are very relevant. We have recently developed additional ways of analyzing TEMPO data, and can now resolve increases in synchrony happening on faster timescales of seconds (Methods, lines 1132-1141). Indeed, we see that immediately following an error during the rule shift, there is an increase in PV interneuron gamma synchrony during the 5 seconds following the choice. This is in fact why we analyzed calcium imaging data during the 10 seconds following each choice – to examine changes in population activity that immediately follow this heightened gamma synchrony. While this new analysis does not capture exactly the same information as the synchrony metric that is the focus of this study (this newer metric is limited to a much shorter period of time, whereas the original metric encompasses a much larger time window including the intertrial interval), we do observe the same basic phenomenon: that the increase in gamma synchrony which normally occurs after error trials is abolished when we inhibit callosal PV+ projections (Extended Data Fig. 9). While we do not wish to supplant our original analysis, which remains the focus of our manuscript, this additional analysis provides some additional context and corroboration.

We are not sure exactly what the Reviewer means by 'raw data' here – we did show multiple examples of raw and filtered mNeon and TdTomato fluorescence traces in the publication in which we originally introduced this method for analyzing synchrony using TEMPO (Cho et al., *Nat Neurosci*, 2020). Differences in synchrony cannot be discerned from these raw data, but we are happy to show them again if the Reviewer feels they would be helpful.

Finally, we have performed new experiments in which we inhibit callosal PV projections during rule shifts on Day 1, then verify that rule shift performance is persistently impaired on Day 2. Then, on Day 3, we optogenetically stimulate these same callosal PV projections. In our initial results, 40 Hz stimulation of callosal PV projections is sufficient to rescue performance on Day 3 (and this rescue persists on 'Day 4' of testing which occurs 1+ weeks later). These data, which demonstrate that not only is this pathway still functional, but actually capable of inducing therapeutic network plasticity, is shown in Extended Data Fig. 7.

Our new experiments do provide some insight into the mechanism of plasticity contributing to the enduring changes in gamma synchrony and activity patterns – this is discussed below within our response to the next comment, in the context of additional new experiments.

Figure 4: This dataset is very preliminary. The evaluation of the similarity of neuronal activity patterns is done using a vector that is obtained by (1) averaging 10 second of activity after the animal decision and (2) additional trial averaging. The authors should reconsider their analyses as this one does not provide a good estimate of neuronal activity pattern. I suggest the authors to evaluate changes in neuronal activity pattern dynamically rather than statically and use machine learning algorithm to decode neuronal information. Changes in neuronal activity should also be clearly link to behavioral analyses that is a potential confound. Without a deep analysis of neuronal activity pattern with the appropriate controls, the data provided do not support the conclusions. An effort should also be made to provide raw data to see how neurons change their activity during the task.

Our response: We appreciate where the Reviewer is coming from – machine learning approaches are obviously being used a lot in systems neuroscience, and we ourselves have made great use of them in the past. However, we thought a great deal about what is the appropriate type of analysis for these datasets before settling on the approach we used. There are two key considerations here. First, we focused on the 10 seconds following the choice (time to dig), because, as described above, our newer, high temporal-resolution TEMPO analysis shows that gamma synchrony increases during the 5 seconds following each choice and we wanted to examine changes in activity that immediately follow this period of heightened synchrony (Extended Data Fig. 9). So, if we want to understand the changes in neural representations caused by these changes in the circuit dynamics, then this is the appropriate window to look at. Second, we are studying a highly naturalistic task which mice perform well, as evidenced by their rapid learning. As a result, we have very small number of trials to use, i.e., in many cases, mice only make 1-2 errors before learning the rule shift. The major weakness of machine learning is that it requires a large amount of data to train. In cases where mice have only 1-2 error trials, we don't see how it is possible (or appropriate) to do a machine learning analysis.

Similarly, we understand the Reviewer is interested in looking at activity patterns dynamically. But there are many caveats here. First, if there is a clear difference in the static activity pattern, why would that not be compelling? This seems like a simpler result, that involves less processing of the data and fewer statistical comparisons compared to finding differences in dynamic activity patterns. Second, many studies that look for differences in dynamic patterns do so by examining trajectories within a lower dimensional subspace, after performing dimensional reduction using standard approaches such as PCA. Again, the identification of dimensions, and the smoothing of trajectories works best in the presence of many trials and a large amount of data. Third, activity within these datasets is quite sparse, i.e., each neuron is typically active only in ~1-10% of frames, which means that in each frame only a handful of neurons are typically active. In this context, it makes sense to average activity over a larger time window, as we do. Again, we have focused on the 10 seconds following the choice because this corresponds to the time window when gamma synchrony is elevated on error trials.

We note that our focus on (static) activity vectors rather than (dynamic) trajectories is also influenced by the fact that we are measuring network activity using calcium imaging, rather than electrophysiology. In calcium imaging, we get a larger number of neurons (~60-100 cells), but activity in these neurons is sparsely distributed. By contrast, electrophysiology records from a smaller number of neurons (typically ~5-15), but levels of activity (firing rates) in these neurons vary continuously over a large dynamic range. Therefore, the key question for our analysis is which neurons are active during a period of time, for which looking at the angle between population vectors (as we have done) seems most appropriate. By contrast, electrophysiological datasets, which record from a smaller number of neurons but measure continuous changes in their firing rates, tend to quantify changes in the relative activity of individual neurons in order to answer a different question: how active is each neuron?

Indeed, the usefulness of our approach is illustrated by some new experiments / analyses done for the revision, as well as others that were present (but not highlighted) in the original manuscript. Specifically, whereas inhibiting callosal PV projections made activity vectors occurring after RS errors more similar to those after RS correct choices and IA errors (indicating that changes in activity patterns after RS errors are attenuated), we have now found that there is no change in these similarities when we inhibit all callosal projections. As discussed above, this supports our interpretation that callosal PV projections normally gate the ability of other callosal inputs to maintain previously-established activity patterns. Furthermore, optogenetic inhibition of callosal PV projections does not affect the similarity of activity patterns that occur before the rule shift (Extended Data Fig. 10q-r). These two controls underscore that our metric (similarity of activity vectors) is highly sensitive and specific for the changes in population-level activity after RS errors that are regulated by callosal PV projections.

Nevertheless, following the Reviewer's suggestion, we did attempt some machine learning analyses. Specifically, we used the approach we developed in a recent publication (Frost et al., *PLoS Biology*, 2021) using a linear neural network classifier, in which the activity of the imaged neurons is mapped onto an input layer with fixed, random, sparse connections to a hidden layer (containing 1000 units), which connect to a single output unit via output weights that undergo training. As we showed in Frost et al., this architecture performs similarly to optimal linear coders (logistic classifiers or SVMs) when information is encoded in the activity levels of individual neurons, and significantly outperforms them when information is encoded by correlated activity. We trained the classifiers on the last 10 trials of the initial association and tested them on the first 8 trials of the rule shift to get at the question of whether there was information that was being lost or inappropriately maintained, when we inhibited callosal PV projections. We made separate classifiers for four different time windows: trial start, dig start, dig end, and intertrial interval. In each case, the time window was up to 10 seconds (but did not cross over into the next trial event). We compared the performance of classifiers trained on real data to performance after labels had been shuffled on a trial-wise basis. We did not observe consistent above-shuffled performance during any of these time periods for decoding the chosen side (left vs. right), reward location, or the locations of the cues associated with either the initial association or rule shift. Next, we decoded trial outcome. In this case, we did see that decoding of trial outcome during the 'dig start' period (i.e., the same period analyzed that is the focus of our original analysis) was consistently better for real than shuffled data. While there were some potentially interesting trends towards between-day differences, these did not reach statistical

significance. Again, this likely reflects the fact that classifier performance was extremely variable, because there is not enough data to properly train classifiers.

Redacted

We have added information about levels of activity during task periods and how these are affected by different manipulations. Specifically, when mice perform the task in the absence of any manipulations, activity immediately after errors is higher than immediately after correct trials (Fig. 4o). This difference is abolished when we inhibit callosal PV projections and remains persistently absent even the next day (Fig. 4p). Interestingly, nonspecifically inhibiting all callosal projections (using synapsin-eNpHR) also eliminates the difference between activity after errors vs. correct trials, but in this case the effect is transient and does not persist (Fig. 4q; this is identical to the effect of inhibiting all callosal projections on gamma synchrony which we observed in new experiments done for the revision, Fig. 3n). This indicates that eliminating just this change in overall activity levels (without altering the changes in activity patterns) following rule shift errors is not sufficient to disrupt rule shift learning. That being said, the fact that inhibiting all callosal projections produces this transient change in activity levels is a useful positive control that confirms that this manipulation does perturb network activity.

Finally, several new observations provide some (limited) insight into the potential mechanism underlying long-lasting deficits in rule shift behavior, gamma oscillations, and abnormal activity patterns induced by inhibiting callosal PV projections. Specifically, we found that: 1) inhibiting all callosal projections disrupts gamma synchrony and activity levels transiently but not persistently; 2) inhibiting all callosal projections does not affect the reorganization of activity patterns that occurs after rule shift errors; 3) inhibiting callosal PV projections during a rule reversal does not impair performance on a rule shift the next day. Observation 3 indicates that just inhibiting callosal PV projections is not sufficient to elicit enduring plasticity – this inhibition must be delivered during the rule shift. Observation 1 indicates inhibition of callosal PV projections must also occur when other forms of callosal communication is intact in order to produce plasticity. This suggests that the mechanism for plasticity is not solely dependent on callosal PV projections, but also requires task-related activity involving other callosal projections. One possibility is that when callosal PV projections are selectively inhibited, mistimed task-related activity in other callosal projections causes those projections to undergo deleterious plasticity.

Referee #3 (Remarks to the Author):

The manuscript of Cho et al builds upon previous studies, including two from the same authors, to address the mechanisms of cognitive flexibility in the mammalian cortex. This previous work established that rule shifts leads to cross-hemispheric gamma synchrony of PV+ neurons in the PFC and that rule shifts results in changes and/or reorganization of PFC activity. The experiments here are described as an effort to connect this current knowledge at the levels of the circuit, network dynamics and neural representations. To this end, the authors employ optogenetic behavior, ex vivo physiology, in vivo cell-type specific voltage imaging and in vivo endoscopic calcium imaging to conclude that callosal projecting PV neurons are required for a switch from maintenance to updating of PFC neural representations in response to changes in the rules of the task. The paper is well written and includes some interesting findings, however in its current form I don't see a significant shift in our understanding compared to what is has been described in the previous work.

Our response: On the one hand, we appreciate what the Reviewer is saying. We have previously shown that prefrontal interneurons and gamma synchrony are required for this task. Now, at first blush, it might seem like we are just adding that callosal PV projections are required for this task, which feels incremental. However, the key advance of this paper is actually a much more fundamental insight into how prefrontal circuits function: **by transmitting synchrony, long-range inhibition can effectively gate how other (excitatory) inputs impact emergent network activity, and it is this gating function which ends up switching prefrontal circuits between different operating modes (maintenance vs. updating).**

This is the first mechanism to causally link specific synapses with the changes in neural dynamics and prefrontal activity patterns that accompany shifts in behavior. Moreover, this mechanism is extremely novel and has surprising implications. To see this, just consider the entirely unexpected finding that selectively inhibiting *only* PV+ callosal projections disrupts both behavior and the evolution of population-level neural activity, whereas neither effect occurs when *all* callosal projections are inhibited. A critical part of this is the new experiment (done for the revision) showing that inhibiting all callosal projections does not disrupt the reorganization of network activity after rule shift errors (Fig. 4j,m). If one were to poll a group of systems neuroscientists, I have a hard time imagining that *any* of them would predict this discordance, wherein the dramatic effects of selectively inhibiting only callosal PV+ synapses on both behavior and activity vectors completely disappear when all callosal synapses are inhibited (we certainly did not). Putting the findings of the original version of the manuscript together with the results of our revision experiments (inhibiting all callosal projections transiently suppresses gamma synchrony but does not suppress changes in population activity vectors after rule shift errors) shows that callosal communication has the potential to maintain previously-established patterns of activity; however when callosal PV projections are able to transmit gamma synchrony, this maintenance operation becomes pre-empted, allowing for the reorganization of prefrontal activity patterns that serves to update behavior. The same net effect can be achieved by simply inhibiting all callosal communication, making interhemispheric gamma synchrony dispensable in this case – this explains why selectively inhibiting only callosal PV projections is worse than nonspecifically inhibiting all callosal projections. In summary, **this study doesn't just identify one new player in prefrontal-dependent cognition; rather, it links together 6 discrete levels of analysis (PV neurons / callosal PV projections targeting specific mPFC projection neurons / gamma synchrony / callosal inputs /**

population-level activity patterns / rule shifting behavior) into an unexpected, complex and multifaceted mechanism underlying the more ‘unstable’ network state that mediates shifts in behavioral strategies. Specifically, callosal projections from PV neurons target mPFC neurons projecting to the MD and dorsal striatum and transmit gamma synchrony, thereby gating the influence of callosal communication on the emergent population-level activity patterns that guide behavior.

Furthermore, this study indicates that the function of long-range GABAergic projections and gamma synchrony may not simply be to facilitate communication between brain regions (i.e., ‘communication through coherence’), but rather to *prevent* that communication from occurring in an *aberrant* manner that would interfere with proper information processing. Long-range GABAergic projections between cortical areas have only recently been identified and their functions are still poorly understood. Conversely, the potential function of gamma synchrony has been one of the most enduring controversies in systems neuroscience for decades – so much so that Jess Cardin and I had the first “Dual Perspectives” debate on this topic at SfN a few years ago in front of a standing-room only crowd. Many studies including our own have shown that gamma oscillations influence behavior and cognition. Nevertheless, many systems neuroscientists continue to resist this idea because the question of exactly *how* gamma synchrony contributes to brain function has been unanswered. Resolving this question is critical because for ~30 years, scientists have speculated about the potential function of gamma synchrony without any hard evidence to back up their ideas, leading to a huge amount of pushback. Here, we show exactly what happens to network-level activity patterns when you disconnect the callosal GABAergic projections that maintain interhemispheric gamma synchrony: under these circumstances, callosal input maintains previously-established representations, preventing the reorganization of activity that should accompany changes in cognitive strategies.

Finally, we would just add that the original manuscript showed that inhibiting callosal PV projections could lead to long-lasting changes in physiology and cognition; in the revision, we now show that stimulating these projections rescues these deficits. This plasticity is not the focus of our study, but it is extremely unique. Finding that transiently manipulating just this one synapse elicits enduring bidirectional changes in cognition, underscores that this is not ‘just another synapse’ but rather a ‘master regulator’ of prefrontal network dynamics and information processing.

Major points:

1) On the level of the circuit, the paper begins with anatomy to establish the postsynaptic targets of the contralaterally projecting PFC PV+ neurons. The data show that these axons synapse on pyramidal neurons, with a significant bias to neurons with a pyramidal tract electrophysiological signature, but stops there. The authors do not investigate input to GABAergic neurons in the contralateral hemisphere, nor do they offer additional evidence on the projection targets of the PT neurons they identify as the major recipient of the PV input; both these shortcomings limit the usefulness and scope of these data. In the discussion of their own previous Nature Neuroscience (2020) paper they highlight the importance of identifying the targets of the pyramidal neurons receiving cross-hemispheric input, but do not include those experiments. Combining the approach used here with the injection of retrograde labels in potential targets (ie. mediodorsal thalamus, nucleus accumbens, striatum) would have made these findings much more useful in interpreting how these circuits shift behavior via the communication between multiple regions.

Our response: We agree with the Reviewer. In our initial experiments, we didn't observe synapses onto fast-spiking neurons but the N for this was fairly modest so we didn't report it. For the revision, we have done exactly the experiment suggested by the Reviewer, and made recordings in slices to assay connectivity of callosal PV+ projections onto fast-spiking neurons as well as retrogradely-labeled neurons projecting to the mediodorsal thalamus, contralateral mPFC, dorsal striatum and nucleus accumbens. These results are shown in Extended Data Fig. 1. Strikingly, we observed that callosal PV projections innervated 22/22 MD-projecting pyramidal neurons compared to 0/18 callosally-projecting pyramidal neurons, 0/21 nucleus accumbens-projecting pyramidal neurons, 7/24 dorsal-striatum projecting pyramidal neurons, and 0/18 fast-spiking neurons. These results provide critical information about the 'wiring diagram' for these novel callosal GABAergic projections. Furthermore, as suspected by the Reviewer, they indicate that callosal PV+ projections regulate prefrontal output in a target circuit-specific manner. The striking difference between MD-projecting neurons (100% connectivity) vs. callosal and nucleus accumbens-projecting neurons (0% connectivity) is particularly notable in light of our previous findings that mPFC-MD outputs are critical for learning in a similar extradimensional rule shifting task (Marton et al., *J Neurosci*, 2016).

2) On the level of neural representation- the calcium imaging data from the PFC is not described in sufficient detail, with the authors only showing measurements which captures the similarity based on angle between two population vectors while remaining agnostic to their magnitude. The raw data is not described or discussed- it is impossible to even determine how many units were observed, how many active units contribute to the vectors used for analysis, etc- these first order data must be included in extended data to appreciate the strength and validity of the similarity claims.

Our response: We apologize – in the original manuscript, we omitted this first order information in the interest of keeping things simple and focusing on only the most salient results, but with the benefit of hindsight, this was a mistake. Numbers of neurons are now shown in Extended Data Fig. 10, and the levels of activity (averaged across all neurons) are now shown in Fig. 4o-q. Examples of calcium traces, an activity raster, and the population activity vector for a correct vs. incorrect RS trial are shown in Fig. 4g. When mice perform the task in the absence of any manipulations, activity immediately after errors is higher than immediately after correct trials (Fig. 4o). This difference is abolished when we inhibit callosal PV projections and remains persistently absent even the next day (Fig. 4p). Interestingly, nonspecifically inhibiting all callosal projections (using synapsin-eNpHR) also eliminates the difference between activity after errors vs. correct trials, but in this case the effect is transient and does not persist (Fig. 4q; this is identical to the effect of inhibiting all callosal projections to transiently blunt gamma synchrony which we observed in new experiments done for the revision, Fig. 3n). This indicates that eliminating just this change in overall activity levels (without altering the changes in activity patterns) following rule shift errors is not sufficient to disrupt rule shift learning. (Although the fact that inhibiting all callosal projections produces this transient effect does serve as a positive control which confirms that this manipulation is perturbing network activity).

Further, previous physiological recordings in the PFC (for example, Durstewitz et al Neuron 2010, Karlsson et al Science 2012, Schmitt et al Nature 2017 among others the authors cite) provided numerous examples of how the activity patterns could be analyzed to understand the link between population activity and behavioral shifts; employing similar approaches here would make these data much more useful to understanding the role of the PV projections in this reorganization process.

Our response: We thought a great deal about the appropriate type of analysis to do on these datasets before settling on the approach we used. There are two key considerations here. First, we focused on the 10 seconds following the choice (time to dig), because, using a new higher temporal-resolution analysis of TEMPO data, we see increased gamma synchrony during the 5 seconds following choices on error trials (Extended Data Fig. 9), and we wanted to examine changes in activity that immediately follow this period of elevated synchrony. Furthermore, this increase in gamma synchrony only lasts during the first few trials of the rule shift. So, if we want to understand the changes in neural representations caused by changes in the circuit dynamics, this is the appropriate window to look at. Specifically, during the trials when the animal is learning the rule shift, we would like to compare activity on error vs. correct trials, focusing on the first 10 sec after the choice.

Second, we are studying a highly naturalistic task which mice are well-suited to perform as evidenced by their rapid learning. As a result, we have a very small number of trials to use. I.e., in many cases, mice only make 1-2 errors before learning the rule shift. The major weakness of machine learning is that it requires a large amount of data to train. In cases where mice have only 1-2 error trials, we don't see how it is possible to do a machine learning analysis like the linear decoder employed by Schmitt et al. (2017). Nevertheless, we did attempt some machine learning analyses as suggested by the Reviewer. Specifically, we used the approach we developed in a recent publication (Frost et al., *PLoS Biology*, 2021) using a linear neural network classifier, in which the activity of the imaged neurons is mapped onto an input layer with fixed, random, sparse connections to a hidden layer (containing 1000 units), which connect to a single output unit via output weights that undergo training. As we showed in Frost et al., this architecture performs similarly to optimal linear coders (logistic classifiers or SVMs) when information is encoded in the activity levels of individual neurons, and significantly outperforms them when information is encoded by correlated activity. We trained the classifiers on the last 10 trials of the initial association and tested them on the first 8 trials of the rule shift to get at the question of whether there was information that was being lost, or inappropriately maintained, when we inhibited callosal PV projections. We made separate classifiers for four different time windows: trial start, dig start, dig end, and intertrial interval. In each case the time window was up to 10 seconds (but did not cross over into the next trial event). We compared the performance of classifiers trained on real data to performance after labels had been shuffled on a trial-wise basis. We did not observe consistent above-shuffled performance during any of these time periods for decoding the chosen side (left vs. right), reward location, or the locations of the cues associated with either the initial association or rule shift. Next, we decoded trial outcome. In this case we did see that decoding of trial outcome during the 'dig start' period (i.e., the same period

analyzed that is the focus of our original analysis) was consistently better for real than shuffled data. While there were some potentially interesting trends towards between-day differences, these did not reach statistical significance. Again, this likely reflects the fact that classifier performance was extremely variable, because there is not enough data to properly train classifiers.

Redacted

Karlsson et al. (2012) analyzed population activity using a metric which quantified the amount of change in high dimensional representations of network activity – this is quite similar to the metric we used, which also quantifies the difference between two different population vectors in a high dimensional space. (In fact, Karlsson et al. was part of the motivation for the approach we took).

Durstewitz et al. (2010) similarly analyzed activity by quantifying the similarity between population vectors (in their case, they used the distance between population vectors). This is again very similar to what we did. Note that we measured data using microendoscopic calcium imaging – as a result we had a much larger number of neurons per dataset (e.g., ~60-100 neurons vs. 5-15 neurons in Karlsson et al. and Durstewitz et al.). Like most microendoscopic calcium imaging studies, we detected periods of activity to generate a binarized activity raster. Activity in this raster was sparse (neurons tended to be active on ~1-10% of frames). Thus, in our dataset, which consists of a large number of neurons with sparse activity the major question is which particular neurons are active – this is best captured by the cosine similarity metric we used. By contrast, in the Karlsson et al. and Durstewitz et al. datasets, which consist of a small number of neurons with continuously varying spike rates, the major question is how active are different neurons. So overall, what we did is quite similar to Karlsson et al. and Durstewitz et al., except that due to the inherent differences between calcium imaging vs. electrophysiological datasets, we measured the similarity of population activity using an angle-based metric whereas those studies used distance-based metrics.

Indeed, the usefulness of our approach is illustrated by some new experiments / analyses done for the revision, as well as others that were present (but not highlighted) in the original manuscript. Specifically, whereas inhibiting callosal PV projections made activity vectors occurring after RS errors more similar to those after RS correct choices and IA errors (indicating that changes in activity patterns after RS errors are attenuated), we have now found that there is no change in these similarities when we inhibit all callosal projections. As discussed above, this supports our interpretation that callosal PV projections normally gate the ability of other callosal inputs to maintain previously-established activity patterns. Furthermore, optogenetic inhibition of callosal PV projections does not affect the similarity of activity patterns that occur before the rule shift (Extended Data Fig. 10q-r). These two controls underscore that our metric (similarity of activity vectors) is highly sensitive and specific for the changes in population-level activity after RS errors that are regulated by callosal PV projections.

3) Regarding neural dynamics- one key unaddressed question is if it is crucial that the inhibition of the contralaterally projecting PV terminals occurs during the first instance of rule shift learning? In the authors' 2015 Neuron paper, bilateral gamma stimulation rescued RS behavior in the *Dlx5/6+/-* mice and this rescue improved behavior on subsequent days even in the absence of optogenetic stimulation, without the reappearance of gamma synchrony. In the 2020 Cho et al Nature Neuroscience paper out-of-phase stimulation impaired RS behavior, but this could be reinstated the next day with in-phase stimulation. In the current work the authors show that optogenetic inhibition of the PV terminals impairs behavior in the first rule shift session and leads to continued poor performance on subsequent days, suggesting what they term a novel form of network plasticity. This result needs to be understood and described more carefully. Can this deficit be rescued with gamma stimulation as in the mutant mice? A key experiment would be to reverse the day 2 and day 3 protocol in figure 3; does successful RS learning on day 2 make this projection dispensable for later RS performance? Does this lead to another dissociation between performance and the increase in gamma synchrony following error trials, as seen in the *Dlx5/6* mice? Is the PV projections and gamma only required for the first instance of rule shift learning and then becomes dispensable?

Our response: With regard to the issue of whether the callosal PV projection is only required for the first instance of rule shift learning and thereafter dispensable – the answer is no. We apologize that this was unclear in our original manuscript, but in most of our experiments, mice did not undergo any manipulations on the first day of testing, i.e., during the first rule shift (Fig. 2c,f; Fig. 3c,e; Fig. 4a,e).

Re: whether the deficit can be rescued – we have now completed experiments which express both NpHR and ChR2 in callosally-projecting PV interneurons via an intersectional labeling (dual recombinase-dependent) strategy. These experiments inhibit callosal PV projections during rule shifts on Day 1, verify that rule shift performance is persistently impaired on Day 2 in the absence of further optogenetic manipulation, then optogenetically stimulate the callosal PV projections on Day 3 and test whether this leads to persistent changes in performance on 'Day 4' (which occurs at least 1 week later). These results, shown in Extended Data Fig. 7, show that 40 Hz stimulation of callosal PV projections elicits a rescue on Day 3 that persists for 1+ weeks. This establishes that the ability of callosal PV projections to induce plasticity is bidirectional – inhibiting these projections causes deleterious plasticity, whereas stimulating them (at 40 Hz) can persistently reverse those deficits.

4) Related to point 3- I am also confused on the interpretation of the data in extended figure 2i-p, the experiments with optogenetic inhibition of all callosal projections. This is an interesting finding, but as this manipulation should inhibit both PV and glutamatergic projections, shouldn't it also disrupt gamma synchrony across the hemispheres- this should be measured. If there is dissociation of initial RS learning and the error-triggered increase in gamma synchrony, do the authors believe a second mechanism is recruited under these conditions? Is the take-home message that just callosal excitatory input is worse for performance than input at all? Related to this is the conceptual question- why is bilateral synchrony required?

The authors refer to communication between the hemispheres becoming “mistimed” when the contralateral PV projections are inhibited. I don’t think they have done experiments to address this and should consider using terminology that better reflects their results. If it is about the timing of convergent input from the two hemispheres to a putative downstream target, then its possible asynchronous input could be more detrimental to performance than no input from one hemisphere at all- is this the authors working model? I agree that the error triggered synchrony is an interesting and compelling observation and may correlate with a shift in the PFC pyramidal cell activity patterns, but with the data included in the paper fall short of making the connections between the anatomy, the circuit dynamics and neural representations that the authors are striving for.

Our response: We appreciate the Reviewer’s suggestion and have performed experiments to test whether inhibiting all callosal projections disrupts gamma synchrony. As is now shown in Fig. 3j–n, this manipulation does transiently disrupt gamma synchrony (but these deficits do not outlast the period of inhibition). Regarding the interpretation: yes, we believe that when callosal communication occurs in the absence of the interhemispheric gamma synchrony produced by callosal PV inhibition, then it is indeed worse than no callosal communication at all. We believe this occurs because when gamma synchrony is lost, callosal communication ends up reinforcing previously-established patterns of activity, as evidenced by our calcium imaging experiments. By contrast, when the hemispheres are synchronized (by callosal PV projections), then the communication ends up occurring in a coordinated manner that allows the network activity occurring after rule shift errors to diverge from previously-established representations, facilitating the rule shift. The Reviewer seems to feel that the current data does not quite establish this interpretation as strongly as they would like. We agree and therefore performed additional microendoscopic calcium imaging experiments in which we inhibit all callosal projections. Under these circumstances, the representations after rule shift errors diverge normally from those after rule shift correct trials (and those after initial association errors) (Fig. 4f,j,m). This experiment represents a strong additional test of our interpretation that callosal communication that occurs in the absence of interhemispheric PV gamma synchrony is worse than no callosal communication at all, because it interferes with the normal evolution of neural activity patterns. It also confirms that the function of callosal PV projections / interhemispheric PV gamma synchrony is to prevent this interference, because when callosal PV projections and all other callosal projections are blocked, there are no longer deleterious effects on activity patterns and behavior (even though interhemispheric PV gamma synchrony is lost).

Minor points:

1) The labels on the y-axes of Extended Data Fig. 2 are confusing. Currently all read “trials to criterion” however what is being reported in these graphs is the number of errors of a given type; shouldn’t the labels simply read “# of errors”?

Our response: Thank you, we have corrected the figure to read ‘Number of errors’ in what is now Extended Data Fig. 5.

Reviewer Reports on the First Revision:

Referees' comments:

Referee #1 (Remarks to the Author):

The response to reviewer 2 argues how important this breakthrough finding may be to the field in terms of understanding basic mechanisms of shifting. This argument made by the authors underscores the need to ensure these effects are generalizable and can be replicated. This remains my major concern, even after the revision. Another major concern is the lack of information about the plasticity effect where a carry over effect is seen on day 3. I am left feeling there are too many uncertainties and unknowns to feel confident about the claims of this paper.

The authors have somewhat strengthened the paper by adding a reversal learning control task. Reversal has been associated with other PFC/OFC areas in previous work, so a negative result is informative and consistent with expectations. However I have two concerns. N was very low (3 mice) and they only reversed odor but not texture. As this is the far less salient of the compound cues, texture should be added as well to ensure the PV colossal NpHR effect is not about moving from a salient cue to a non salient cue or vice versa.

"Reviewer fig 1 " shows that the long silencing stimulation through multiple phases (outcome and ITI) is critical but it is not clear why this period need be so long. These data should be shown to the reader in the published materials. Also, the figure raises rather than allays my concern that there is something extra (and complex to interpret) about this particular training protocol and its multiple elements. What are the series of cognitive processes that occur and need to be disrupted in series to affect the behavior? Lack of effects by temporally limited stimulation suggests the observations may not generalize to other task formats that lack this extra complexity.

When challenged to adjust this unusual and likely complex manipulation of timeout in my first review the authors responded "Changing this [timeout] duration could obviously open up a can of worms, as some mice would fail to learn the association and the variability of performance would increase dramatically." I disagree. An alternative way to run this shifting task inside the same cage with shorter timeout and little handling is well established and groups of n=10-12 are regularly used to reveal significant effects of brain manipulations on performance in this task. In this more established but less efficient version the animal does not leave the holding cage would still test shift odor to texture or vice versa (the key cognitive construct of interest). Even mice and rats with a bilateral PrL lesion eventually will learn an extradimensional shift task, just with more trials to criterion.

To spell it out clearly, the behavioral change would be more convincing if it were shown in another shifting protocols. Given how highly honed and unusual this task protocol is with the holding cage timeout for 2 minutes, and very low trial times, and very low animal numbers, it is not a given that observed effects will generalize. If the effect were to be lost in a less honed version of the task, then the authors might seek to understand what aspect of the effective but complex punishment period was disrupted by NpHR stimulation.

Plasticity and day 3 effects are still underdeveloped but critical to the paper. What mechanisms could change and last at circuit level due to PV cre NphR stim? This has not been explored with any cellular or synaptic metric. Inexpensive experiments could help fill this in.

Also timing and task context is still a major open question that would help contextualize this very curious day 3 effect.

The authors write: To determine if optogenetic inhibition needs to be delivered specifically during the rule shift task in order to elicit lasting effects, we examined rule shift behavior in mice which had previously received optogenetic inhibition of callosal PV projections while performing an intradimensional rule reversal (instead of an extradimensional rule shift) on Day 1. On Day 2, these mice learned an initial association and rule shift in the absence of any additional optogenetic inhibition. In these mice, rule shift performance was normal, i.e., not different from control (opsin-negative) mice (Extended Data Fig. 6). This confirms that terminal inhibition needs to be done specifically during a rule shift to produce a long-lasting deleterious effect”

These 3 reversal mice indeed did undergo stimulation and were able to reverse but it is not clear if they can still shift odor to texture, or texture to odor. Therefore, we STILL do not know if PV colossal inhibition affects shifting even if the inhibition occurs outside the shifting task.

Also “Reviewer figure 1” does not move to look at Day 3 effects.

More minor

I still prefer the plots compare control group data to NpHR data directly. It is an imposition to ask the reader to look back and forth between graphs. Also, the narrative that we get a significant effect in one treatment and no significant change in another treatment is not as convincing as head to head outcome comparison. I would strongly prefer same panel plotting and always to see statistical comparison with both data sets in the model prioritized.

Referee #2 (Remarks to the Author):

In their revision the authors only partially addressed my comments. Although I acknowledge their efforts in providing novel data and discussion, I am bit puzzled that the authors did not further analyze the data they already have to evaluate the robustness of their findings. My main criticism (behavioral, synchrony and vector activity) was related to the lack of detailed temporal analyses. Neuronal synchrony is an exquisite phenomenon allowing firing of individual neurons in phase within short time windows but what is critical is the timing. At the conceptual level, a measure of synchrony computed over 10 seconds of activity sounds meaningless as at lower time scales the precise firing of neurons could be completely uncoupled and neurons might not show any synchronicity in firing activity. It is possible that neuronal synchrony is maintained over long time periods that are relevant from a behavioral perspective, but this is not what is shown in this manuscript.

Conceptually, the manuscript is appealing but the data observed should be further validated using finer temporal resolution to support the authors conclusions.

The main comment that authors did not fully address in their rebuttal is the lack of fine time scale analyses (not just averaging over 10's of seconds). For instance, concerning the behavioral data, what is analyzed is the performance, the latency to perform the choice and the choice bias which are important measures. However, we are missing the overall speed behavior of the animal, the time spent exploring the two bowls, the first move of the animal toward the correct or incorrect choice, for example. The fine scale analysis of behavioral data is critical for the subsequent correlation with physiological markers.

Of course, it is always possible to correlate an overall behavioral performance with physiological markers (PV gamma synchrony, population vector...), but in the particular case of this manuscript, what is expected is to provide a fine scale resolution of the increase in PV synchrony and population vector similarities during behavior, not just a rough estimate based on the averaging of tens of seconds of data. There is indeed a potential bias in the analysis provided by the authors: it is possible that physiological and behavioral data correlate on a large but not on a finer timescale and this is exactly what needs to be shown to support the authors conclusions. As for neuronal synchrony of PV neurons at gamma frequencies, the formal way to demonstrate it is to first demonstrate that individual PV are indeed gamma modulated, then to identify the phase locking of the neurons and finally determine the temporal or phase synchrony. For gamma synchrony, I suggest the authors to show the trace of the PV membrane potential fluctuations for individual mice during the entire sequence of 10 seconds following each choice, correct or incorrect to evaluate the dynamic of this signal. Besides the fact that PV neurons might not be synchronous at lower time frames, it is also possible that the increase in gamma synchrony is driven by specific behavioral events not related to the overall behavioral performance.

Something anecdotal here that could be eventually discussed by the authors is how did they control for a given mouse that the increased in Gamma synchrony is not driven by a very small number of PV cells ?

Concerning calcium imaging data, the authors should analyze the data dynamically in relation to behavior for the same reason mentioned above. They could also, besides calcium transient, determine activity vectors using smaller bin size (200-300 ms) and look at the dynamics of this vectors during the entire 10 second period following the choice. As mentioned in first review, I found these analyses still preliminary. The authors have the data and no additional experiments are required. They should explore the robustness of their findings and how vector activity, gamma synchrony and behavior correlate in lower time frames. I also appreciated the effort to use machine learning analyses as suggested but the authors did not provide the data in the rebuttal (or I missed it), nor evaluate smaller time windows using these decoders.

Overall, I am still not convinced that the observations provided in this manuscript (gamma synchrony, vector population and the correlation between these two metrics and the behavior) reflect a true phenomenon and are not simply due to the timescale at which they are analyzed.

Referee #3 (Remarks to the Author):

I appreciate the additional data, edits and clarifications provided by the authors in response to my comments, as well as those of the other reviewers. The new experiments and analyses have addressed my prior concerns and strengthened the manuscript's conclusions and contributions to understanding the circuit mechanisms of cognitive flexibility. Nice work.

Author Rebuttals to Initial Comments:

We appreciate the Referees' constructive feedback, which has helped us identify areas of potential confusion and greatly improve our manuscript. We have endeavored to be fully responsive. Below, please find our point-by-point responses to the previous reviews. Our responses are in blue. As a brief summary, in addition to resolving several misunderstandings, we have performed the following major new experiments and analyses:

1. As requested by Referee #1, we have confirmed that inhibiting callosal PV⁺ connections disrupts rule shift learning *even when we use a much shorter inter-trial interval* (30 sec instead of ~2 min) (Reviewer Figure 1.1).

Redacted

3. As requested by Referee #2, we have re-analyzed changes in synchrony (Reviewer Figure 2.2) and the similarity of activity patterns using shorter timeframes (Reviewer Figure 2.4), time-locked to specific behavioral events.
4. As requested by Referee #2, we have added additional behavioral measurements (Reviewer Figure 2.1).
5. As requested by Referee #2, we have added additional details about our previous decoding analyses and performed new decoding analyses using smaller time windows (all of which failed to show evidence of meaningful decoding).

Referee #1 (Remarks to the Author):

The response to reviewer 2 argues how important this breakthrough finding may be to the field in terms of understanding basic mechanisms of shifting. This argument made by the authors underscores the need to ensure these effects are generalizable and can be replicated. This remains my major concern, even after the revision. Another major concern is the lack of information about the plasticity effect where a carry over effect is seen on day 3. I am left feeling there are too many uncertainties and unknowns to feel confident about the claims of this paper.

The authors have somewhat strengthened the paper by adding a reversal learning control task. Reversal has been associated with other PFC/OFC areas in previous work, so a negative result is informative and consistent with expectations. However I have two concerns. N was very low (3 mice) and they only reversed odor but not texture. As this is the far less salient of the compound cues, texture should be added as well to ensure the PV colossal NpHR effect is not about moving from a salient cue to a non salient cue or vice versa.

Our response: Both of these concerns – that “N was very low” and “they only reversed odor but not texture” seem to reflect misunderstandings. First, we performed rule reversals in **8 mice** (5 experimental animals plus 3 eNpHR-negative controls), not 3. Rule reversal performance in both groups was similar to what we previously observed in multiple cohorts of mice and multiple publications (Cho et al., *Neuron*, 2015; Cho et al., *Nat Neurosci*, 2020). Second, it is not correct that we only reversed odor but not texture – in 2 experimental mice, the initial association was an odor-based rule, whereas in the other 3 experimental mice it was a texture-based rule. (We apologize if the schematic example caused confusion about this). For convenience, we have shown these data below in Reviewer Figure 1.1. Additionally, we have now plotted cue shifts in Extended Data Figure 4 (i.e., from odor to texture or from texture to odor) to clearly show the distribution of different types of cue shifts across days – for convenience, these plots are shown below in Reviewer Figure 1.2.

REVIEWER FIGURE 1.1 (from Extended Data Figure 6): a, Schematic illustrating the rule reversal task, in which mice chose one of two bowls, each baited by an odor (O1 or O2) and texture (TA or TB) cue, to find a hidden food reward (the stimulus associated with reward is indicated in orange). Mice first learn an initial association (IA) between

either an odor or texture cue (odor O1 in this example) and food reward. Once mice reach the learning criterion (eight correct out of ten consecutive trials), this association undergoes an intra-dimensional rule reversal (RR; e.g., from O1 to O2 in this case). **b**, There is no difference in the number of IA trials needed to reach the learning criterion for control (AAV-DIO-eYFP injected; DIO-eYFP) vs. eNpHR-expressing (AAV-DIO-eNpHR-mCherry injected; DIO-eNpHR) mice across days (two-way ANOVA (task day × virus); interaction: $F_{(1,6)} = 1.127$, $P = 0.3292$). **c**, Experimental design: Day 1, continuous light for optogenetic inhibition of callosal PV terminals during the RR; Day 2, no light delivery during the rule shift (RS). **d**, **e**, Representative images showing viral expression in the mPFC ipsilateral to the injection (ipsi), and labeled callosal terminals in the contralateral mPFC (contra). Scale bars, 250 μm and 100 μm , respectively. **f**, **g**, Performance of eYFP-expressing controls ($n = 3$, **f**) is similar to eNpHR-expressing mice ($n = 5$, **g**) from Day 1 to Day 2 (two-way ANOVA (task day × virus); interaction: $F_{(1,6)} = 0.4286$, $P = 0.5370$). **h**, **i**, Optogenetic inhibition of callosal PV terminals does not change the total number of errors (perseverative and random) in eNpHR-expressing mice ($n = 5$ mice, **i**) compared to eYFP-expressing controls across days ($n = 3$ mice, **h**; two-way ANOVA (task day × virus); interaction: $F_{(1,6)} = 1.095$, $P = 0.3358$). Two-way ANOVA followed by Bonferroni post hoc comparisons was used.

REVIEWER FIGURE 1.2 (from Extended Data Figure 4): d–e, Distribution of cue shifts (odor to texture, texture to odor) across task days in experiments where animals ($n = 17$ eNpHR-negative controls, $n = 19$ DIO-eNpHR-expressing mice) performed two days of the rule shift task (two-way ANOVA (virus × task day); interaction: $F_{(3,64)} = 24.00$, $P < 0.0001$); control odor to texture from Day 1 to optogenetic stimulation on Day 2: Tukey’s post hoc $q_{(64)} = 1.943$, $P = 0.5203$; control texture to odor from Day 1 to optogenetic stimulation on Day 2: Tukey’s post hoc $q_{(64)} = 3.559$, $P = 0.0668$; DIO-eNpHR odor to texture from Day 1 to optogenetic stimulation on Day 2: Tukey’s post hoc $q_{(64)} = 16.16$, $P < 0.0001$; DIO-eNpHR texture to odor from Day 1 to optogenetic stimulation on Day 2: Tukey’s post hoc $q_{(64)} = 13.30$, $P < 0.0001$).

“Reviewer fig 1 ” shows that the long silencing stimulation through multiple phases (outcome and ITI) is critical but it is not clear why this period need be so long. These data should be shown to the reader in the published materials. Also, the figure raises rather than allays my concern that there is something extra (and complex to interpret) about this particular training protocol and its multiple elements. What are the series of cognitive processes that occur and need to be disrupted in series to affect the behavior? Lack of effects by temporally limited stimulation suggests the observations may not generalize to other task formats that lack this extra complexity.

When challenged to adjust this unusual and likely complex manipulation of timeout in my first review the authors responded “Changing this [timeout] duration could obviously open up a can of worms, as some mice would fail to learn the association and the variability of performance would increase dramatically.” I disagree. An alternative way to run this shifting task inside the same cage with shorter timeout and little handling is well established and groups of $n=10-12$ are regularly used to reveal significant effects of brain manipulations on performance in this task. In this more established but less efficient version the animal does not leave the holding cage would still test shift odor to texture or vice versa (the key cognitive construct of interest). Even mice and rats with a bilateral PrL lesion eventually will learn an extradimensional shift task, just with more trials to criterion.

To spell it out clearly, the behavioral change would be more convincing if it were shown in another shifting protocols. Given how highly honed and unusual this task protocol is with the holding cage timeout for 2 minutes, and very low trial times, and very low animal numbers, it is not a given that observed effects will generalize. If the effect were to be lost in a less honed version of the task, then the authors might seek to

understand what aspect of the effective but complex punishment period was disrupted by NpHR stimulation.

Our response: Our initial revision addressed the Referee's concern regarding a possible relationship between duration of optogenetic silencing and working memory, through a combination of arguments and new experiments. Specifically, we found that callosal inhibition has no effect on rule reversals which use the same intertrial interval, indicating that the effects we observed could not be due simply to disruptions of working memory during the intertrial interval (ITI). We have also previously published (and for convenience show below in Reviewer Figure 1.3) the results of an experiment in which we optogenetically inhibited prefrontal interneurons (labeled using *Dlx1/2-Cre* mice) during the initial association. The overall mechanics of the initial association (IA), rule reversal (RR), and rule shift (RS) tasks are identical, and in particular, all three tasks utilize the same duration ITI. The fact that optogenetic inhibition did not affect learning of the IA or RR either across days or in comparison to control animals shows that the impaired performance caused by optogenetic inhibition during the RS is not due simply to mice forgetting the rule during the ITI.

REVIEWER FIGURE 1.3: Optogenetic inhibition of all prefrontal interneurons has no effect on learning of an initial association (IA). From Supplementary Figure 7 in Cho et al., 2015. $n = 4$ experimental and 3 control mice.

In response to the Referee's request, we have now performed **additional experiments which reduce the duration of the ITI following errors to just 30 seconds** (instead of ~2 min). Using this modified protocol, we examined rule shift behavior in mice in the absence of optogenetic inhibition on Day 1 and found that both experimental ($n = 8$ *PV-Cre* mice injected with DIO-eNpHR) and control ($n = 6$ *PV-Cre* mice injected with DIO-eYFP) performed similarly to our previous datasets which used a 2 minute ITI following errors. On Day 2, delivery of optogenetic inhibition during the rule shift portion of the task specifically impaired the experimental group, with no effect on the control group, in accordance with our previous results. Moreover, perseverative behavior in the experimental group persists on Day 3, with no additional optogenetic inhibition. This shows, **using a large number of additional mice, and a new protocol that conforms to the Referee's suggestion**, that callosal PV+ projections are required for learning a rule shift, and that this effect does not depend on the long duration of the intertrial interval following error trials. These data are now shown in Extended Data Figure 4 and included below in Reviewer Figure 1.4. In total, we have now reproduced our finding that inhibiting callosal PV+ projections disrupts rule shifts in a **total of 27 experimental (eNpHR-expressing) mice** (8 mice with behavior only; 5 mice with behavior + TEMPO; 6 mice with behavior + Ca^{2+} imaging; 8 mice using a 30 sec ITI).

REVIEWER FIGURE 1.4 (from Extended Data Figure 4): **p**, Experimental design: Day 1, no light delivery and 30 second ITI; Day 2, continuous light for optogenetic inhibition of callosal PV terminals during the RS with a 30 second ITI; Day 3, no light was delivered and 30 second ITI. **q, r**, Representative image showing viral expression in the mPFC ipsilateral to the injection (ipsi), and labeled callosal terminals in the contralateral mPFC (contra). Scale bar, 100 μ m. **s,w**, Initial association performance with a 30 second ITI in eNpHR-negative mice ($n = 6$) and eNpHR-expressing mice ($n = 8$; two-way ANOVA (task day \times virus); interaction: $F_{(2,24)} = 0.1585$, $P = 0.8543$). **t,x**, Optogenetic inhibition of mPFC callosal PV terminals with a 30 second ITI impairs rule shift performance in DIO-eNpHR-expressing mice ($n = 8$) compared to controls ($n = 6$) (two-way ANOVA (task day \times virus); interaction: $F_{(2,24)} = 50.79$, $P < 0.0001$). **t**, Performance of DIO-eYFP-expressing controls did not change from Day 1 to Day 2 (Tukey's post hoc $q_{(5)} = 1.606$, $P = 0.5356$), Day 1 to Day 3 (Tukey's post hoc $q_{(5)} = 1.035$, $P = 0.7567$), nor Day 2 to Day 3 (Tukey's post hoc $q_{(5)} = 0.4344$, $P = 0.9498$). **x**, Inhibition disrupts rule shift performance in DIO-eNpHR-expressing mice from Day 1 to Day 2 (Tukey's post hoc $q_{(7)} = 15.34$, $P < 0.0001$), Day 1 to Day 3 (Tukey's post hoc $q_{(7)} = 21.75$, $P < 0.0001$), but not Day 2 to Day 3 (Tukey's post hoc $q_{(7)} = 2.679$, $P = 0.2101$). **u, y**, Optogenetic inhibition of callosal PV terminals increases perseverative errors in DIO-eNpHR-expressing mice ($n = 8$ mice) compared to DIO-eYFP-expressing controls ($n = 6$ mice; two-way ANOVA (task day \times virus) interaction: $F_{(2,24)} = 19.79$, $P < 0.0001$). **v, z**, Optogenetic inhibition of callosal PV terminals has no effect on random errors (two-way ANOVA (task day \times virus); interaction: $F_{(2,24)} = 1.079$, $P = 0.3559$). **u, v**, Light delivery does not affect the number of perseverative (post hoc $t_{(5)} = 0.000 - 1.000$, $P > 0.9999$) or random (post hoc $t_{(5)} = 0.4152 - 0.7906$, $P > 0.9999$) errors in DIO-eYFP-expressing controls across days. **y, z**, Optogenetic inhibition of callosal PV terminals on Day 2 increased the number of perseverative (post hoc $t_{(5)} = 0.8885$, $P = 0.0016$ from Day 1 to Day 2; post hoc $t_{(5)} = 1.000$, $P = 0.0019$ from Day 1 to Day 3; post hoc $t_{(5)} = 0.000$, $P = 0.9093$ from Day 2 to Day 3) but not random (post hoc $t_{(7)} = 0.000 - 2.198$, $P = 0.1918 - > 0.9999$) errors compared to no stimulation across days. Two-way ANOVA followed by Bonferroni post hoc comparisons was used unless otherwise noted.

We also address the Referee's concern regarding our original Reviewer Figure 1, whereby a 'lack of effects by temporally limited stimulation' suggests there may be something happening during the intertrial interval that can contribute to learning. We believe that increases in gamma synchrony which drive learning can occur both during the post-decision period and the intertrial interval (increased gamma synchrony during the ITI was shown in Extended Data Figure 4L of Cho et al., *Nat Neurosci*, 2020). For clarity, here we have focused on analyzing activity patterns and gamma synchrony during the post-decision period before the ITI. In particular, because the ITI is so long, it is possible that some changes occur during specific parts of the ITI but not others, making it difficult to know what part of the ITI to focus on. Nevertheless, we have now added further analyses to Extended Data Figure 9a-d confirming our previous finding that interhemispheric gamma synchrony increases during ITIs following rule shift errors, relative to ITI following correct decisions or correct and incorrect decisions during the initial association. Furthermore, this increase does not occur when callosal PV+ connections are inhibited (on Day 2 in eNpHR-expressing mice). For convenience, these results are also shown below in Reviewer Figure 1.5. Together, with our new experiments showing that inhibiting callosal PV projections disrupts rule shift performance even when we use much shorter ITIs, these data should hopefully allay the Referee's concerns, demonstrate the generalizability of our findings across different task protocols, and validate that RS error-driven increases in gamma synchrony may occur during multiple time periods but consistently depend on callosal PV+ projections.

REVIEWER FIGURE 1.5 (from Extended Data Figure 9): **a**, Experimental design: Day 1, no optogenetic inhibition; Day 2, continuous light for optogenetic inhibition during the rule shift (RS); Day 3, no optogenetic inhibition (light for TEMPO was delivered on all days). **b–e**, Gamma synchrony during the intertrial interval (ITI). **b–c**, Gamma synchrony during ITIs does not change following correct or error trials during the initial association (IA) in either control ($n = 4$ mice; two-way ANOVA (trial type \times task day); interaction: $F(2,16) = 2.407$, $P = 0.1219$) or eNpHR-expressing mice ($n = 5$ mice; two-way ANOVA (trial type \times task day); interaction: $F(2,12) = 1.750$, $P = 0.2154$) across days. **d**, Gamma synchrony during ITIs is higher after rule shift (RS) errors than after RS correct choices across days in control mice (Day 1: post hoc $t(9) = 2.972$, $P = 0.047$; Day 2: post hoc $t(9) = 2.964$, $P = 0.0476$; Day 3: post hoc $t(9) = 4.020$, $P = 0.0091$). **e**, In eNpHR-expressing mice, gamma synchrony during ITIs is higher after RS errors than after RS correct choices on Day 1 (Day 1: post hoc $t(12) = 2.914$, $P = 0.0390$), but not on Day 2 when mice receive optogenetic inhibition of callosal PV+ terminals (post hoc $t(12) = 1.041$, $P = 0.9545$) nor on Day 3 (post hoc $t(12) = 1.153$, $P = 0.8136$). Two-way ANOVA followed by Bonferroni post hoc comparisons was used. * $P < 0.05$, ** $P < 0.01$.

Plasticity and day 3 effects are still underdeveloped but critical to the paper. What mechanisms could change and last at circuit level due to PV cre NpHR stim? This has not been explored with any cellular or synaptic metric. Inexpensive experiments could help fill this in.

Redacted

Also timing and task context is still a major open question that would help contextualize this very curious day 3 effect.

The authors write: To determine if optogenetic inhibition needs to be delivered specifically during the rule shift task in order to elicit lasting effects, we examined rule shift behavior in mice which had previously received optogenetic inhibition of callosal PV projections while performing an intradimensional rule reversal (instead of an extradimensional rule shift) on Day 1. On Day 2, these mice learned an initial association and rule shift in the absence of any additional optogenetic inhibition. In these mice, rule shift performance was normal, i.e., not different from control (opsin-negative) mice (Extended Data Fig. 6). This confirms that terminal inhibition needs to be done specifically during a rule shift to produce a long-lasting deleterious effect”

These 3 reversal mice indeed did undergo stimulation and were able to reverse but it is not clear if they can still shift odor to texture, or texture to odor. Therefore, we STILL do not know if PV colossal inhibition affects shifting even if the inhibition occurs outside the shifting task.

Our response: In the rule reversal task, as well as for the rule shift task, mice are randomly assigned cues and the following shifts of odor to texture and texture to odor are balanced across experimental and control groups. As noted above, there were 5 experimental mice in the rule reversal task, and 2/5 shifted odor to texture while 3/5 shifted texture to odor. None of these mice were impaired on either type of shift by prior callosal inhibition occurring outside the context of a rule shift. We have additionally plotted cue shifts (i.e., from odor to texture or from texture to odor) to demonstrate distribution of cue shifts across days in Extended Data Figure 4 and Reviewer Figure 1.2.

Also “Reviewer figure 1” does not move to look at Day 3 effects.

Our response: These data were from an experiment performed long ago (in 2016), well before we began the current study. As described above, we have now performed additional experiments, using a shorter ITI, to further allay the Referee’s concerns about this aspect of the experiment.

More minor

I still prefer the plots compare control group data to NpHR data directly. It is an imposition to ask the reader to look back and forth between graphs. Also, the narrative that we get a significant effect in one treatment and no significant change in another treatment is not as convincing as head to head outcome comparison. I would strongly prefer same panel plotting and always to see statistical comparison with both data sets in the model prioritized.

Our response: We appreciate the further clarification from the Referee here. There are really two issues – one about the plotting and the other about the statistical comparisons. What makes both of them complicated is that we often have three groups (e.g., control, DIO-eNpHR, and synapsin-eNpHR), three different experimental days, and in some cases, two different trial types (correct vs. error) as well. Below, we show examples of what Figures 3 and 4 would look like if we plotted the different groups on top of each other – we find it very difficult to discern trends within these overlaid plots when we also show data points from individual animals. But at this point if the Referee feels strongly about this, we would certainly oblige.

That being said, we do fully appreciate where the Referee is coming from, especially with regard to the statistics. We agree that claims are more convincing when head-to-head outcome comparisons are included. Indeed, for the TEMPO measurements shown in Figure 3, we have performed two-way ANOVA with appropriate post-hoc tests corrected for multiple comparisons which confirm our original findings. Specifically, gamma synchrony is lower for DIO-eNpHR mice compared to matched controls on error but not correct trials on both Days 2 ($p = 0.0089$) and 3 ($p = 0.0160$). Gamma synchrony is also lower for synapsin-eNpHR mice compared to matched controls on error but not correct trials on Day 2 ($p = 0.0028$) (but not Day 3, $p = 0.4909$). These head-to-head statistical comparisons have been added to the revised

manuscript in the legend for Figure 3. (Note that the original manuscript did contain head-to-head statistical comparisons of gamma synchrony in the same group on different days).

For Ca^{2+} imaging data, there is greater mouse-by-mouse variability in the baseline (Day 1) activity vector similarity, necessitating a slightly more complex analysis which accounts for this between-mouse variability. We performed a two-way ANOVA we compared the change in activity vector similarity from Day 1 to Day 2 in the DIO-eNpHR group to a comparison group consisting of the eNpHR-negative and synapsin-eNpHR mice, using mouse, day, and day \times group interaction as factors. This revealed a significant day \times group interaction ($p = 0.040$). This comparison has been added to the legend for Figure 4. Furthermore, Referee 2 had requested a finer timescale analysis of these data, e.g., how does the activity vector similarity evolve over time in relation to the start of digging? When we analyzed the difference between activity vector similarity on Day 1 vs. Day 2 across these timepoints (shown in Reviewer Fig. 2.3 and Extended Data Figure 10), ANOVA (using timepoint and group as factors) revealed a significant difference between DIO-eNpHR, eNpHR-negative (control), and synapsin-eNpHR groups ($p = 0.0306$). Furthermore, post hoc tests corrected for multiple comparisons revealed a significant difference between these groups at specific timepoints. This information has been added to the legend for Extended Data Figure 10.

Example of plotting groups together for Figure 3:

Example of plotting groups together in Figure 4:

Referee #2 (Remarks to the Author):

In their revision the authors only partially addressed my comments. Although I acknowledge their efforts in providing novel data and discussion, I am bit puzzled that the authors did not further analyze the data they already have to evaluate the robustness of their findings. My main criticism (behavioral, synchrony and vector activity) was related to the lack of detailed temporal analyses. Neuronal synchrony is an exquisite phenomenon allowing firing of individual neurons in phase within short time windows but what is critical is the timing. At the conceptual level, a measure of synchrony computed over 10 seconds of activity sounds meaningless as at lower time scales the precise firing of neurons could be completely uncoupled and neurons might not show any synchronicity in firing activity. It is possible that neuronal synchrony is maintained over long time periods that are relevant from a behavioral perspective, but this is not what is shown in this manuscript.

Our response: The Referee's comment that "At the conceptual level, a measure of synchrony computed over 10 seconds of activity sounds meaningless as at lower time scales the precise firing of neurons could be completely uncoupled and neurons might not show any synchronicity in firing activity," suggests a simple misunderstanding about what synchrony means here and how we calculated it. Specifically, the Referee's statement seems to conflate the time window over which a metric of synchrony is averaged (here 10 seconds), with the timescale over which co-activity must occur to be reflected in that metric (milliseconds). We apologize if this distinction was unclear. As was stated in the original manuscript, we filtered signals in the 30-50 Hz band before computing correlations to measure synchrony. That means that synchrony will necessarily reflect activity in the two hemispheres that co-occurs within a narrow window (less than 1/2 of a cycle which corresponds to ~10-15 msec). This is because correlation measures the instantaneous overlap of activity, and filtering removes fluctuations on slower timescales while smoothing out fluctuations on faster ones. Once we have calculated this measure of synchrony, we then average the calculated values over a much longer time window (e.g., 10 seconds). Neither the time window over which the correlation is calculated, nor the time period over which it is averaged, in any way changes the fact that this synchrony metric specifically reflects co-activity occurring within a timescale shorter than 10-15 msec. In other words, it is not true that "at lower time scales the precise firing of neurons could be completely uncoupled and neurons might not show any synchronicity in firing activity." Rather, based on how we performed our calculations, our metric will only show synchrony when activity in PV neurons in the left mPFC occurs within milliseconds of activity in PV neurons within the right mPFC. In particular, in response to one of the Referee's other comments, we have now added examples of the filtered voltage indicator traces to Extended Data Figure 9j-l – our metric of synchrony is based on the correlation between these filtered (and 'cleaned') traces. Thus, this metric is positive when peaks and troughs in these traces are aligned and overlap to a meaningful extent. Because the frequency range is 30-50 Hz, the width of these peaks and trough at half maximum will be ~4-8 msec. Thus, this is the timescale on which co-activity must occur to be reflected in our synchrony metric.

Conceptually, the manuscript is appealing but the data observed should be further validated using finer temporal resolution to support the authors conclusions.

The main comment that authors did not fully address in their rebuttal is the lack of fine time scale analyses (not just averaging over 10's of seconds). For instance, concerning the behavioral data, what is analyzed is the performance, the latency to perform the choice and the choice bias which are important measures. However, we are missing the overall speed behavior of the animal, the time spent exploring the two bowls, the first move of the animal toward the correct or incorrect choice, for example. The fine scale analysis of behavioral data is critical for the subsequent correlation with physiological markers.

Our response: In their original review, this Referee simply said, "I found overall that the behavioral data should be described in much more details." In response to this request, we added the latency to dig (aka the 'time to choice') and the fraction of trials in which the mouse chose the bowl that was ipsilateral vs. contralateral to the side of optogenetic inhibition. Now that the Referee has made several more specific requests, we can easily provide the requested information. The overall speed of the animal, the time spent exploring the two bowls, and whether the first move of the animal is towards the correct or incorrect choice are all shown below in Reviewer Figure 2.1 and have been added the manuscript in

Extended Data Figure 4. (This behavioral analysis was performed specifically for the mice included in the Ca²⁺ imaging experiments).

Furthermore, we have attempted to re-analyze much of our data on finer timescales in relation to behavior as requested by the Referee. Before presenting these results, it is important to think about what the appropriate timescale on which to analyze our data is. In principle, we could always analyze data on shorter and shorter timescales, but in practice, the choice of meaningful timescales is limited by multiple factors. One is the resolution with which we can score behavior. Specifically, in this task, it is unlikely we can identify specific behavioral timepoints with a precision much finer than ~0.5-1 sec. A second factor is the timescale of the behavior itself. E.g., the time between the start of a trial and the time of digging ranges between a few seconds and tens of seconds. Similarly, the time between the start of digging (signifying a choice) and the end of digging (signifying the end of a trial) ranges between a few seconds and tens of seconds. Both this degree of variability and the duration of different behavioral epochs are in marked contrast to operant or head-fixed behaviors, in which task-related cues are presented for very precise (and typically much shorter) intervals, and which correspondingly elicit behavioral responses with much shorter and more precise latencies. By contrast, in our more naturalistic task, behavior occurs on a timescale of seconds to tens of seconds. In this context, it is difficult to imagine how a behaviorally-relevant neural signal could appear on timescales that are much more precise or transient than ~1 second. Even if there was such a signal, given the trial-to-trial variability in the duration of behavioral epochs, it seems likely that upon averaging, such a signal would be effectively smeared out over multiple seconds. This is why we originally analyzed our signals over an interval instead of at precise timepoints. However, we do agree with the Referee that it is possible these signals will manifest on timescales shorter than 10 sec, and in light of the preceding context, have attempted to analyze them on timescales of 1 second. There is of course a third factor which also limits the usefulness of examining signals on ever shorter timescales, which is that when we do so, we are relying on fewer experimental measurements to perform our analyses. This will be an issue when measuring synchrony using genetically-encoded voltage indicators, because these have inherently low signal-to-noise and require averaging to make meaningful observations. Furthermore, they will make our analysis of activity vectors more vulnerable to the sparse nature of activity measured using Ca²⁺ imaging (see below).

Reviewer Figure 2.2, which is now included in the manuscript as part of Extended Data Figure 9, shows the difference in our gamma synchrony metric from Day 1 to Day 2 (the cross-correlation between 'cleaned' 30-50 Hz filtered Ace-mNeon signals from PV interneurons in the left and right mPFC) calculated in 1 second windows relative to specific behavioral events for correct (open symbols) or incorrect (filled symbols) rule shift (RS) trials. We have performed this analysis for three behavioral events: trial start, dig start, and dig end. Overall, these traces are very noisy, highlighting the challenge noted above – that low signal-to-noise makes it necessary to average over multiple timepoints in order to obtain robust measurements. To the extent that this higher temporal resolution (but much noisier) analysis yields interpretable information, it may just be to identify the (approximate) timepoints which drive our main finding: that gamma synchrony normally (on Day 1) rises after incorrect RS trials, but this is abolished on Day 2 when we optogenetically-inhibit callosal projections in eNpHR-expressing mice. These plots now suggests that this deficit in gamma synchrony following incorrect RS trials specifically appears a few seconds after the beginning of digging in eNpHR-expressing mice. A similar deficit can also be seen around the end of digging. This timing seems reasonable and experimentally plausible in light of the other behavioral timescales noted above and the fact that this change in gamma synchrony reflects the animal's realization that the chosen bowl does not contain a hidden food reward (and this realization can only occur after the mouse digs through the bowl's contents).

REVIEWER FIGURE 2.1 (from Extended Data Figure 4): j-k, The overall speed (meters per second) of mice during the first 5 IA and RS trials across days in the miniscope experimental dataset ($n = 6$ eNpHR-negative controls; $n = 6$

DIO-eNpHR-expressing mice; control two-way ANOVA (type of task × task day); interaction: $F_{(1,10)} = 0.00271$, $P = 0.9595$; DIO-eNpHR-expressing mice two way-ANOVA (type of task × task day); interaction: $F_{(1,10)} = 1.378$, $P = 0.2677$). **l–m**, There is no difference in the amount of time (seconds) mice spent exploring bowls before making a decision during the first 5 IA and RS trials across days in the miniscope experimental dataset ($n = 6$ eNpHR-negative controls; $n = 6$ DIO-eNpHR-expressing mice; control two-way ANOVA (type of task × task day); interaction: $F_{(1,10)} = 0.5053$, $P = 0.4934$; DIO-eNpHR-expressing mice two way-ANOVA (type of task × task day); interaction: $F_{(1,10)} = 0.1147$, $P = 0.7419$). **n–o**, The first move of the mouse toward the correct bowl (percent) during the first 5 IA and RS trials across days in the miniscope experimental dataset ($n = 6$ eNpHR-negative controls; $n = 6$ DIO-eNpHR-expressing mice; control two-way ANOVA (type of task × task day); interaction: $F_{(1,10)} = 3.347$, $P = 0.0973$; DIO-eNpHR-expressing mice two way-ANOVA (type of task × task day); interaction: $F_{(1,10)} = 0.1316$, $P = 0.7244$).

REVIEWER FIGURE 2.2 (from Extended Data Figure 9): g–i, We performed a shorter-timeframe re-analysis of the TEMPO data collected from *PV-Cre*; Ai14 injected in one mPFC with AAV-DIO-eNpHR to identify specific times when optogenetic inhibition of callosal PV+ projections disrupts gamma synchrony. We measured the change in gamma synchrony (calculated in 1 second windows for various timepoints relative to behavioral events) between Day 1 (control) and Day 2 (optogenetic inhibition) for both errors (filled circles) and correct choices (open circles) during the first 5 RS trials in *PV-Cre* mice expressing DIO-eNpHR. **g**, This change in gamma synchrony (change = Day 2 – Day 1) is not significantly different for error vs. correct trials around the time of trial start (two-way ANOVA (timepoint × error vs. correct); interaction: $F_{(15,64)} = 1.642$, $P = 0.0874$). **h**, This change in gamma synchrony is more negative for error vs. correct trials around the start of digging (two-way ANOVA (timepoint × error vs. correct); interaction: $F_{(15,64)} = 1.931$, $P = 0.0360$). **i**, This change in gamma synchrony is more negative for error vs. correct trials around the end of digging (two-way ANOVA (timepoint × error vs. correct); interaction: $F_{(10,44)} = 2.096$, $P = 0.0454$). Two-way ANOVA followed by Bonferroni post hoc comparisons was used. * $P < 0.05$.

Of course, it is always possible to correlate an overall behavioral performance with physiological markers (PV gamma synchrony, population vector...), but in the particular case of this manuscript, what is expected is to provide a fine scale resolution of the increase in PV synchrony and population vector similarities during behavior, not just a rough estimate based on the averaging of tens of seconds of data.

Our response: First, the comment that “Of course, it is always possible to correlate an overall behavioral performance with physiological markers (PV gamma synchrony, population vector...)” is not correct in the context of what we have shown. It is true that finding one correlation is ‘always possible’ and does not on its own establish a strong link between the behavioral effect and physiological marker. I.e., just because we see decreased gamma synchrony or increased population vector similarity when we inhibit callosal PV+ projections and disrupt rule shift performance does not imply a strong relationship between these physiological markers and the behavioral effect. However, we further see that both of these markers persist along with impaired rule shift performance on the day after callosal PV inhibition. Furthermore, neither of these markers persist the day after inhibiting all callosal synapses (when behavior is normal). Thus, while finding a single correlation is ‘always possible,’ this type of three-fold correlation is not. Rather, this type of three-fold correlation is strong evidence for a relationship between these physiological markers and the behavioral effect.

Second, regarding the request for a ‘fine scale resolution of the increase in PV synchrony and population vector similarities during behavior’, we are happy to oblige. We analyzed the population vector similarities in 1 second time windows relative to the trial start, dig start, and dig end timepoints. This is now shown below as Reviewer Figure 2.3 and included in the manuscript within Extended Data Figure 10. (Note that the sparsity of activity measured using Ca^{2+} imaging prevents us from analyzing activity on ever-finer timescales. Specifically, even using 1 second bins, some mice have occasional bins without

activity, necessitating the exclusion of some trials – an issue which becomes even worse at smaller bin sizes. The smallest bin size for which this is not an issue, i.e., does not result in excluding any trials from this analysis, is 2.5 seconds. Using 2.5 second windows yields results similar to what we observed using 1 second windows (and if the Referee preferred, we could switch to this window size).

In addition, to quantify how activity evolves over the course of each RS error trial, we measured the similarity between population activity vectors (computed using a 1 second window) occurring at the beginning of the 10 second period following the start of digging, and those at the end of this period on the same trial, for each mouse on Day 1 versus Day 2. As shown below in Reviewer Figure 2.4 (and included in the manuscript in Extended Data Figure 10), on Day 2 these patterns at the beginning and end of each post-dig period on RS error trials become more similar. Thus, inhibiting callosal PV+ projections also disrupts the evolution of fast timescale activity patterns that normally occurs over the course of a single trial.

REVIEWER FIGURE 2.3 (from Extended Data Figure 10): **v**, There is a main effect of virus (two-way ANOVA; $F_{(2,285)} = 16.95$; $P < 0.0001$) between groups in the change in the population activity vector similarity from Day 1 to Day 2 around the time of trial start. Tukey's post hoc tests reveal statistically significant differences at specific timepoints. **w**, There is a difference between groups (main effect of virus) for the change in population vector similarities following the start of digging (two-way ANOVA; $F_{(2,285)} = 23.86$, $P < 0.0001$). Tukey's post hoc tests reveal statistically significant differences at specific timepoints. **x**, There is a difference between groups (main effect of virus, two-way ANOVA; $F_{(2,285)} = 11.63$, $P < 0.0001$) for the change in population vector similarities around the end of digging (two-way ANOVA; main effect of virus: $F_{(1,190)} = 9.696$, $P = 0.0021$). Tukey's post hoc tests reveal statistically significant differences at specific timepoints. Two-way ANOVA followed by Tukey post hoc comparisons was used unless other noted. * $P < 0.05$, ** $P < 0.01$.

REVIEWER FIGURE 2.4 (From Extended Data Figure 10): To quantify how activity evolves over the course of each RS error trial, we measured the similarity between population activity vectors (computed using a 1 second window) occurring at the beginning of the 10 second period following the start of digging, and those at the end of this period on the same trial, for each mouse on Day 1 versus Day 2. **q**, The similarity of activity patterns from the beginning to end of the post-dig period on RS error trials does not change from Day 1 to 2 in controls (two-tailed paired t -test, $t_{(5)} = 1.332$, $P = 0.2403$; $n = 6$); **r**, however this similarity increases from Day 1 to 2 for DIO-eNpHR-expressing mice (two-tailed paired t -test, $t_{(5)} = 3.921$, $P = 0.0112$; $n = 6$ mice); **s**, this similarity does not differ from Day 1 to 2 for Syn-eNpHR-expressing mice (two-tailed paired t -test, $t_{(5)} = 0.9724$, $P = 0.3755$; $n = 5$ mice).

There is indeed a potential bias in the analysis provided by the authors: it is possible that physiological and behavioral data correlate on a large but not on a finer timescale and this is exactly what needs to be shown to support the authors conclusions. As for neuronal synchrony of PV neurons at gamma frequencies, the formal way to demonstrate it to first demonstrate that individual PV are indeed gamma

modulated, then to identify the phase locking of the neurons and finally determine the temporal or phase synchrony. For gamma synchrony, I suggest the authors to show the trace of the PV membrane potential fluctuations for individual mice during the entire sequence of 10 seconds following each choice, correct or incorrect to evaluate the dynamic of this signal. Besides the fact that PV neurons might not be synchronous at lower time frames, it is also possible that the increase in gamma synchrony is driven by specific behavioral events not related to the overall behavioral performance.

Our response: Again, as noted above, there seems to be a misunderstanding about what it means for a measure of synchrony to be calculated for specific frequencies and then averaged over longer time windows. The specific frequency band over which the signal is filtered before we compute the correlation determines timescale over which coactivity / correlated activity must occur. Furthermore, the requirement for coactivity to occur on this timescale (determined by the filtering) is not altered by the time window used to compute the correlation (rather this determines the number of cycles used to determine if phase or amplitude relationships are consistent), nor by subsequent averaging.

Re: 'the trace of the PV membrane potential fluctuations for individual mice during the entire sequence of 10 seconds following each choice, correct or incorrect' – for this, we would need to show traces from individual mice on individual trials. (Because activity is not phase-locked from trial to trial, simply averaging these traces across trials or mice would eliminate any rhythmic fluctuations, even if we filtered within a frequency band of interest before averaging). Given the low signal-to-noise ratio for voltage imaging, traces from single trials are not particularly informative. Nevertheless, we have tried to use example traces to illustrate the basic dynamics of activity measured using voltage indicators, and our method for computing short-timescale synchrony. Specifically, Extended Data Figure 9j-l, shown below as Reviewer Figure 2.5, now shows examples traces of raw (unfiltered) Ace-mNeon signals from each hemisphere 0–10 and 6–7 seconds after dig start on a correct and incorrect RS trial from the same mouse, along with the filtered and 'cleaned' versions of these signals. This figure also shows the timecourse of the correlation computed from these signals on these two example trials. As noted above, hopefully this makes clear the nature of these signals, and illustrates the principle that our synchrony metric (correlation) is based on the degree to which rhythmic peaks and troughs in the filtered signals overlap, which necessarily occurs on timescales of a few milliseconds.

REVIEWER FIGURE 2.5 (From Extended Data Figure 9): j, Example traces of left (L) and right (R) Ace2n-4AA-mNeon (Ace-mNeon) traces (green and red, respectively), from 0–10 seconds after dig start on RS correct and error trials in a DIO-eNpHR-expressing mouse and k, zoomed in on the period 6–7 seconds after dig start. l, The timecourse of gamma synchrony (correlation values calculated from filtered and cleaned Ace-mNeon signals) from 0–10 seconds after dig start for these two example trials.

Something anecdotal here that could be eventually discussed by the authors is how did they control for a given mouse that the increased in Gamma synchrony is not driven by a very small number of PV cells ?

Our response: We are making bulk measurements of fluorescence. As a result, we cannot directly infer the number of cells driving increases in synchrony. However, we observe that our measures

of synchrony increase by ~50%. It is therefore unlikely that these changes are driven by only a small fraction of cells. We have added discussion of this point on lines 222-225 on page 10 of the revised manuscript.

Concerning calcium imaging data, the authors should analyze the data dynamically in relation to behavior for the same reason mentioned above. They could also, besides calcium transient, determine activity vectors using smaller bin size (200-300 ms) and look at the dynamics of this vectors during the entire 10 second period following the choice. As mentioned in first review, I found these analyses still preliminary. The authors have the data and no additional experiments are required. They should explore the robustness of their findings and how vector activity, gamma synchrony and behavior correlate in lower time frames.

Our response: This is similar to previous comments. We have now added plots for the timecourse of gamma synchrony and population activity vector similarity relative to the start of each trial, to the start of digging and the end of digging on fine timescales (1 second) in Extended Data Figures 9 and 10.

I also appreciated the effort to use machine learning analyses as suggested but the authors did not provide the data in the rebuttal (or I missed it), nor evaluate smaller time windows using these decoders.

Our Response: As requested, we will present the results of our original decoding analysis, as well as a new analysis using smaller time windows. To begin with, consider what variables can be decoded from the type of data that is available. Under normal (unperturbed) conditions, mice learn both the initial association and rule shift rapidly, e.g., they reach the learning criterion within ~12-15 trials, making only ~1-3 errors. This means one cannot really decode whether a trial is correct vs. incorrect, because in many cases there might be only 1 error during the rule shift or initial association. Similarly, one could decode task phase (rule shift vs. initial association), however it would be unclear whether decoding of this variable reflects task-related changes in patterns of activity vs. just the passage of time. By contrast, the location of cues (odors, textures) in the left vs. right bowl varies randomly from trial to trial. So too does the location of the reward. Therefore, each block of trials contains an approximately 50/50 split of cue locations, reward locations, and choices (left vs. right), making these variables well-suited to a decoding analysis (because there will be sufficient numbers of trials exemplifying both conditions for training). Originally, this decoding analysis used all of the individual frames (sampled every 50 msec) occurring during a window of up to 10 seconds during either the **pre-dig** period, the **post-dig** period, the period immediately preceding the **end of digging**, or the initial portion of the **inter-trial interval** (ITI), i.e., we built a separate classifier for each of these trial periods. We trained a neural network classifier, which we previously described (Frost et al., *PLoS Biology*, 2020), on the last 10 initial association trials, and measured its performance on the first 8 rule shift trials. (Of note: Frost et al. showed that this type of neural network performs similarly to optimal linear classifiers when information is encoded via activity levels and significantly outperforms optimal linear classifiers when information is encoded via patterns of correlated activity / coactivity). We hypothesized that task-related activity might include representations for these task variables, and that these representations might be abnormally stabilized or de-stabilized when we inhibit callosal projections and elicit perseveration. The tables below summarize the classifier performance (averaged over 6 mice) for each of these variables and trial periods for real (left) and shuffled (right) data on Day 1 (no inhibition of callosal projections) and Day 2 (optogenetic inhibition of callosal projections):

Day 1 (no opto- inh)	real data				shuffled data			
	reward loc	chosen side	IA cue	RS cue	reward loc	chosen side	IA cue	RS cue
pre-dig	0.471503	0.483176	0.482076	0.484916	0.491402	0.49697	0.496494	0.504647
post-dig	0.472541	0.46864	0.487037	0.496363	0.487463	0.481036	0.50024	0.508059
dig end	0.504619	0.495876	0.48729	0.504375	0.512072	0.48852	0.475049	0.487473
ITI	0.478116	0.465234	0.494562	0.480817	0.481968	0.458494	0.490049	0.473178
Day 2 (+ opto- inh)	real data				shuffled data			
	reward loc	chosen side	IA cue	RS cue	reward loc	chosen side	IA cue	RS cue

pre-dig	0.489276	0.478709	0.495418	0.479542	0.506503	0.476111	0.486624	0.477218
post-dig	0.519143	0.481858	0.502379	0.501167	0.511597	0.483846	0.502728	0.505024
dig end	0.528674	0.496856	0.482803	0.470897	0.508797	0.503675	0.49673	0.486444
ITI	0.557622	0.532423	0.504556	0.567769	0.535792	0.535752	0.507625	0.571928

As can be seen from these tables, classifier performance is not meaningfully higher for real data than shuffled data for any of these variables or trial periods on either Day 1 or Day 2. This negative result (despite examining 32 different cases!) illustrates why we do not believe this is an informative way to analyze our data. In particular, we do not understand why this type of analysis, involving more extensive processing of the data and many additional comparisons, would be seen as superior to our original activity vector-based analysis, which showed a strong relationship to behavior.

In response to the Referee's more specific request, we have now attempted to decode the same variables, but using much smaller time windows, and performing the analysis on a trial-by-trial basis. I.e. instead of training a single decoder using all the relevant frames during a given trial period (e.g., pre-dig) and the last 10 IA trials, now for each **1 second window and 0.5 second increment from -5 to +10 seconds relative to the start of digging** (i.e., when the mouse makes a choice), we trained a classifier based on frames located within that window (or the two adjacent 1 second windows) on the preceding 5 trials. We did this for the **4 trials preceding, and 8 trials following, the first error during the RS**. We performed this classification using the matlab function 'fitlinear' for two reasons. First, this new analysis involves training an extremely large number of classifiers, for which 'fitlinear' is much faster. Second, for the neural network classifier, we only use frames with at least 3 active neurons, and many 1 second windows contained a very small number of frames satisfying this criterion. We calculated classifier performance for the original dataset and for **10 shuffled datasets**. In addition to the reward location, chosen side, IA cue location, and RS cue location, we also attempted to decode whether the IA and RS cue were located in the same or different bowls.

This results in an extremely large number of datapoints – for 12 datasets (6 Day 1 datasets, and 6 Day 2 datasets) and 5 variables to be decoded, we obtain the classifier performance at each of 13 trials and 27 timepoints = 21,840 classifier performances based on real data. If we convert each of these classifier performances to a z-score based on the mean and standard deviation of performances obtained from shuffled datasets, then average across all control datasets, or all experimental datasets, we obtain $13 \times 27 \times 5 \times 2 = 3,510$ z-scores. Remarkably, **none** (0 / 3,510) of these average z-scores exceed 1.5, indicating that we do not find that any of these variables are consistently encoded on any trials or timepoints. Even if we reduce this z-score threshold to 1, i.e., look for cases where classifier performance exceeds the performance obtained using shuffled data by 1 standard deviation on average, there are only 25 / 3,510 such cases. By contrast, there are 89 / 3,510 cases where the classifier performance is on average more than 1 standard deviation *below* the average performance obtained using shuffled data. This indicates that the sparse, isolated cases where average classifier performance is ~1 standard deviation above the average performance obtained using shuffled data simply reflects chance variation. Thus, even using much smaller (1 second time windows), a time resolution of 0.5 sec, and a trial-by-trial analysis, we do not see evidence for meaningful encoding of these variables.

Overall, I am still not convinced that the observations provided in this manuscript (gamma synchrony, vector population and the correlation between these two metrics and the behavior) reflect a true phenomenon and are not simply due to the timescale at which they are analyzed.

Our response: Again, there seems to have been a misunderstanding about the timescales over which synchrony was measured – our metrics reflect the co-occurrence of activity on timescales ~4-8 msec even if they are subsequently averaged over longer timescales. Furthermore, we have now measured both synchrony and the similarity of population activity vectors on much shorter timeframes (specifically using 1 second windows – as explained above, it is problematic to analyze these data on even shorter timeframes). These analyses validate our original results and show that these phenomena are time-locked to behavior. In particular, both the callosal PV⁺ projection-dependent changes in gamma

synchrony and the similarity of activity patterns both occur a few seconds after the onset of digging, which signifies a choice and leads to the mouse receiving feedback about whether its choice was correct or incorrect. Furthermore, we found that inhibiting callosal PV⁺ projections disrupts the normal evolution of fast timescale activity patterns over the course of a single trial. Finally, in response to requests from Referee 1, we have now found that inhibiting callosal PV⁺ projections does not affect learning during a rule reversal task, and replicated our finding that this inhibition does disrupt rule shift learning even when we use a much shorter inter-trial interval (30 sec instead of ~2 min). These experiments further strengthen the specific causal relationship between this callosal PV⁺ circuit and rule shift behavior.

Referee #3 (Remarks to the Author):

I appreciate the additional data, edits and clarifications provided by the authors in response to my comments, as well as those of the other reviewers. The new experiments and analyses have addressed my prior concerns and strengthened the manuscript's conclusions and contributions to understanding the circuit mechanisms of cognitive flexibility. Nice work.

Reviewer Reports on the Second Revision:

Referees' comments:

Referee #1 (Remarks to the Author):

The authors have argued in the past rebuttal how important this breakthrough finding may be to the field in terms of understanding basic mechanisms of shifting. This argument made by the authors underscores the need to ensure these effects are generalizable and can be replicated.

We still don't know if this stimulation effect works in any other task context. The authors perhaps misunderstood my request to use a different task. They did add an experiment which shortened the time-out in response to my concerns. However, they did not test more commonly established shifting task designs to strengthen their case that theirs is a robust finding. The current behavioral paradigm (at 2 minutes or 30 seconds) still relies on removing the mouse from its homecage and leaving it out for sometime after an error. This handling procedure inside of a behavioral test of prefrontal function is highly invasive and may operate through special punitive effects that are custom to the Sohal lab and their favorite protocols but not replicated in other more standard protocols. Homecage testing and removal from the test cage in the task testing time is not typically used in rodent protocols testing prefrontal function. I am concerned this removal and possibly also use of the homecage as the test cage are possibly critical features of the protocol that have gone underrecognized. These details are also hard to notice in the streamlined version of the paper and past methods published for this group. Removal is used in water maze testing hippocampal function but these time out periods are not when measures of brain function are taken. The confidence in these odd and striking results would therefore be very much improved if an alternate shifting task was used with levers, nose pokes, or past ASST standards that used a non homecage for the test and did not physically remove the mouse from the cage 'mid-cognition.' Then we might know that the odd and exciting results are robust and not a specific to Sohal lab only protocols.

Reading the submitted manuscript, we still don't know what happens to neural connections in terms of plasticity after stimulation, leaving no explanation of the 24h effects on behavior. This lack of information about the plasticity effect where a carry over effect is seen on subsequent days is a major hole. The new figure in the rebuttal is in the right direction but it is not actually included in the paper and it is possibly underdeveloped. Without any information, there is a major hole that in my opinion makes it too premature for this journal. It seems worth it to invest in this publication how the lasting effects manifest, or to publish this second manuscript which works these effects out with Nature. This will be the more valuable information.

The behavioral effect on non task specific stimulation in later shifting is I think still an open question. It is too bad experiments that could have addressed this were done too long ago to have included later test days, but perhaps they could be redone to answer this important question. If plasticity occurs overnight that creates the 'magic' changes then it may affect RS shifting on the next day. This is an important to clarify the state of the circuit required for stimulation effects on behavior to occur. It has implications for speculations about the phenomenon under study and what is important for clinical or therapy translation.

Statistics and plots comparing groups directly in the main data and figures (for behavior and functional metrics) would also strengthen the paper. I do not think aesthetic reasons are enough

reason to leave these out. Individual data could be shown with lighter lines or in supps. Thus in revision 2, I am still left feeling there are too many uncertainties and unknowns to feel confident about the claims of this paper for a premier journal.

Referee #2 (Remarks to the Author):

The authors have provided important additional analyses, arguments and clarification in their rebuttal that addressed all my remaining points. I appreciate their efforts in re-analyzing the data and congratulate them for this nice piece of work.

Author Rebuttals to Second Revision:

Referee #1 (Remarks to the Author):

The authors have argued in the past rebuttal how important this breakthrough finding may be to the field in terms of understanding basic mechanisms of shifting. This argument made by the authors underscores the need to ensure these effects are generalizable and can be replicated.

We still don't know if this stimulation effect works in any other task context. The authors perhaps misunderstood my request to use a different task. They did add an experiment which shortened the time-out in response to my concerns. However, they did not test more commonly established shifting task designs to strengthen their case that theirs is a robust finding. The current behavioral paradigm (at 2 minutes or 30 seconds) still relies on removing the mouse from its home cage and leaving it out for sometime after an error. This handling procedure inside of a behavioral test of prefrontal function is highly invasive and may operate through special punitive effects that are custom to the Sohal lab and their favorite protocols but not replicated in other more standard protocols. Home cage testing and removal from the test cage in the task testing time is not typically used in rodent protocols testing prefrontal function. I am concerned this removal and possibly also use of the home cage as the test cage are possibly critical features of the protocol that have gone underrecognized. These details are also hard to notice in the streamlined version of the paper and past methods published for this group. Removal is used in water maze testing hippocampal function but these time out periods are not when measures of brain function are taken. The confidence in these odd and striking results would therefore be very much improved if an alternate shifting task was used with levers, nosepokes, or past ASST standards that used a non home cage for the test and did not physically remove the mouse from the cage 'mid-cognition.' Then we might know that the odd and exciting results are robust and not a specific to Sohal lab only protocols

Our response: A key experimental detail is that the mouse is removed from its home cage after both correct and incorrect choices (not just errors as the Referee seems to be assuming). This is shown clearly and explicitly in past methods published from our group, e.g., Fig. 1c of Cho et al., *Nat Neurosci*, 2020. One can of course raise any number of arguments about the details of an experimental protocol. For example, the referee questions why we test mice in their home cage. By contrast, we would actually argue that home cage testing is more naturalistic, and less likely to introduce anxiety-related confounds, than behavioral testing performed in a separate 'testing arena.' We have further addressed the Referee's concern that the behavioral effects we observe after inhibiting callosal PV+ projections reflect the difficulty of maintaining a newly learned rule while the mouse is in a holding cage during the inter-trial interval (ITI) three ways: (1) by showing that inhibiting callosal PV+ projections does not affect learning when we use identical procedures but perform a rule reversal instead of a rule shift; (2) by showing that we observe the same effect even when we shortened the ITI from ~2 min to 30 sec; (3) in new experiments (performed for this revision) which delivered inhibition only during the ITI and found no effect on behavior on either the day that inhibition is delivered or the next day (Reviewer Fig. 1, also shown in Extended Data Fig. 2l-o). These results are all consistent with our finding that inhibiting callosal PV+ projections disrupts the normal evolution of activity patterns occurring during the 10 seconds following incorrect decisions during rule shifts (during which time the mouse is still in the home cage) (Extended Data Fig. 10r). In other words, inhibiting callosal PV projections seems to affect learning that occurs in the home cage at the end of a trial, and therefore does not seem to be some special process that only occurs when the mouse is placed in the holding cage during the ITI that is responsible for the effects we observe.

We have added the following sentences describing these results to the manuscript (lines 186-190): **"This suggests that inhibiting callosal PV+ projections disrupts learning which occurs, at least in part, during each trial. Consistent with this, inhibiting callosal PV+ projections only during inter-trial intervals did not affect rule shift learning (Extended Data Fig. 2l-o)."** Of course,

as with any paper, we can only be 100% confident about the particular task we tested. Therefore, we have added the following sentences near the end of the Discussion (lines 271-273): **“Future studies could also explore the role of callosal PV+ projections (and potential plasticity) using behavioral paradigms which differ from those used here in key respects (e.g., automated lever-press tasks).”**

REVIEWER FIGURE 1 (from Extended Data Figure 2): **i**, Experimental design: Day 1, no light delivery; Day 2, light delivery during the inter-trial interval (ITI) during the RS for optogenetic inhibition of callosal PV terminals; Day 3, no light was delivered; Day 4, continuous light during the RS. **m**, Representative image showing DIO-eNpHR-mCherry (DIO-eNpHR) expression in one mPFC and a fiber-optic cannula implanted in the contralateral mPFC. Scale bar, 100 μ m. **n, o**, Optogenetic inhibition of mPFC callosal PV terminals impairs rule shift performance in DIO-eNpHR-expressing mice only when delivered both during the trial and inter-trial intervals of the RS ($n = 6$) compared to controls ($n = 3$) (two-way ANOVA (task day \times virus); interaction: $F_{(3,28)} = 23.31, P < 0.0001$).

n, Performance of controls was not different from eNpHR-expressing mice on Day 1 (post hoc $t_{(28)} = 0.3489, P > 0.9999$), Day 2 (post hoc $t_{(28)} = 0.6978, P > 0.9999$), nor Day 3 (post hoc $t_{(28)} = 0.4652, P > 0.9999$).

o, Inhibition disrupts rule shift performance in DIO-eNpHR-expressing mice compared to controls when light is delivered continuously throughout the RS (post hoc $t_{(28)} = 9.886, P < 0.0001$). Two-way ANOVA followed by Bonferroni post hoc comparisons was used. **** $P < 0.0001$.

Reading the submitted manuscript, we still don't know what happens to neural connections in terms of plasticity after stimulation, leaving no explanation of the 24h effects on behavior. This lack of information about the plasticity effect where a carry over effect is seen on subsequent days is a major hole. The new figure in the rebuttal is in the right direction but it is not actually included in the paper and it is possibly underdeveloped. Without any information, there is a major hole that in my opinion makes it too premature for this journal. It seems worth it to invest in this publication how the lasting effects manifest, or to publish this second manuscript which works these effects out with Nature. This will be the more valuable information.

Our response: **Redacted** We have acknowledged this in the sentence added near the end of the Discussion (lines 269-271): **“It will be important to explore how this plasticity depends on changes in callosal PV+ projections, callosally-projecting PV+ neurons, and/or additional circuit elements.”**

The behavioral effect on non task specific stimulation in later shifting is I think still an open question. It is too bad experiments that could have addressed this were done too long ago to have included later test days, but perhaps they could be redone to answer this important question. If plasticity occurs overnight that creates the ‘magic’ changes then it may affect RS shifting on the next day. This is an

important to clarify the state of the circuit required for stimulation effects on behavior to occur. It has implications for speculations about the phenomenon under study and what is important for clinical or therapy translation.

Our response: As requested, for this revision we performed new experiments which delivered inhibition only during the ITI and found no effect on behavior on either the day that inhibition is delivered, or the next day (Reviewer Fig. 1, also shown in Extended Data Fig. 2I–o). This suggests there is no plasticity that creates “magic” changes which only appear on the next day, and clarifies that inhibition of callosal PV+ projections needs to include the trial periods in order to impact learning. As noted above, this is consistent with our previous finding that inhibiting callosal PV+ projections disrupts the evolution of activity patterns that normally occurs within a single trial.

Statistics and plots comparing groups directly in the main data and figures (for behavior and functional metrics) would also strengthen the paper. I do not think aesthetic reasons are enough reason to leave these out. Individual data could be shown with lighter lines or in supps. Thus in revision 2, I am still left feeling there are too many uncertainties and unknowns to feel confident about the claims of this paper for a premier journal.

Our response: For the previous revision, we added the statistics comparing groups directly, as requested by the Reviewer. Re: figure, we appreciate the point the Referee is making, but we are not excluding plots which compare groups directly purely for aesthetic reasons. Rather, as shown in our previous responses, those plots with overlaid groups are very difficult to interpret – specifically it is difficult to discern the across-day trends within each group. In all cases, we only plotted individual data points (never group means), so we don’t see how this problem would be addressed by showing individual data with lighter lines as the Reviewer suggests (this would just make the entire figure lighter).

Referee #2 (Remarks to the Author):

The authors have provided important additional analyses, arguments and clarification in their rebuttal that addressed all my remaining points. I appreciate their efforts in re-analyzing the data and congratulate them for this nice piece of work.

Reviewer Reports on the Third Revision:

Referees' comments:

Referee #2 (Remarks to the Author):

No further comments.